# Non-decameric NLRP3 reveals a TGN/MTOC-distal pathway of inflammasome activation

María Mateo-Tórtola [1], Inga V. Hochheiser[2,11], Gaopeng Li [1,11], Lukas Funk [1,9,11], Atousa Hashemi [1], Xiao Liu[1], Jane Torp [2], Lena Erlebach [3], András Szolek[1], Jelena Grga [1,4,10], Francesca Bork [1], Jana S. Müller[1], Deborah Kronenberg-Versteeg [3], Matthias Geyer [2], Alexander N. R. Weber [1,4,5,6] ✉ & Ana Tapia-Abellán [1,4,6,7,8] ✉

The NLRP3 inflammasome contributes to a wide range of conditions from infections to Alzheimer's disease. NLRP3 forms an inactive decameric cage, that upon interaction with the trans-Golgi network (TGN) and microtubule organization center (MTOC), leads to inflammasome activation, yet whether non-decamer NLRP3 species form functional inflammasomes remains unclear. Here, we design a NLRP3 exon 3 deletion variant that forms low molecular weight NLRP3 assemblies. Spatially and dynamically highly resolved microscopy in THP-1 and human macrophages shows that nigericin, a K⁺-dependent NLRP3 stimulus, can trigger two distinct activation pathways: (i) the rapidly engaged decameric cage-dependent pathway; and (ii) a decameric cage-independent, TGN/MTOC-distal, and slow-reacting pathway employed by low molecular weight NLRP3 species, that dominates in human neutrophils. Collectively, our results delineate two parallel yet biologically distinct NLRP3 activation pathways, thereby providing a framework to understand NLRP3-driven inflammation across a wide range of pathological context and cell types.

Inflammation is an essential innate immune response that enables rapid defense against infection and tissue damage. Central to this process are inflammatory cytokines of the interleukin-1 (IL-1) family, whose maturation and release are controlled by cytosolic multiprotein complexes named inflammasomes, assembled in immune cells such as macrophages or neutrophils[1]. Among them, the nucleotide-binding domain and leucine-rich repeat pyrin domain-containing protein 3

(NLRP3) inflammasome is the most extensively studied and is strongly associated with human disease. *NLRP3* gain-of-function mutations cause autoinflammatory disorders collectively termed Cryopyrin-Associated Periodic Syndromes (CAPS)[2,3]. Furthermore, aberrant activation of NLRP3 contributes to a broad spectrum of conditions ranging from gout to Alzheimer's disease[4]. The clinical relevance of this pathway has driven extensive efforts to elucidate the molecular mechanism

[1]Institute of Immunology, Department of Innate Immunity, University of Tübingen, Tübingen, Germany. [2]Institute of Structural Biology, University of Bonn, Bonn, Germany. [3]German Center for Neurodegenerative Diseases and Hertie Institute for Clinical Brain Research, University of Tübingen, Tübingen, Germany. [4]CMFI—Cluster of Excellence (EXC 2124). "Controlling Microbes to Fight Infections", Tübingen, Germany. [5]German Cancer Consortium (DKTK), DKTK Partner Site Tübingen, Tübingen, Germany. [6]iFIT—Cluster of Excellence (EXC 2180). "Image-Guided and Functionally Instructed Tumor Therapies", University of Tübingen, Tübingen, Germany. [7]Group of Immunology, Department of Biochemistry and Molecular Biology (B) and Immunology, Campus de Ciencias de la Salud, Universidad de Murcia, Murcia, Spain. [8]Group of Molecular Pathology and Pharmacogenetics, Department of Pathology, Instituto Murciano de Investigación Biosanitaria (IMIB), Hospital General Universitario Santa Lucía, Cartagena, Spain. [9]Present address: Institute for Clinical Chemistry and Clinical Pharmacology, University and University Hospital Bonn, Bonn, Germany. [10]Present address: Toulouse Biotechnology Institute. University of Toulouse, CNRS, INRAE, INSA Toulouse, Toulouse, France. [11]These authors contributed equally: Inga V. Hochheiser, Gaopeng Li, Lukas Funk. ✉e-mail: alexander.weber@uni-tuebingen.de; ana.tapia@um.es

underlying NLRP3 inflammasome activation[5]. NLRP3 responds to a wide range of microbial and sterile stimuli, and thus functions as a general sensor of cell homeostasis, whose minimum requirement to become active appears to be the intracellular loss of potassium ($K^+$)[6–8]. Besides that, NLRP3 is activated by a $K^+$-independent stimulus named imiquimod[9]. However, the molecular mechanisms of this activation step are not well understood. NLRP3 is a tripartite protein composed of the signal-driving pyrin (PYD) domain, the nucleotide binding and oligomerization (NACHT) domain and the leucin-rich repeat (LRR) domain. Upon activation, NLRP3 undergoes conformational changes that enable recruitment of the adaptor ASC, leading to the formation of prion-like filaments that engage and activate the effector protein, caspase-1, forming an active NLRP3 inflammasome[10–12]. Active caspase-1 subsequently cleaves inflammatory cytokines such as pro-IL-1β and pro-IL-18 and the pore-forming protein gasdermin D (GSDMD), causing pyroptotic cell death and release of IL-1β, IL-18 and other alarmins[13,14].

Recently, cryo-EM studies changed previous assumptions, according to which inactive NLRP3 was monomeric and only the active NLRP3 an oligomer. Namely, inactive NLRP3 was shown to form steady-state oligomeric 'cages' (decamers for human and dodecamers for mouse NLRP3) which shield the effector PYD domain in the center of the cage and thereby prevent ASC recruitment[15–17].Upon activation, these cages are proposed to reorganise into disk-like structures that permit ASC engagement[12]. However, non-oligomeric NLRP3 species have also been observed in macrophages extracts[16], and an NLRP3 construct lacking the LRR, a domain critical for cage formation, still formed active inflammasomes, complicating the understanding of how non-oligomeric species can function[18]. To add more complexity, the dynamic subcellular localization of NLRP3 has emerged as key determinant of its activation[5,19]. In macrophages, NLRP3 cages were proposed to traffic from the dispersed trans-Golgi network (TGN) to other membranous compartments and ultimately to the microtubule organization center (MTOC), where inflammasome assembly occurs[3,20–23]. Here, polybasic regions in inactive NLRP3 seem critical for TGN recruitment and subsequent NLRP3 activation[16,24]. However, the biological importance of the NLRP3 decamer in the proposed TGN/MTOC activation pathway and the role of lower molecular weight non-cage NLRP3 species remain unclear, particularly as many studies have relied on ASC-deficient cellular systems[20,23].

To address whether non-decameric NLRP3 species can form functional inflammasomes, we generate a NLRP3 exon 3 deletion variant that limits cage formation and enables the study of lower molecular weight NLRP3 assemblies. Using finely time-resolved live cell microscopy in THP-1, we delineate two spatially and mechanistically distinct NLRP3 activation pathways: a known canonical pathway associated with decameric cages and TGN/MTOC localization; and a novel TGN/MTOC-distal pathway employed by lower molecular weight NLRP3 species. We further show that these pathways are differentially engaged with different kinetics and depending on the nature of the activating stimulus and cellular context. Notably, the TGN/MTOC-distal pathway coexists with the canonical in macrophages and induced pluripotent stem cell-derived microglia but appears to predominate in human neutrophils. This work provides a revised conceptual basis for NLRP3 signaling, with potential implications for therapeutic targeting of inflammasome-driven diseases.

## Results

### Non-decameric NLRP3 is not recruited to TGN membranes before activation

Between the PYD and the NACHT domains, NLRP3 contains two adjacent polybasic regions described to be critical for membrane association: the KMKK motif (residues 131-134 encoded by exon 3, Fig. 1A, blue box) and the polybasic region (residues 135-147 which together with the FISNA and NACHT domains is encoded by exon 4, Fig. 1A, yellow box). In the decamer structure of inactive NLRP3, we observed

that five of the polybasic motifs are each exposed on the same side (Fig. 1B, dark blue alpha helices), potentially enabling TGN binding via concerted orientation and increased avidity of multiple binding sites, similar to clustered septin polybasic motifs[25]. Interestingly, NLRP3 lacking the PYD domain and its linker forms a hexamer[24] with sub-optimal/buried polybasic region orientation (Fig. 1B), which prevents cage formation and TGN recruitment. Thus, PYD and/or the linker region seem critical for cage organization and membrane association. Since the PYD domain is essential for downstream signaling, we decided to solely delete the linker region (residues 95-134) in order to prevent decamer formation. This approach is based on the principle that exons often encode distinct functional blocks within a protein, which was specifically shown for NLRP3[26]. The linker comprises 40 amino acids including the KMKK motif and is fully encoded by a single short exon in *NLRP3*, exon 3 (Fig. 1A), that is absent in some NLRs such as NLRP9, NLRP10 and NLRP11 which can form functional inflammasomes[27–29] (Supplementary Fig.1A). Furthermore, chicken (*Gallus gallus*) and zebrafish (*Danio rerio*) NLRP3s have been shown to form functional inflammasomes despite lacking this exon[30,31,32] (Supplementary Fig. 1B). The second polybasic region within the FISNA domain belongs already to a different exon, exon 4, and was left intact to fulfil its reported function in NLRP3 activation[12] (Fig. 1A, B).

To explore whether a NLRP3 protein lacking the exon 3 linker region (termed Δexon3), forms decamers, human NLRP3(Δexon3) protein fused to maltose-binding protein (MBP) was expressed in baculovirus-infected *Sf*9 insect cells and purified to homogeneity. Size-exclusion chromatography (SEC) revealed one elution peak close to the void volume, representing an undefined high-molecular-weight aggregate also observed for WT NLRP3[17], and a wider peak ranging from 158 to 670 kilodaltons (kDa) (Fig. 1C). By comparison WT NLRP3 eluted in a second peak with a predicted MW bigger than 670 kDa. Thus, purified NLRP3(Δexon3) elutes in lower molecular weight species than decameric WT NLRP3. Negative-stain electron microscopy (EM) images of MBP-NLRP3(Δexon3) confirmed the absence of the typical decamer structure and presence of smaller species (Fig. 1C). Additionally, multi-angle light scattering (SEC–MALS) suggested an average molecular mass of 193-319 kDa for MBP-NLRP3(Δexon3), indicative of monomers and dimers (Fig. 1D). Interestingly, replacement of exon 3 with a short flexible linker sequence (GSGAGG) resulted in slightly bigger, albeit still non-decameric species (Fig. 1E). Thus, NLRP3(Δexon3) is unable to form the typical oligomeric cage structure, suggesting that the 40 aa linker region is required for decamer formation.

As the NLRP3(Δexon3) protein was soluble and fully folded when purified, we explored the subcellular localization of NLRP3(Δexon3) in cells. THP-1 *NLRP3* KO cells were reconstituted with NLRP3 WT or Δexon3 linked C-terminally via a TSGSGSGSG flexible linker to the monomeric fluorophore mNeonGreen (mNG), in line with previous studies that have tagged NLRP3 C-terminally without compromising function[16]. The localization of NLRP3 WT-mNG closely resembled endogenous NLRP3 as determined by confocal microscopy (Supplementary Fig. 2A). Interestingly, compared to WT, NLRP3(Δexon3)-mNG cells showed decreased NLRP3 localization at the TGN as expected (Fig. 1F, quantified in 1G). In confirmation, HeLa cells stably expressing NLRP3(Δexon3)-mNG also showed lower colocalization with the TGN46 marker (Supplementary Fig. 2B, quantified in Supplementary Fig. 2C). Whereas the KKKK motif in mouse Nlrp3 is critical for TGN binding and activation[20], the KMKK motif in human (Fig. 1A blue box) was dispensable as the removal of positive charges (KMKK > AMAA) did not alter TGN localization[11] (Supplementary Fig. 2B, C). Thus, the second polybasic region appears sufficient for human NLRP3 to localize to TGN membranes: Since the Δexon3 construct preserves the second polybasic region (located in the FISNA domain), but is still unable to localize to the TGN, we conclude that the role of the exon 3 linker is to place the second polybasic region for proper membrane lipid association in a decameric cage arrangement. Although more

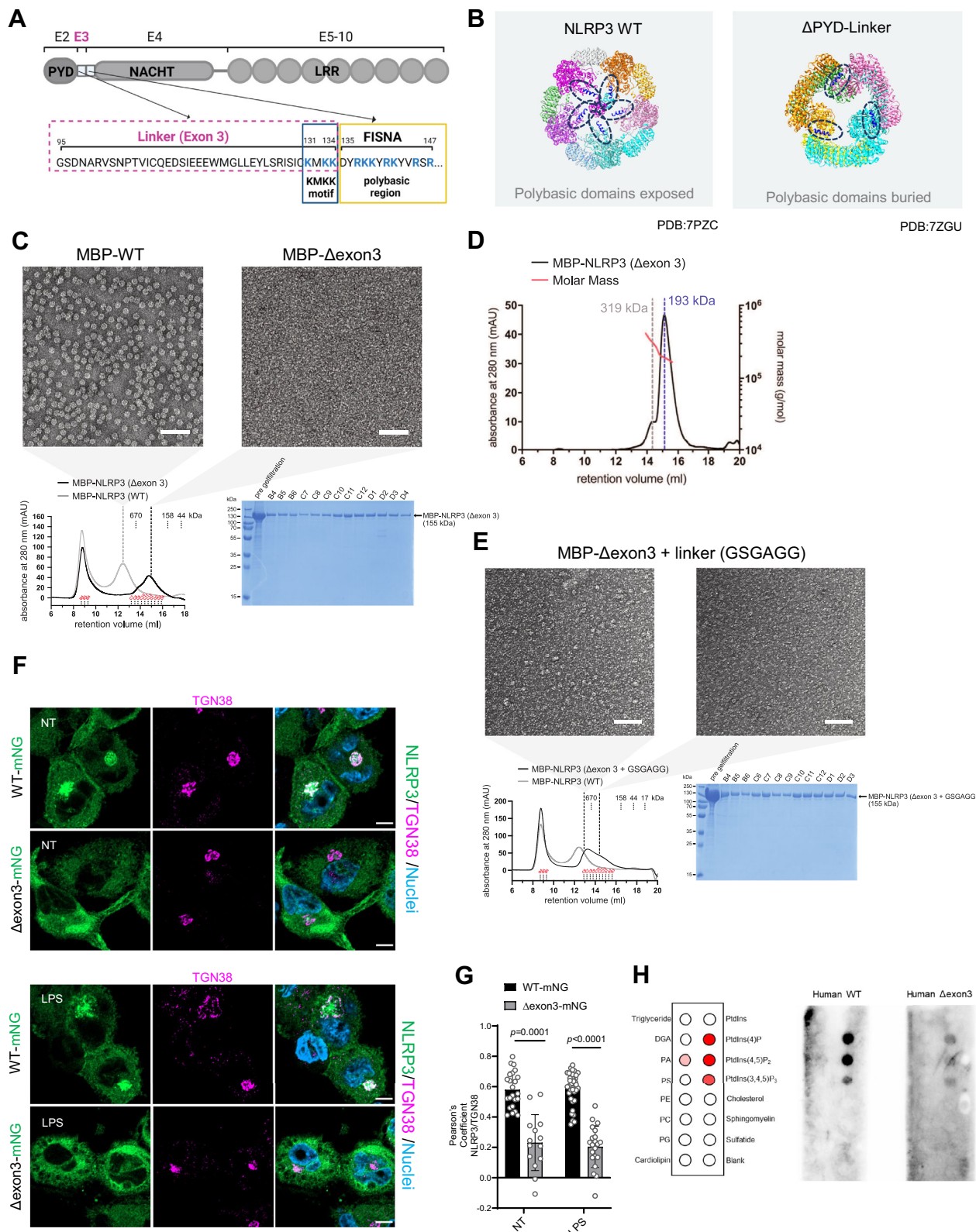

details about human NLRP3-lipid interactions await elucidation, in line with this, lipid strip binding assays confirmed that in a side-by-side comparison, purified human WT NLRP3 bound the same lipids as its mouse counterpart: (PtdIns-(4)-P, PtdIns-(4,5)-P$_2$, PtdIns-(3,4,5)-P$_3$, sulfatide, phosphatidic acid (PA) and cardiolipin (Supplementary Fig. 2D), whereas NLRP3(Δexon3) significantly reduced this ability (Fig. 1H). Supporting our in vitro experiments, size exclusion

chromatography of lysates obtained from these THP-1 cells confirmed that the majority of NLRP3(Δexon3)-mNG protein eluted in a manner consistent with predominantly monomers and dimers, whereas NLRP3 WT-mNG and endogenous NLRP3 eluted mainly in larger oligomers (Fig. 2A-E quantified in Fig. 2F), in agreement with in vitro data (*cf.* Figure 1). Together, these results indicate that removal of the exon 3 linker sequence shifts the NLRP3 equilibrium toward smaller molecular

**Fig. 1 | NLRP3(Δexon3) is mostly organized in monomers and dimers with reduced ability to bind TGN membranes. A** Scheme of exons and domain distribution of human NLRP3. Created in BioRender. Weber, A. (2026) https://BioRender.com/ghhupge. **B** Structure of the human double-ring NLRP3 decamer (left, pdb 7PZC) and human NLRP3 ΔPYD-linker hexamer (right, pdb 7VTP). Second polybasic regions are highlighted and colored in dark blue. Different colors of the NLRP3 structures indicate different protomers. **C** Negative-stain EM micrographs of the MBP-NLRP3 WT peak 2 (decamer) and the MBP-NLRP3(Δexon3) peak 2 (showing a lawn of NLRP3 particles smaller than the decamer). Scale bar 100 nm. Below, SEC of MBP-NLRP3 WT (grey trace) and MBP-NLRP3(Δexon3) (black trace). In red, expected molecular weights based on a gel filtration standard and elution fractions of MBP-NLRP3(Δexon3) further analyzed by Coomassie-stained SDS-PAGE. **D** SEC-MALS of recombinant human MBP-NLRP3(Δexon3) peak 2 on a Superose 6 Increase 10/300 GL column. The estimated molar masses are indicated by the red line. The calculated mass of the MBP-NLRP3(Δexon3) monomer is 154.8 kDa. **E** Negative-stain EM micrographs of the left and right part of the peak 2 of MBP-NLRP3(Δexon3) + GSGAGG linker. The left part of peak 2 already shows a more pronounced capability to form higher-order oligomers in the presence of GSGAGG linker. Scale bar 100 nm. SEC as in C of MBP-NLRP3 WT (grey trace) and MBP-NLRP3(Δexon3) + GSGAGG (black trace). On the right, Coomassie-stained SDS-PAGE analysis of the SEC run of MBP-NLRP3(Δexon3) + GSGAGG. **F** Representative immunofluorescence micrographs of LPS primed or not (NT) THP-1 *NLRP3* KO cell line reconstituted with NLRP3 WT- or Δexon3-mNG (green). TGN stained with a specific TGN38 antibody (magenta) and nuclei with Hoechst 33342 (blue). Scale bar 10 μm. *N* = 4 independent experiments. **G** Co-localization levels between WT- or NLRP3(Δexon3)-mNG and TGN38 in cells treated as in **F**. Each bar represents the mean ± SD and each datapoint represent single cells combined from 3 independent experiments. One-way ANOVA. **H** In vitro lipid strip assay of purified human WT and Δexon3 NLRP3. On the left, the arrangement of different lipids on the membrane is shown. Lipids bound by WT NLRP3 are highlighted in varying intensities of red, corresponding to the strength of binding signal. *N* = 2 independent experiments.

---

weight assemblies in cells and these smaller species are unable to engage the TGN in resting conditions.

### Non-decameric NLRP3 forms TGN- and MTOC-distal specks in response to nigericin

Our previous data indicate that NLRP3(Δexon3) might provide a unique opportunity to study the functionality of an NLRP3 inflammasome that is not cage- and membrane binding-associated. Therefore, LPS-primed THP-1 NLRP3 WT- or Δexon3-mNG cells treated with nigericin confirmed both cell lines able to form NLRP3 specks (Fig. 3A). Interestingly, while NLRP3(Δexon3) mostly formed TGN-distal specks, WT NLRP3 presented both, TGN-associated and distal speck formation (Fig. 3A, quantified in Fig. 3B), as assessed by co-localization (or not for NLRP3(Δexon 3)) with the TGN via TGN38 specific staining.

Furthermore, upon nigericin stimulation, WT- or NLRP3(Δexon3)-mNG both co-localized, i.e. engaged, ASC, forming NLRP3-ASC double-positive specks as a hallmark of functional inflammasomes[1] (Fig. 3C). In WT NLRP3-mNG cells, we again observed two clear phenotypes: NLRP3-ASC platforms surrounded by multiple smaller NLRP3-ASC puncta, resembling the TGN staining imaged above (Fig. 3A, C NLRP3 WT example 1); and single, well-delineated NLRP3-ASC platforms distal to TGN staining (Fig. 3A, C NLRP3 WT example 2). However, in the case of NLRP3(Δexon3)-mNG, only the TGN-distal type could be observed even upon profiling approximately 100 cells presenting specks from a total of 600 cells (Fig. 3A, C). When comparing the single TGN-distal NLRP3-ASC platforms from WT- *vs* Δexon3-mNG NLRP3-expressing cells, no differences were detected between them in terms of fluorescence intensity and area (Supplementary Fig. 3A). Additionally, in regular THP-1 WT cells, the ASC platform surrounded with multiple and smaller ASC puncta also co-stained for TGN membranes (TGN38 staining), whereas the ASC single platform typically did not (Supplementary Fig. 3B).

As the MTOC was proposed as the final destination of NLRP3 to culminate in the formation of an active inflammasome[21], we also stained for the specific MTOC marker, γ-tubulin[33]. Again, nigericin stimulation produced two clear phenotypes for NLRP3 WT-mNG specks: proximal and distal to the MTOC (Fig. 3D, quantified in Fig. 3B). Notably, staining for the MTOC using the centrosome marker pericentrin-specific antibody confirmed the localization of NLRP3-ASC specks within this region (Supplementary Fig. 3C). However, in line with our previous results, NLRP3(Δexon3)-mNG specks were predominantly observed distal to the MTOC (Fig. 3D). For NLRP3(Δexon3), only a minor fraction co-stained with γ-tubulin, potentially due to MARK4-dependent recruitment via its PYD[34], but these specks were not ASC-positive. Thus, whereas WT NLRP3 consistently formed both TGN/MTOC-associated and -distal functional NLRP3 specks upon nigericin activation, NLRP3(Δexon3) predominantly formed TGN/MTOC-distal ones. Nonetheless, the quantitative analysis (Fig. 3B) indicated a minor subpopulation of NLRP3(Δexon3) specks that colocalized with TGN or MTOC markers, suggesting that partial or transient membrane/microtubule interactions may still occur in these cells. These data led us to hypothesize that the NLRP3 decamer might be associated with the described TGN/MTOC-associated inflammasome activation pathway, whereas smaller NLRP3 species might activate and form specks via a separate, non-membrane associated pathway more likely to involve the cytosol as a signaling competent environment.

To prove that the observation of two types of functional NLRP3 pathways was not a consequence of a NLRP3 overexpression phenomenon, we endogenously disrupted the expression of the NLRP3 linker sequence by skipping the *NLRP3* exon 3 in THP-1 cells using an intra-exonic anti-oligonucleotide (AON), an approach used for exon 5[26]. Exon 3 was specifically targeted and skipped in about 80% of transcripts without affecting the abundance of other NLRP3 exons like exon 7 and hence total mRNA NLRP3 levels (Fig. 3E). We further validated the effect of exon 3-skipping AONs at the protein level, which revealed an expected lower-migrating NLRP3 species consistent with Δexon3 (Supplementary Fig. 3D). Strikingly, in terms of IL-1β release from AON-treated THP-1 cells no significant differences were detected whether NLRP3 exon 3 was skipped or not, while exon 5 skipping significantly affected early IL-1β production as expected[26] (Fig. 3F). In addition, TNF-α release was not affected by any exon-skipping approaches (Fig. 3G). Collectively, these results indicate that an NLRP3 protein organised into smaller assemblies (i.e. the Δexon 3 protein) was still able to form a functional inflammasome, albeit distal to the TGN.

### Microtubule-associated NLRP3 specks accelerate inflammasome activation

Having confirmed that NLRP3(Δexon3) is functional in cells, we investigated its microtubule dependence in detail. Highly resolved live cell imaging experiments using the specific probe for microtubules, SIR-tubulin[35], showed that upon LPS and nigericin stimulation only the WT NLRP3-mNG speck consistently associated with the MTOC (Fig. 4A left and Supplementary Movie 1, Supplementary Fig. 3E for WT NLRP3 speck in the MTOC example 2), while the NLRP3(Δexon3)-mNG speck appeared in different cellular locations in the cytosol, as the signal was completely distal from the MTOC (Fig. 4B and Supplementary Movie 2). Indeed, we also visualized the formation of a distal MTOC WT NLRP3-mNG speck over time, indicating the dynamic presence of both types of specks—and hence pathways—for the WT NLRP3 (Fig. 4A right and Supplementary Movie 3). Taking a closer look at WT NLRP3, we observed NLRP3 to be recruited to vesicular structures upon nigericin stimulation which trafficked and collapsed at the MTOC by 25 min, resulting in actual NLRP3 speck formation (Fig. 4A left). Interestingly, MTOC-associated WT NLRP3-mNG specks formed faster (i.e. within 30 min), while MTOC-distal specks formed more slowly (i.e.

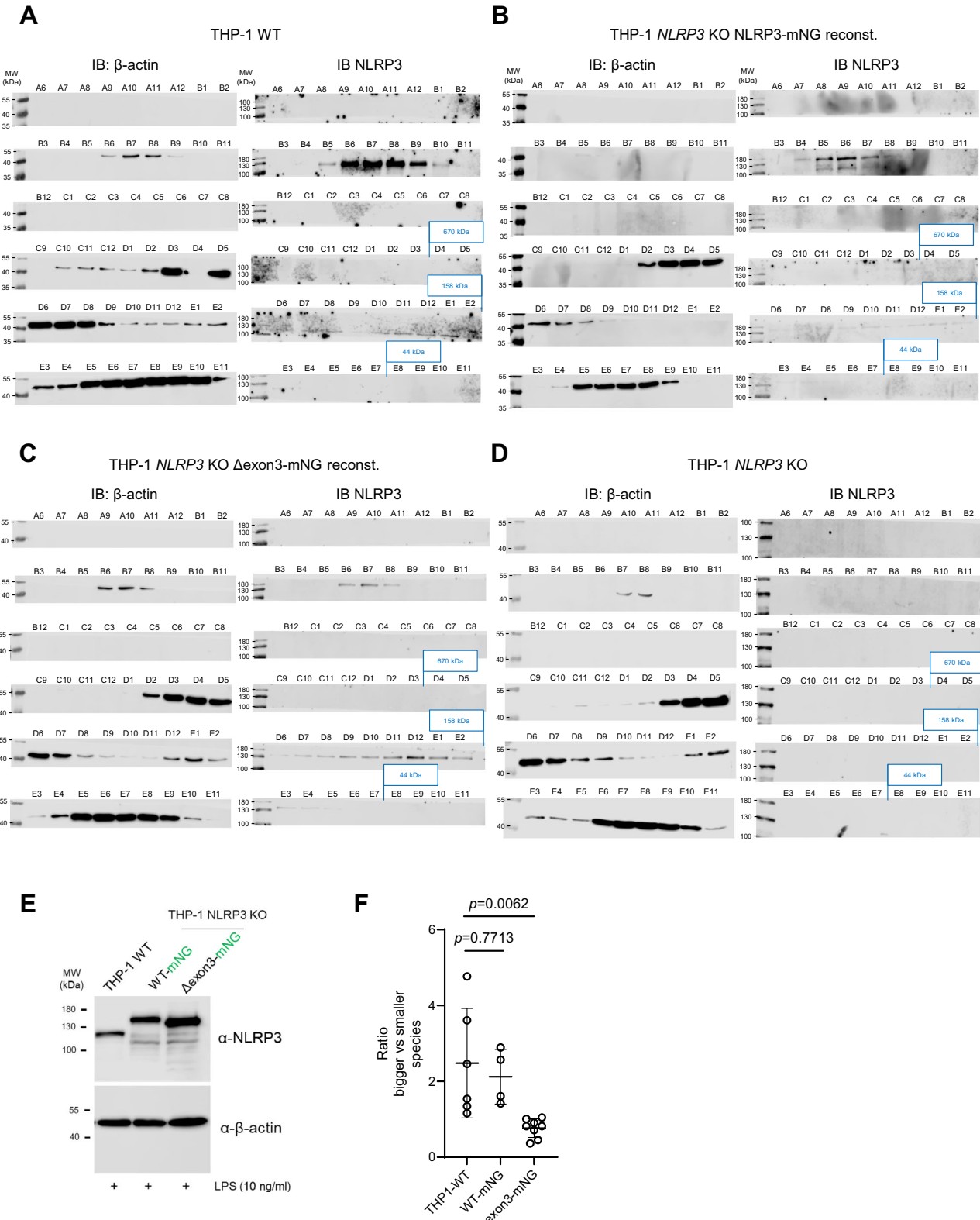

**Fig. 2 | NLRP3(Δexon3)-mNG elutes predominantly in monomers and dimers, whereas endogenous and WT-mNG NLRP3s in larger oligomers. A–D** 150 µl of each lysate prepared by sonication from LPS-primed normal THP-1 cells (**A**), *NLRP3* KO THP-1 reconstituted with WT NLRP3-mNG (**B**), with Δexon-mNG (**C**) or non-reconstituted (**D**) were separated with identical buffer, flow and fractions collection settings. Fractions B6-8 were considered oligomers, D11-12 dimers, and E1-2 monomers, according to the elution volumes of the MW standard (see methods). Blue boxes indicate the fraction in which the standards thyroglobulin (670 kDa), γ-globulin (158 kDa) and ovalbumin (44 kDa) eluted. All blots were exposed under the same conditions (exposure time 10 min for NLRP3 and 2 min for β-actin). Representative of at least 3 independent experiments shown. **E** Immunoblot of total cell lysates from THP-1 cells used as input controls for the gel filtration assay shown in **A–C**. **F** Quantification of immunoblot analysis of size exclusion chromatography (mean ± SD, one datapoint per biological replicate combined from 3 independent experiments). One-way ANOVA.

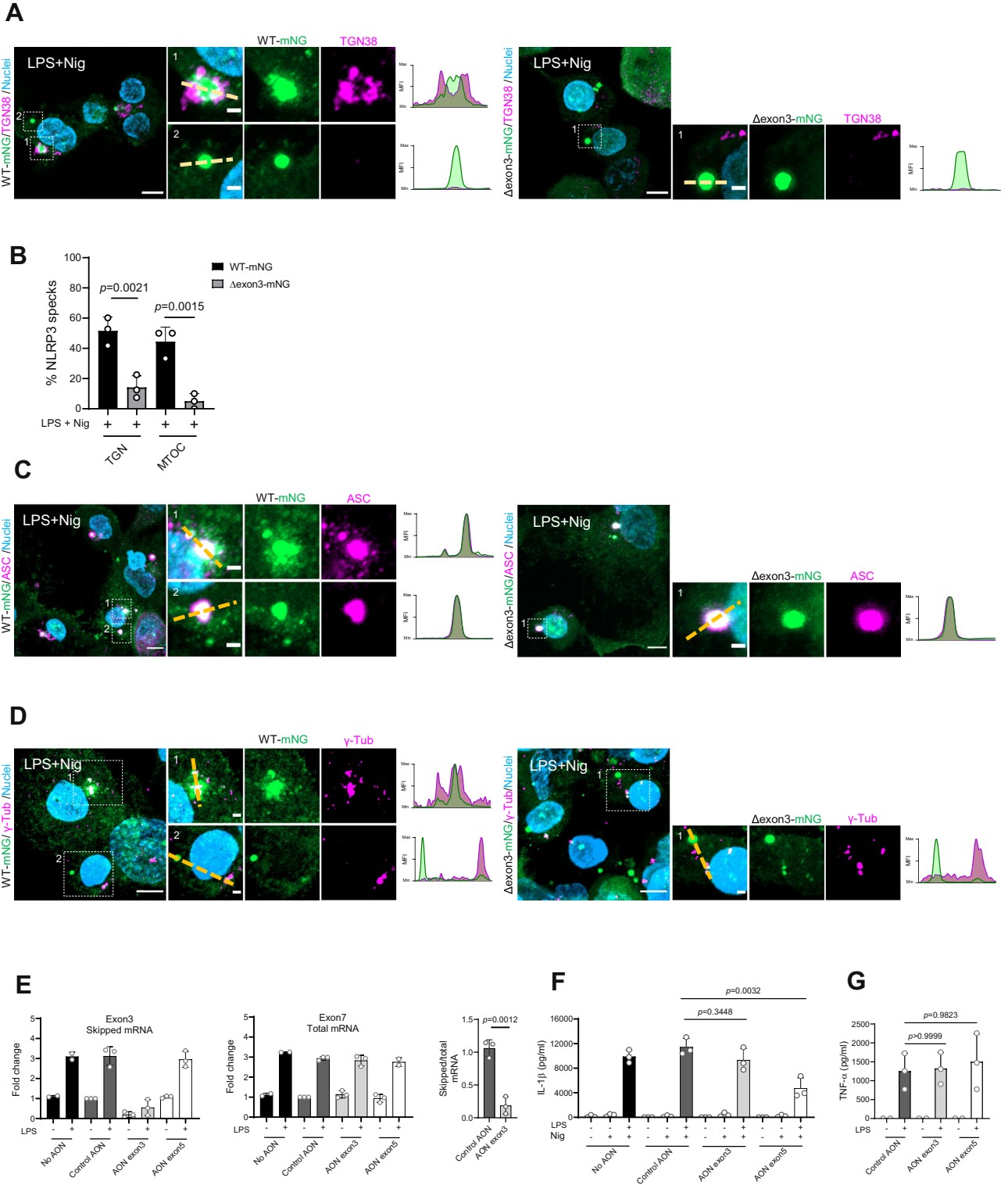

**Fig. 3 | NLRP3(Δexon3) forms a functional inflammasome and shows NLRP3 activation in the absence of membrane binding. A** Representative immuno-fluorescence micrographs of the indicated LPS-primed and nigericin-stimulated THP-1 cell lines. TGN was stained with the TGN38 antibody (magenta) and nuclei with Hoechst 33342 (blue). Scale bar 5 μm, 1 μm for close-ups. Representative line graphs depicting changes in NLRP3 and TGN normalized mean fluorescence intensity (MFI) are shown. **B** Percentage (%) of membrane (TGN) and (MTOC)-associated NLRP3 specks (WT- or Δexon3-mNG) of the microscopy data examples shown in **A** and **D**. Each datapoint represents the average of 3 tiles (3 × 3 FOV) (200 cells approx.). One-way ANOVA. **C, D** As in **A** but endogenous ASC marked with a specific ASC antibody (**C**) or MTOC marked with the γ-tubulin antibody (**D**) (both in

magenta). NLRP3 and ASC or MTOC representative line graphs depicting changes in normalized MFI are shown. **E** RT-qPCR showing endogenous skipping of the *NLRP3* exon 3 in PMA-differentiated THP-1 cells. Upon exon skipping for 24 h, the cells were primed or not with LPS. Left, fold change of *NLRP3* exon 3 mRNA expression. Middle, fold change of *NLRP3* exon 7 mRNA expression used as a negative control. Right, ratio of transcripts lacking exon 3 compared to total NLRP3 mRNA (quantifying exon 7 as control). Two-sided Student's t-test. **F, G** IL-1β or TNF-α release from THP-1 WT treated as in **E** and in **F** further stimulated with nigericin. Skipping of the *NLRP3* exon 5 was used as a positive control of reduced NLRP3 activation. One-way ANOVA. All data were obtained from 3 independent experiments (unless specified) and represented by mean ± SD, unless otherwise indicated.

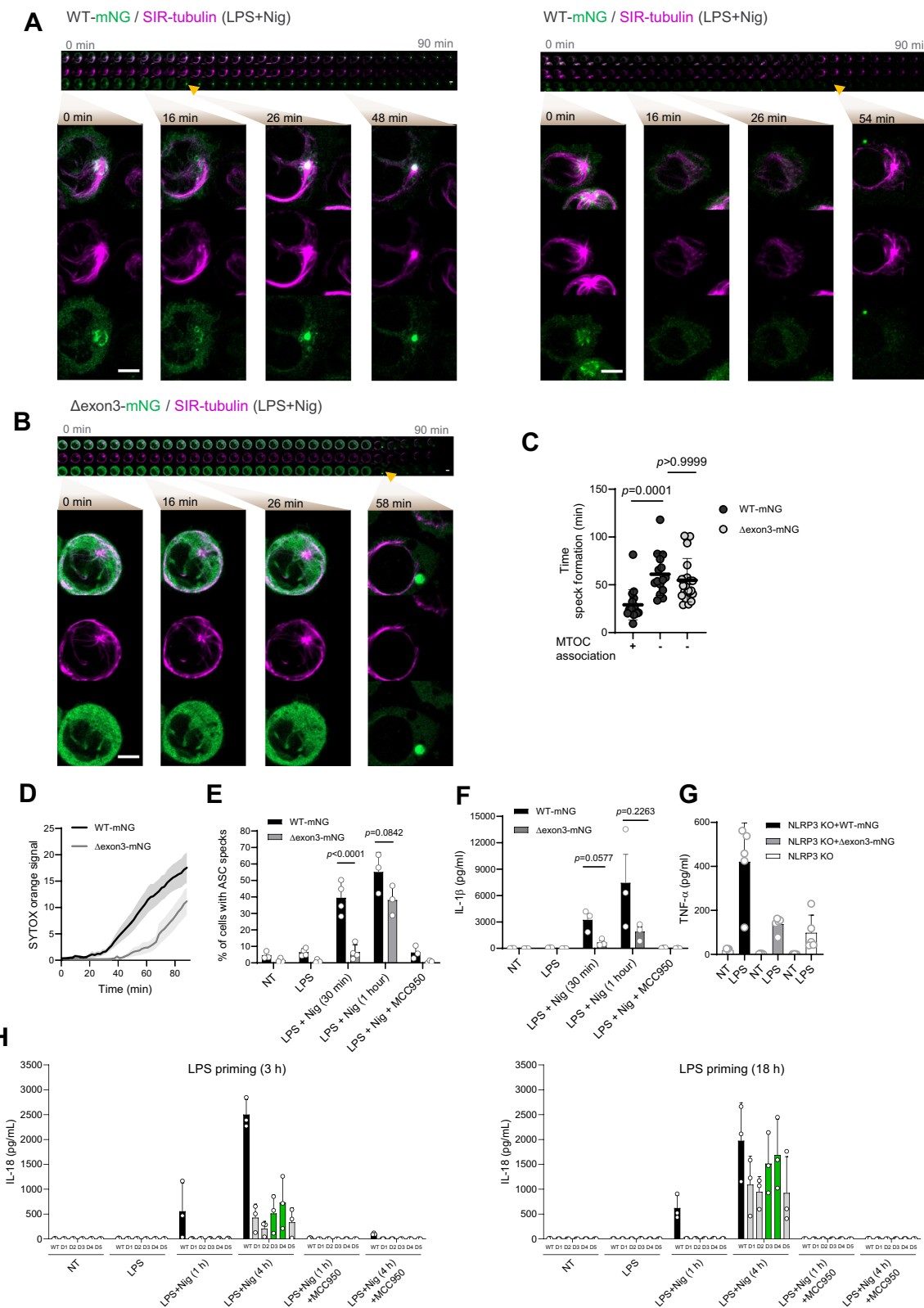

within 55 min) (Fig. 4C). Interestingly, there were no differences in the time to speck formation for the WT- and NLRP3(Δexon3)-mNG MTOC-distal specks (Fig. 4C). This phenomenon observed for WT NLRP3 correlated with accelerated cell death (measured by NLRP3 speck imaging microscopy-based SYTOX uptake, Fig. 4D), faster ASC speck formation (Fig. 4E), and earlier IL-1β release (Fig. 4F), that was abolished in the presence of MCC950 (Fig. 4E, F). TNF-α release was similar

in THP-1 *NLRP3 KO* and NLRP3(Δexon3)-mNG cells but higher from WT NLRP3-mNG cells, possibly due to a more effective LPS priming (Fig. 4G).

In summary, our data demonstrate the existence of two parallel and biologically distinct NLRP3 activation pathways in macrophages: a previously described MTOC-associated pathway, directly related to the NLRP3 decamer; and a second MTOC-distal, somewhat slower, related

**Fig. 4 | Membrane-associated NLRP3 results in earlier speck formation.**
**A** Representative live cell imaging timeline of the MTOC-associated (left) and
-distal (right) NLRP3 specks formation from LPS-primed and nigericin-stimulated
THP-1 WT NLRP3-mNG cells (green). Microtubules were stained with the specific
probe SIR-tubulin (magenta). Arrows denote first occurrence of fully formed
NLRP3 speck. Close-up images at 0, 16, 26 and 48 or 54 min after nigericin addition
are shown. Sequential images acquired every 3 min. Scale bar 5 μm. **B** As in A but
showing the MTOC-distal NLRP3 speck formation from THP-1 NLRP3(Δexon3)-
mNG cells. Close-up images at 0, 16, 26 and 58 min after nigericin addition are
shown. **C** Time of MTOC-associated and -distal speck formation from **A** and **B**
(one datapoint per speck). Two-sided Kruskal-Wallis test with Dunn's multiple
comparisons test. **D** Cell death in time measured by SYTOX orange uptake from

**A** and **B**. **E** Percentage (%) of ASC speck formation from the indicated THP-1 cell
lines upon the treatments shown. Each datapoint represents the average of 3 tiles
(3 × 3 FOV) (200 cells approx.). One-way ANOVA. $N = 3$ or 4 independent experi-
ments depending on the condition (see individual data points). **F** IL-1β release as
treated in E. Two-sided Student's t-test. **G** TNF-α release from the indicated THP-1
cell lines LPS-primed or not. $N = 5$ independent experiments. Two-sided Student's
t-test. **H** IL-18 release from WT THP-1 cells and endogenous CRISPR-Cas9 NLRP3
Δexon3 clones (D1–D5) after LPS priming (3 h or 18 h) followed by nigericin sti-
mulation (1 h or 4 h), as indicated. $N = 3$ independent experiments. All data were
obtained from 3 independent experiments and represented by mean ± SD, unless
otherwise indicated.

to non-decameric and smaller NLRP3 species. To confirm these results
for endogenous NLRP3, we generated THP-1 single clones cells in which
*NLRP3* exon 3 (together with the adjacent introns) was deleted by
CRISPR-Cas9 editing[36] (Supplementary Fig. 4), and validated at the
genomic (Supplementary Fig. 4A, B) and transcript level by RNA-seq:
here Sashimi plots across the exon2–exon4 region showed the expec-
ted exon2→exon4 junction reads in Δexon3 clones, consistent with loss
of exon 3 (Supplementary Fig. 4C). Analysis of the broader tran-
scriptomic context in PCA analysis nevertheless confirmed a high
degree of overall similarity to WT cells, with D3 and D4 emerging as the
most WT-like clones (Supplementary Fig. 4D). Of note, immunoblot
analysis of NLRP3 protein expression confirmed the expected subtle
size shift (Supplementary Fig. 4E). Functionally, we then measured IL-18
release—a cytokine also processed by the inflammasome but whose
expression is independent of LPS priming[37,38]—in these endogenously
NLRP3 Δexon3 clones and intentionally extended the assay window (3 h
*vs* prolonged 18 h LPS priming, and 1 h *vs* 4 h nigericin stimulation) to
capture potentially slower kinetics. We extended the stimulation win-
dow because the MTOC-distal activation we observed in the over-
expression system was delayed, and we expected this delay to be at
least as pronounced (if not more) when Δexon3 is expressed endo-
genously. In addition, we also extended LPS priming based on prior data
showing that a typically dysfunctional NLRP3 Δexon5 variant regains
activity after prolonged priming[39]. In line with that idea, Δexon3 clones
showed little to no IL-18 release at 1 h under 3 h LPS-priming, but robust
IL-18 became detectable upon extended stimulation and/or prolonged
priming, with the most WT-like clones (D3/D4) showing the strongest
recovery (Fig. 4H). This confirms further the functionality of the Δexon3
NLRP3 in an endogenous setting. To link this back to our spatial model,
we quantified ASC-speck proximity to pericentrin using fluorescence
microscopy (Supplementary Fig. 4F, G). This confirmed that in the
unedited WT THP-1 cells, ASC specks were predominantly MTOC-
associated at 1 h and shifted toward a higher fraction of MTOC-distal by
4 h, the time point at which IL-18 release was comparable (Fig. 4H). At
this time point of equal functionality specks in Δexon3 THP-1 cells were
largely MTOC-distal at 4 h as expected (Supplementary Fig. 4F, G).
Taken together, these results support that Δexon3 NLRP3, when
expressed from its endogenous locus, can form functional inflamma-
somes endogenously, with a clear shift toward MTOC-distal assembly
and delayed kinetics compared to WT.

## Microtubule disruption fails to block MTOC-distal NLRP3 inflammasome activation

We further explored the mobility of both MTOC-associated and -distal
NLRP3 specks by single object tracking analysis (see Methods). We
observed that MTOC-distal specks clearly showed increased mobility
(Supplementary Fig. 5A), consistent with the absence of membranes
and microtubule binding. To corroborate this further, we investigated
the sensitivity of MTOC distal NLRP3(Δexon3)-mNG speck formation
to the microtubule depolymerizing agents, colchicine or nocodazole[21].

Indeed, inhibitor treatment of the cells did not affect cell death in
Δexon3 cells measured by SYTOX assay in a 96 well-plate
format, whereas both WT and WT NLRP3-mNG THP-1 cells were
clearly and significantly affected (Fig. 5A). Furthermore, similar results
were obtained upon trafficking disruption via specific inhibition of
dynein (using ciliobrevin D) or HDAC6 (using ricolinostat) (Supple-
mentary Fig. 5B). Live cell imaging experiments in the presence of
nocodazole clearly showed the formation of MTOC-distal WT NLRP3-
mNG specks despite a completely disrupted microtubule system
(Fig. 5B and Supplementary Movie 4) that occurred at a similar time
compared to the NLRP3(Δexon3)-mNG speck formation (Fig. 5B
quantified in Fig. 5C). These data further evidence the existence of a
functional NLRP3 inflammasome acting independently of the micro-
tubular system.

## NLRP3 activation by imiquimod favors the TGN/MTOC-asso-ciated pathway

A question arising from previous results was whether different stimuli
might favor one pathway over the other. Therefore, using LPS-primed
THP-1 cells stably expressing WT- or Δexon3 NLRP3-mNG, we explored
whether the MTOC-associated and distal NLRP3 activation pathway
previously described for nigericin could be also extrapolated to other
K⁺-dependent stimuli. The introduction of other NLRP3 K⁺-dependent
activation stimuli, such as the monosodium urate (MSU) crystals[40] and
the *Staphylococcus aureus* leukocidin A/B (LukAB) toxin[41] also induced
IL-1β release and cell death in WT and NLRP3(Δexon3) that were effi-
ciently blocked in the presence of high extracellular KCl (Supple-
mentary Fig. 6A, B). Nevertheless, although high extracellular KCl
efficiently blocked MSU-induced IL-1β release, it did not reduce the
associated low level of cell death (which remained below 10%) (Sup-
plementary Fig. 6A, B). Thus, other K⁺-dependent stimuli were also able
to activate both MTOC-associated and distal NLRP3 pathways.

Interestingly, THP-1 cells expressing NLRP3(Δexon3)-mNG treated
with imiquimod, an NLRP3 activator considered to be K⁺-
independent[9], presented dampened IL-1β release and cell death
(Fig. 6A, B). In line with other studies in human cells[42], the minor
residual IL-1β release and cell death was completely abolished in the
presence of high extracellular KCl concentration, indicating that imi-
quimod acted as a primarily, but not exclusively, K⁺-independent sti-
mulus, and that the residual IL-1β in this system was still related to K⁺
efflux (Fig. 6A). Both NLRP3 WT-mNG and NLRP3(Δexon3)-mNG
responded robustly to nigericin stimulation, although L-1β release was
higher in the WT (Fig. 6A). The reduced IL-1β secretion observed in
NLRP3(Δexon3)-mNG could be attributed to lower LPS priming effi-
ciency (Fig. 6C). To control this, we measured IL-18 release instead and
following nigericin stimulation, IL-18 levels were comparable between
the two cell lines (Fig. 6D). In contrast, imiquimod stimulation led to a
similarly reduced IL-18 secretion in NLRP3(Δexon3)-expressing THP-1
cells, supporting a specific impairment of inflammasome activation in
response to this stimulus (Fig. 6D). Furthermore, THP-1 NLRP3 WT- or

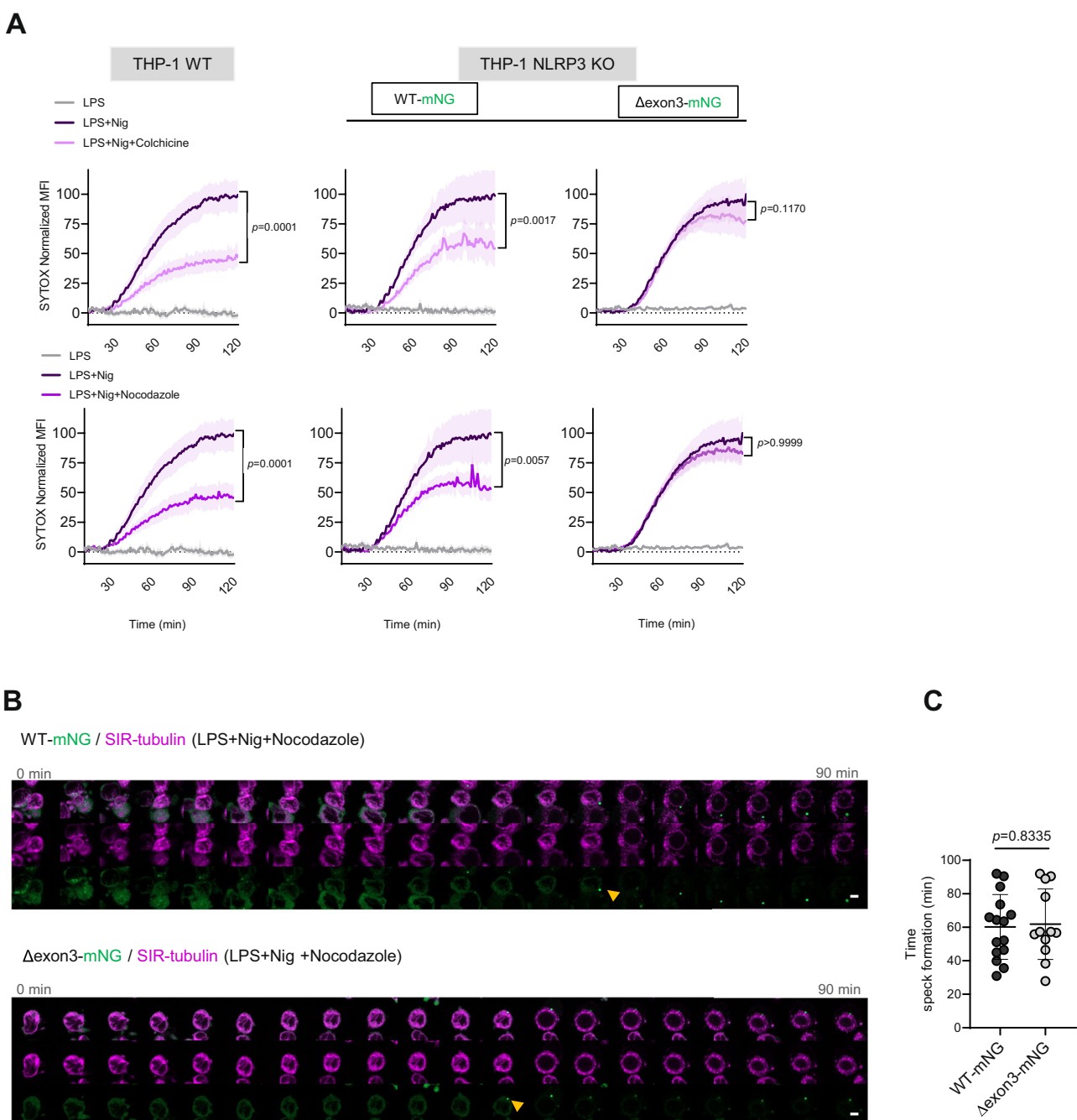

**Fig. 5 | MTOC distal NLRP3 activation is not affected by microtubule disruption. A** Kinetics of cell death from the indicated THP-1 cell lines upon the treatments shown. SYTOX orange uptake is represented as the MFI ± SEM normalized to the signal of the first 5 min of each condition. Two-sided Kruskal-Wallis test with Dunn's multiple comparisons test. **B** Representative live cell imaging timeline of the NLRP3 specks formation from LPS-primed and nigericin-stimulated THP-1 WT and Δexon3 NLRP3-mNG cells (green) including the microtubule disrupting agent nocodazole. Microtubules were stained with the specific probe SIR-tubulin (magenta). Shown are 20 frames from a time-lapse sequence, spaced 4.6 min apart. Arrows denote first occurrence of fully formed NLRP3 speck. Scale bar 5 μm. **C** Quantification of MTOC-associated and -distal NLRP3 specks. Each dot represents one speck from 2 independent experiments. Two-sided Student's t-test. All data were obtained from 3 independent experiments and represented by mean ± SD, unless otherwise indicated.

Δexon3- expressing cells sorted to equal mNG fluorophore levels to achieve comparable NLRP3 expression levels and similar TNF-α release (Supplementary Fig. 6C, D) further validated previous findings. NLRP3 medium expression levels for both cell lines released similar amounts of IL-1β in response to nigericin stimulation, but not following imiquimod treatment (Supplementary Fig. 6E). Notably, prolonged and stronger imiquimod stimulation eventually led to comparable IL-1β release in both cell lines (Supplementary Fig. 6E). Whereas nigericin activation led to pro-caspase-1, GSDMD, and pro-IL-1β cleavage to a

similar extent in WT- and Δexon3 NLRP3-mNG (Fig. 6C), caspase-1 and IL-1β cleavage and ASC speck formation (and to a lesser extent GSDMD cleavage) were strongly reduced in NLRP3(Δexon3) upon imiquimod stimulation (Fig. 6C, E and Supplementary Fig. 6F). These results reveal that the MTOC-distal NLRP3 pathway was not efficiently targeted by imiquimod. This observation was in line with microscopy experiments showing that, unlike nigericin (*cf.* Figure 3), WT NLRP3-mNG formed predominantly TGN-associated specks upon imiquimod stimulation (Fig. 6F, quantified in Fig. 6G) that were further MTOC-associated

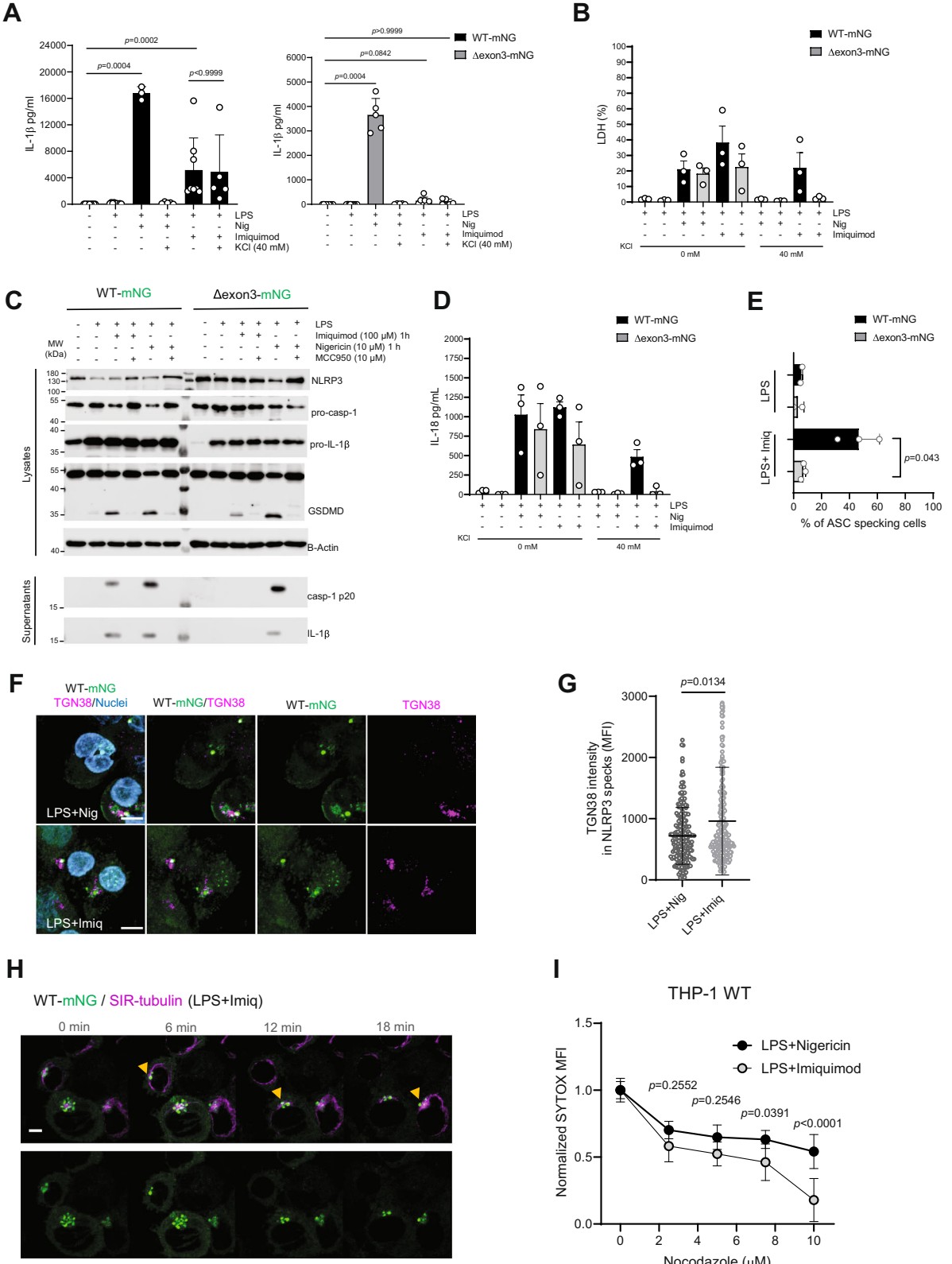

(Fig. 6H). Consistently, imiquimod-induced cell death was significantly more sensitive to the microtubule-depolymerizing drug nocodazole than nigericin-induced cell death in THP-1 WT cells (Fig. 6I). These results suggest that imiquimod stimulation favors the described canonical NLRP3 decamer/TGN/MTOC-associated pathway, while nigericin efficiently activated both the TGN/MTOC-associated and distal pathways.

## Human iPSC-microglia and primary macrophages activate both NLRP3 pathways, while neutrophils rely on the MTOC-distal pathway

So far, our data show that in macrophages two parallel NLRP3 pathways exist that are typically co-activated by strong $K^+$ stimuli such as nigericin. To explore whether other immune cells with functional inflammasomes also employ both NLRP3 pathways and hence

**Fig. 6 | K⁺ efflux agonists activate the MTOC-distal NLRP3 pathway whereas imiquimod favors the MTOC-associated NLRP3 activation. A, B** Reconstituted THP-1 cell lines IL-1β and LDH release after LPS-priming and 1 h stimulation with nigericin or imiquimod in low and high K⁺ conditions as indicated in the graphs. Two-sided Kruskal-Wallis test with Dunn's multiple comparisons test. *N* = 3 or 8 independent experiments depending on the conditions (see individual data points). **C** Immunoblots from cells treated as indicated. **D** IL-18 release in reconstituted THP-1 cell lines treated as in **A**. **E** Percentage (%) of ASC speck formation from reconstituted LPS-primed THP-1 cells in the presence or absence of imiquimod (microscopy images shown in supplementary Fig. 6F). Two-sided Student's t-test. **F, G** Representative immunofluorescence micrographs of the THP-1 WT-mNG cells LPS-primed and nigericin- or imiquimod-stimulated. TGN marked with the TGN38 antibody (magenta) and nuclei with Hoechst 33342 (blue). Scale bar 10 μm (**F**). Quantification of TGN38 MFI in NLRP3 specks. Each dot represents one speck from 3 combined independent experiments (**G**). Two-sided Mann-Whitney U test. **H** Representative live cell imaging of LPS-primed THP-1 NLRP3 WT-mNG upon imiquimod stimulation. Microtubules are marked with the specific probe SIR-tubulin (magenta). **I** Cell death measured by SYTOX orange uptake in THP-1 cells treated with nigericin or imiquimod in the presence of different concentrations of nocodazole. Two-way ANOVA. All data were obtained from 3 independent experiments and represented by mean ± SD, unless otherwise indicated.

demonstrate its physiological relevance, we next investigated induced-Pluripotent Stem Cells (iPSC) KOLF2.1J-derived human microglial cells[43]. Upon nigericin stimulation and similar to THP-1 macrophages-like cells, these showed a partial resistance to colchicine or nocodazole in terms of IL-1β release and were only poorly responsive to imiquimod treatment (Fig. 7A), also suggesting both MTOC-associated and distal routes acting in parallel in these cells. Next, we analyzed primary human monocyte-derived macrophages (MDMs) and human polymorphonuclear neutrophils (PMNs) obtained from the same blood donor[44]. In line with previous data, nigericin treatment induced robust IL-1β release for both primary cells (Fig. 7B). However, similar to iPSC-microglia, primary MDMs were only partially sensitive to microtubular disruption (Fig. 7B). Moreover, primary PMNs from the same donors were fully resistant (Fig. 7B), despite efficient disruption of their microtubular network as confirmed by confocal microscopy (Supplementary Fig. 7A). Further analysis of IL-1β release and cell death in primary PMNs confirmed their insensitivity to microtubule disruption (Fig. 7C). Consistently, in PMNs, ASC specks were predominantly localized distally from the core MTOC/centrosome, as indicated by pericentrin staining (Fig. 7D, quantified in Fig. 7E), whereas in primary MDMs, ASC specks were observed both near the Golgi and at distal sites (Fig. 7F). This provides evidence that in neutrophils, the most abundant leukocyte population in humans, NLRP3 activation follows predominantly a microtubule-independent pathway. Collectively, our data suggest a model of two distinct NLRP3 activation pathways (Fig. 8, Supplementary Table 1), with MTOC-distal signaling dominating in some human immune cell types.

## Discussion

Over the years, NLRP3 activation has emerged to involve a staggering number of factors, including protein conformation, modification, subcellular localization, and trafficking. Our goal was to elucidate the interplay between different oligomeric states of NLRP3 and the widely cited model of a spatiotemporal maturation of the NLRP3 inflammasome, initiating at the TGN and terminating at the MTOC[16,20,21]. Within and beyond this transition, different and controversial locations of NLRP3 during activation have been shown (Supplementary Table 2). Our results establish new insights regarding the links between structural arrangements, localization and cell biology underpinning NLRP3 inflammasome activation. Whilst we confirm that lipid-binding decameric human NLRP3 cages can follow the already described canonical TGN/MTOC-associated NLRP3 activation pathway[12,16], we demonstrate that other existing smaller NLRP3 species can engage a separate, TGN/MTOC-distal NLRP3 activation pathway. For WT NLRP3 both pathways seem to co-exist in macrophages and our work thus reconciles previously puzzling or ambiguous findings into a unified mechanism (Fig. 8, Supplementary Table 1). For instance, both microtubule and MTOC-distal specks have been observed in previous reports[21] but they have never been considered truly separate pathways or correlated with the oligomeric state of NLRP3. In agreement with our findings, Liu Y et al., and Bai S et al., also failed to locate endogenous ASC specks to the MTOC[45,46]. Furthermore, inflammasome activation requiring a

functional microtubule network has also been shown to depend on the strength of the stimuli (nigericin)[47].

Our detection of smaller ASC-positive puncta surrounding a central speck may initially seem to diverge from previous cryo-ET and super-resolution studies that reported ASC as a dense filamentous lattice[45,48]. However, this difference likely reflects variations in resolution, imaging context, and subcellular localization. Our live-cell confocal microscopy (~200 nm resolution) cannot resolve individual ASC filaments (~10–15 nm) and instead visualizes denser bundles as discrete foci. Time-lapse imaging shows that these smaller puncta are transient and rapidly merge into a central speck, suggesting they are early assembly intermediates. In contrast, super resolution microscopy studies examined fixed, cytoplasmic (MTOC-distal) specks and did not capture the MTOC-proximal assemblies. We consistently observed these "satellite" puncta in the pericentrosomal region, which may reflect a distinct architectural organization. Thus, the apparent differences are likely due to both temporal and spatial heterogeneity in speck formation that are not straightforward to harmonize. Nevertheless, our findings then add complementary insight into the dynamic nature of inflammasome assembly.

Regarding the NLRP3 structure, deletion of only the 40 aa linker sequence encoded by exon 3 led to smaller human NLRP3 species highly-enriched in monomers and dimers. That omission of entire exons encoding for LRR domains may also yield functional NLRP3 constructs, supports the notion that functional entities in NLRP3 are organized by exons[18,26]. Although omission of exon 3 in human cells as a result of alternative splicing was not observed by us and others (for analyses of RNAseq data, see Methods and Hoss et al.,[26]), human NLRP9, NLRP10 and NLRP11, and, more importantly, chicken and zebrafish Nlrp3, all lack this exon but nevertheless form functional inflammasomes[27–31]. Together with the demonstration that the NLRP3(Δexon3) formed functional inflammasomes, this variant proved a valuable nature-inspired tool for probing NLRP3 function and yielded the delineation of pathways that are naturally occurring in human primary cells and possibly common in non-mammals. From a structural point of view, the emergence of higher order assembly states of NLRP3 solely by the presence of an artificial short flexible linker (GSGAGG) suggests that the exon 3 linker is normally responsible for conferring the flexibility that allows for the PYD domains to be buried in the oligomer[16,17], and/or to orient the polybasic regions at the same interface for NLRP3 TGN/MTOC interactions. In line, our lipid affinity assay showed binding of NLRP3 to specific lipid species that were consistent with previous findings reported by Andreeva et al. and Chen et al.[16,20]

NLRP3 was proposed to shift from monomers to oligomers in a stable equilibrium[16], albeit it is unclear how, under cellular conditions, different NLRP3 species are sorted between cytosol and TGN. Our study demonstrated the existence of two parallel NLRP3 activation pathways: a TGN/MTOC-associated pathway modulated by the oligomeric status of NLRP3, and a TGN/MTOC-distal one that appears governed by smaller NLRP3 species. Consistently, the Δexon3 NLRP3 construct displays fully functional inflammasome properties upon nigericin treatment, independently of TGN/MTOC binding. In addition,

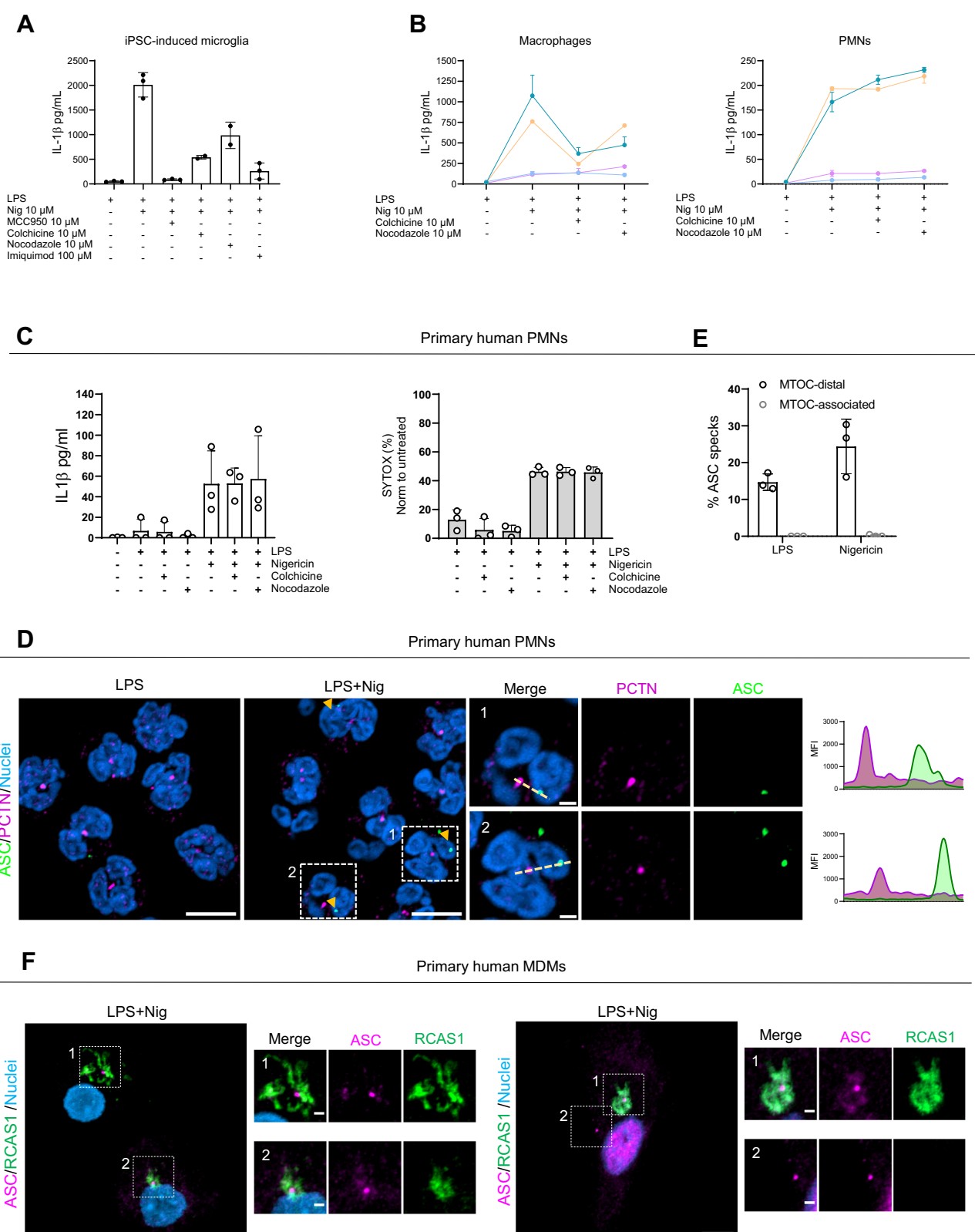

neither endogenous skipping of exon 3 nor endogenous CRISPR-Cas9-mediated exon 3 editing blocked nigericin NLRP3 inflammasome activation. A plausible explanation is that we shifted the NLRP3 equilibrium from decamers to smaller species, consequently favoring the TGN/MTOC-distal pathway.

The two-pathway model provides some room to accommodate the role of NEK7 in NLRP3 activation. NEK7 was observed to be part of

the active human NLRP3 disc with 1:1 heterodimeric NLRP3-NEK7 complexes[12]. Moreover, it was unable to bind the inactive NLRP3 oligomer, as the interaction sites are buried in the NLRP3 structure[16,17]. NEK7 KO cells respond poorly to imiquimod, as previously shown by Groβ et al.[9] and recently by Schmacke et al.,[42] in the same way that NLRP3(Δexon3) does. By analogy, we speculate that non-oligomeric, not TGN- or MTOC-associated NLRP3 may act in an NEK7-independent

**Fig. 7 | K⁺ efflux activates both MTOC-associated and MTOC-non associated NLRP3 pathways in macrophages, whereas the MTOC-non associated pathway predominates in neutrophils. A** IL-1β release from iPSC-derived microglia upon the treatments shown. Data are presented as mean ± SD from 3 independent experiments, for colchicine and nocodazole, data are derived from 2-3 independent experiments (see individual data points). **B** IL-1β release from primary human macrophages and neutrophils derived from the same blood donors in response to nigericin and treated in the presence or not of microtubule disruptors. Each color represents an individual donor, $N = 4$ independent experiments. **C** IL-1β release and cell death measured by SYTOX orange uptake from primary human neutrophils treated as in **B**. **D** Representative immunofluorescence images of LPS-primed and nigericin stimulated human primary neutrophils with endogenous ASC stained using a specific ASC antibody (green) and MTOC region with pericentrin antibody

(magenta). Nuclei were stained using Hoechst 33342 (blue). Scale bar 5 μm, 1 μm for close-ups. Representative line graphs depicting changes in NLRP3 and MTOC mean fluorescence intensity (MFI) are shown. **E** Percentage (%) of MTOC-distal and -associated ASC specks of the microscopy data examples shown in **D**. Data combined from 3 independent experiments. Each data point represents the mean of each experiment, averaged from the five randomly selected FOV per condition. **F** Representative immunofluorescence images of LPS-primed and nigericin stimulated human primary macrophages with endogenous ASC stained using a specific ASC antibody (magenta) and Golgi membranes with RCAS antibody (green). Nuclei were stained using Hoechst 33342 (blue). Scale bar 5 μm, zoom in 1 μm. All data were obtained from 3 independent experiments and represented by mean ± SD, unless otherwise indicated.

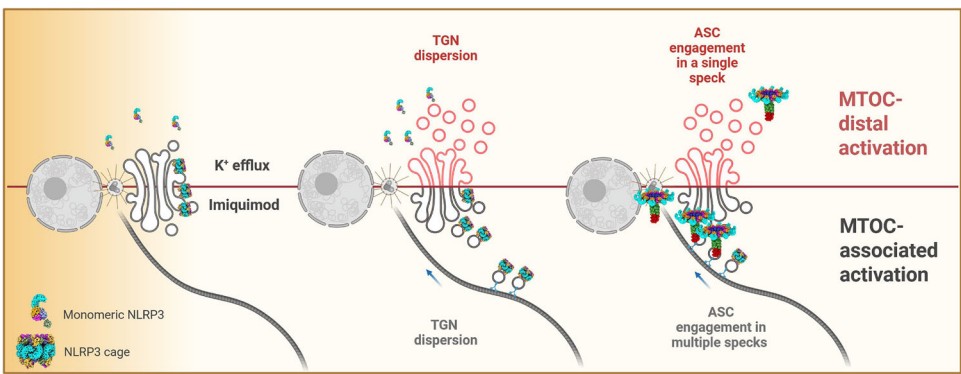

**Fig. 8 | Proposed mechanism of TGN/MTOC-associated and -distal NLRP3 inflammasome activation.** The NLRP3 double-ring cage resides at the TGN, and upon K⁺-dependent or imiquimod stimulation, a TGN dispersion is triggered. This allows NLRP3 to travel to vesicular structures, which in turn culminate in MTOC nucleation, resulting in a TGN/MTOC-associated NLRP3 inflammasome activation.

However, smaller NLRP3 species (e.g. monomers, dimers) with reduced ability to bind membranes respond to K⁺-dependent stimulation. Here, NLRP3 oligomerizes independently of membrane association and induces the distal TGN/MTOC NLRP3 activation. Created in BioRender. Weber, A. (2026) https://BioRender.com/fwomx3w.

fashion, which would be consistent with the MTOC localization of NEK7. However, further work outside the scope of this present work will be necessary to fully confirm this[49] and also explore a potential influence of IKKβ[42].

Importantly, the disruption of the microtubule system, which would trap oligomers of NLRP3 on membranes and prevent them from reaching the MTOC[21] is a feature critical for the TGN/MTOC-associated pathway. Yet, it failed to abolish the NLRP3 inflammasome activation associated to the K⁺-dependent stimulus nigericin in human macrophage-like cells and primary PMN, further proving the existence of a TGN/MTOC-distal pathway in physiologically relevant settings. Interestingly, our results clearly showed NLRP3 inflammasome specks that can sometimes be MTOC-associated or not, unifying different reports on NLRP3 membrane dependency (see also Supplementary Table 2). Additionally, our data explain why an LRR-deleted NLRP3 protein, which cannot form the double-ring cage and hence should exist in monomers, may form an active inflammasome when overexpressed[18], namely by the cytosolic fraction of this protein responding via the MTOC-distal pathway. Strikingly, NLRP3 activation upon the K⁺-independent stimulus imiquimod showed an NLRP3 preferentially localized at the TGN, and the addition of the microtubule disrupting agent nocodazole significantly dampened inflammasome activation. Accordingly, the NLRP3(Δexon3) construct—existing mostly in monomers and dimers and with reduced ability to bind TGN and to form MTOC-associated specks—was less responsive to imiquimod stimulation, although prolonged stimulation and high doses of this compound also induced IL-1β release in this cell line. These results suggest that NLRP3 binding to membranes and interacting with the MTOC system is relevant for imiquimod responsiveness (Fig. 8).

Our results showed reduced IL-1β release from the NLRP3(Δexon 3) cells compared to WT, likely due to less effective priming. However,

levels of IL-18 were similar between both cell types (Fig. 6A-D) and after extended priming in THP-1 cells carrying endogenous CRISPR-Cas9-mediated exon 3 editing (Fig. 4H), which suggests comparable inflammasome activation. Intriguingly, when high extracellular K⁺ was applied in the presence of imiquimod, IL-18 and LDH signals were abolished in NLRP3 exon 3–deficient cells but remained largely intact in WT cells. We interpret this to mean that imiquimod still induces a modest K⁺ efflux; the cage-independent pathway (NLRP3 (Δexon3)) is sensitive to even a small drop in K⁺, whereas the cage-based pathway in WT cells appears comparatively K⁺-independent.

Regarding the physiological role of NLRP3 cages residing in the Golgi area, our live cell imaging experiments suggest that an increased concentration of NLRP3 located in this defined area triggers a more efficient and rapid inflammasome response which manifests itself in earlier ASC specks formation, IL-1β or IL-18 release, and pyroptotic cell death. This may be a mechanism to ensure rapid NLRP3 activation under suboptimal stimulation conditions, facilitated by the ability of NLRP3 to readily cluster at membranes[50]. Differences observed in slope between Figs. 5A and 4D stem from different experimental approaches, monitoring cell death in bulk populations *vs* per-cell timing. Consequently, subtle variations in individual cell death timing are averaged in the population readout, masking the delayed kinetics seen at the single-cell level. Conversely, in primary human PMNs, which are highly reactive cells, proper MTOC organization appears nonessential for NLRP3 activation and ASC specks were exclusively MTOC-distal, highlighting the physiological relevance of the MTOC-non-associated pathway in this most abundant human leukocyte population. This and another previous study[51] highlight that PMNs respond very differently in terms of NLRP3 regulation[52], although their dependency on factors such as ASC and caspase-1 remains consistent.

Our results also align well with observations regarding NLRP3-mediated disease in CAPS patients. Cells in which NLRP3 CAPS mutants were expressed ectopically present a pattern of NLRP3 foci distributed around the cytosol[10,20,22,53], suggesting that constitutively active NLRP3 disease mutants bypass TGN dependence[20,53]. Moreover, an LRR-deleted NLRP3 variant unable to organize in double-ring cages and further carrying the R260W or T348M CAPS mutations still supported constitutive activity[18]. Moreover, unlike gout flares caused by WT NLRP3[54], CAPS symptoms showed only a partial or no response[55] to colchicine, indicating a poor efficacy of microtubule disruption. This clinical observation squares well with the dominance of the MTOC-distal activation pathway in CAPS macrophages, but in patients may also relate to the insensitivity of primary PMN to microtubule disruption in vitro. Translationally, our work suggests that elucidating to which extent TGN/MTOC-distal pathways control NLRP3 activation in both genetic (e.g. CAPS) and non-genetic NLRP3-related pathologies might be necessary and useful for developing new and efficient strategies to block inflammation in each setting.

## Limitations of the study

In our study, we provide evidence for the existence of the TGN/MTOC-associated and -distal NLRP3 inflammasome activation pathways. However, we primarily focused our work on the in vitro situation and the experimental results have been obtained by mainly using human THP-1 macrophage-like cells as a model because they are amenable to genetic manipulation and tagging for highly-resolved live microscopy. We opted to use both pools of THP-1 cell lines and single clones because we believe the combined results obtained are more biologically relevant. Nevertheless, further expansion of the data presented for human primary cells into additional NLRP3-competent human cell types[3] beyond what is presented here would be valuable.

## METHODS

### Chemicals and reagents

Phorbol 12-myristate 13-acetate (PMA, tlr-pma), LPS ultrapure from *Escherichia coli* K12 (LPS-EK, tlr-peklps), nigericin sodium salt (tlr-nig), imiquimod (tlrl-imq), and monosodium urate crystals (MSU, tlr-msu) were obtained from InvivoGen. KCl (6781.1, Roth). MCC950 was purchased from Cayman Chemicals (Cay17510-1), LukAB was a kind gift from C. Wolf (Medical Microbiology, University of Tübingen)[41]. Colchicine (C9754), Nocodazole (M1404), and Ciliobrevin D (250401) were acquired from Sigma Aldrich, Ricolinostat (S8001) from SelleckChem.

### Cell lines

HeLa and HEK293T cells were cultured in complete DMEM (4.5 g L$^{-1}$ glucose, (Sigma Aldrich D5796) containing 10 % heat-inactivated fetal bovine serum (FBS) (Th Geyer, 11682258), 2 mM L-Glutamine (Gibco, 25030081), 100 U/mL penicillin/100 μg/mL streptomycin (Gibco, 15140122)) and sub-cultured every 2–3 days.

Regular THP-1 cells were maintained in complete RPMI 1640 (Sigma Aldrich R8758) containing 10% FBS, 25 mM HEPES, 2 mM L-Glutamine, 100 U/mL penicillin/100 μg/mL streptomycin, and sub-cultured every 3 days. THP-1 Null 2 (thp-nullz, InvivoGen) and THP-1 Null 2 *NLRP3* KO (thp-konlrp3z, InvivoGen) were maintained in complete RPMI 1640 (containing 10 % heat-inactivated fetal bovine serum, 25 mM HEPES, 2 mM L-Glutamine, 100 U/mL penicillin,100 μg/mL streptomycin, 100 μg/mL Normocin (ant-nr-05, Invivogen), and 100 μg/mL of Zeocin (ant-zn-05, Invivogen) and sub-cultured every 3 days. All cell lines were cultured at 37 °C in 5 % CO$_2$ and regularly checked by PCR to avoid *Mycoplasma* contamination.

### Study participants and human blood acquisition

All healthy donors included in this study provided written informed consent prior participation. Approval for use of biomaterials was obtained from the Ethics Committee of the Medical Faculty of Tübingen in accordance with the principles laid down in the Declaration of Helsinki as well as applicable laws and regulations. The research protocol was also reviewed by the Ethics Committee of the Hospital General Universitario Santa Lucía, Cartagena, Spain, on March 25, 2025 (protocol code: INFLA-COL). The handling of biological samples used in this study was conducted in accordance with the ethical principles governing research involving biological samples, as established by Spanish Law 14/2007 on July 3, on Biomedical Research.

### Plasmids and cloning procedures

The different ENTRY plasmids of human NLRP3 were generated by PCR from an NLRP3 template (Uniprot #Q96P20) and cloned into pENTR20-mNG (kind gift from Kay Oliver Schink, University of Oslo). The NLRP3 sequences and the mNG fluorophore are separated by a long flexible linker of 8 amino acids (TSGSGSGSG). Sequencing was performed to confirm correct amplification and the absence of unwanted mutations. pTGN46-mScarlet2 was a kind gift from Michael Schindler (Molecular Virology, University of Tübingen) in which mScarlet2 was replaced with mCherry. All other plasmids were generated using standard molecular cloning techniques. Detailed cloning procedures are available from the authors.

### Cell treatments and transfection

HeLa cells were transfected with TGN46 plasmid, and human NLRP3 WT and Δexon3 using Lipofectamine 2000 (11668027, Invitrogen) according to the manufactures' instructions. After 24 h of transfection the experiment was performed.

For assessing NLRP3 activation, all the THP-1 cell lines used in the present work were first differentiated with 100 ng/mL of PMA overnight, rested for 2 days, then primed with 10 ng/mL of LPS-EK and stimulated or not with 10 μM nigericin, 100 μM imiquimod, 250 μg/mL MSU or 5 μM LukAB at the indicated times. For NLRP3 inflammasome inhibition 10 μM MCC950 was added into the cultures 30 min before and during inflammasome activation. Where indicated, 40 mM KCl was added to the culture media in order to prevent potassium efflux during inflammasome activation.

For pharmacological inhibition, LPS-primed THP-1 cell lines were pretreated for 1 h with the indicated inhibitor. HDAC6 inhibitor Ricolinostat was used at 10 μM; dynein-dependent microtubule transport was inhibited using Ciliobrevin D at 10 μM; for microtubule polymerization disruption, nocodazole at 2.5–10 μM or colchicine at 10 μM was used. Then, cells were treated for inflammasome activation as indicated above.

### Generation of stable cell NLRP3-mNG reconstituted cell lines

Third-generation lentivirus was generated using procedures and plasmids as previously described in ref. 56. NLRP3 WT, Δexon3, and AMAA constructs, all C-terminally mNG-tagged and previously generated as Gateway ENTRY plasmids, were transferred by Gateway LR recombination (Gateway ™ LR Clonase ™ II Enzyme mix, 11791020, Thermo Fisher) into lentiviral destination vectors[57] (Gateway-enabled vectors derived from pCDH-EF1a-MCS-IRES-BLAST, (CD532A-2, SystemBiosciences). VSV-G pseudotyped lentiviral particles were packaged using a third-generation packaging system[58] (Addgene plasmids 12251, 12253, 12259).

For the generation of lentiviral particles, 0.5 μg pMD2.G containing pCMV-VSVG, 1 μg pMDLg/pRRE and 1 μg pRSV-REV plus 2 μg of the gene of interest expressing plasmid were transfected into HEK293T cells using Lipofectamine 2000 on a final volume of 1 mL and following the manufacturer's instructions. The day before transfection, 1×10$^6$ of freshly thawed HEK293T were seeded in a 6-well plate in complete DMEM. After 72 h, the supernatants were collected and filtered using 0.45 μm Millex-HV Syringe Filters and directly used for transduction or kept at −80 °C. HeLa cells were infected with a 25 % (v/v) of lentivirus-containing media and incubated for 3–5 days. To

improve the efficiency of transduction in THP-1 NLRP3 KO, both cells and 50 % of lentivirus-contained supernatant were centrifuged at 1000 x *g* at 30 °C for 1 h. Then, the THP-1 cells were resuspended up and down gently, and seeded and incubated for 3–5 days. To obtain stable expressing populations, HeLa and THP-1 cells were subjected to antibiotic selection using blasticidin (Gibco, A1139-03) at 5 μg/mL. The expression of the NLRP3 constructs is under the control of the human promoter EF-1α.

### Generation and flow cytometric sorting of THP-1 NLRP3 WT-mNG and Δexon3-mNG cells

THP-1 *NLRP3* KO cells stably expressing NLRP3 WT-mNG or Δexon3-mNG were flow-sorted at the Flow Cytometry Core Facility, University of Tübingen. Cells were prepared by centrifuging one T75 cm$^2$ flask per cell line, resuspending them in phenol red-free RPMI supplemented with 10 % FBS at $1 \times 10^7$ cells per 100 μL, and passing the suspension through a cell strainer immediately before sorting. Sorting was performed on a BD FACSAria III using FACSDiva v9.0.1 with a 100 μm nozzle, 20 psi sheath pressure, 30.0 frequency, and purity sort settings. An unstained THP-1 control was used to define background fluorescence and set the gates. Cells were sequentially gated based on forward and side scatter, followed by singlet discrimination using FSC-A and SSC-A parameters, and were then separated into mNG-low, mNG-medium and mNG-high fractions. The same strategy was applied to both NLRP3 WT-mNG and Δexon3-mNG reporter lines. Detailed gating and sorting strategies are shown in Supplementary Fig. 8.

### Generation of CRISPR-Cas9-edited Δexon 3 THP-1 single clones

To specifically remove exon 3, two guide RNAs targeting flanking regions were used: 5′-AGTGCACATAGTGTACAATT−3′ and 5′-TAGAT-TACCGTAAGAAGTAC−3′. 0,5 ×10⁶ of THP-1 cells were resuspended in 50 μL of working mixture (Lonza Cell line Nucleofector Kit V, according to manufacturer´s protocol) and Cas9:guideRNA RNP complexes, electroporation enhancer, and the ssDNA donor oligo were added (all from Integrated DNA Technologies (IDT) to a final volume of 64 μL and final concentrations of 5.7 μM, 4.0 μM and 4.6 μM, respectively, based on[36]. Prior to mixing the reagents, Cas9:RNP complexes were prepared by adding 2 μL of each hybridized crRNA:tracrRNA complex (50 μM) (1072533, IDT) with 6 μL of Cas9 Nuclease V3 (61 μM) (1081058, IDT) and incubating them for 15 min at 21 °C. Electroporation was performed in the Lonza Nucleofector 2b with the pre-set THP-1 (ATCC), high efficiency program. After electroporation, cells were immediately transferred to a 48-well plate with 500 μL pre-warmed complete RPMI-1640 supplemented with IDT HDR Enhancer V2 (10029790, IDT9 (1:588 diluted) and Y-27623 (Rock inhibitor) (100-1044, StemCell Technologies). After 12 h, medium was renewed with complete RPMI-1640. After cells started proliferating again, they were expanded, and single clones were generated by limiting dilution.

Screening PCR of single clones: The 96-well plate of single clones was split and a duplicate plate centrifuged for 5 min at 300 x *g* and the supernatant was carefully removed. 25 μL of DNAzol Direct (DN 131, Molecular Research Center) was added to each well and the plate was incubated for 30 min at room temperature. 1 μL of crude DNA lysate was used for the screening PCR reaction containing 3 primers: Exon 2 forward: 5′-GCAGACCATGTGGATCTAGC-3′; Exon 3 forward: 5′-GCACGTGTTTCGAATCCC-3′; Exon 4 reverse: 5′-CCTGTCTTCAATG-CACTGG-3′.

The reaction was carried out using the KOD One Blue Polymerase master mix (KMM-201NV, Toyobo) with a reaction volume of 30 μL. PCR reactions were screened for the absence of an Exon3-Exon4 amplicon and for the presence of an Exon2-Exon4 amplicon with the expected length. Selected clones were expanded and DNA was purified with the DNA Blood Isolation Kit (51104, Qiagen). The PCR was then repeated using approx. 80 ng of purified DNA per reaction and PCR products subjected to Sanger sequencing of the exon junction.

### Generation of iPSC-derived microglia

KOLF2.1J iPSCs[43] were maintained in mTeSR Plus media (100-0276, Stem Cell Technology) in 6-well plates coated with Geltrex™ LDEV-Free Reduced Growth Factor Basement Membrane Matrix (A1413202, ThermoFisher) diluted in Advanced DMEM/F-12 (12634010, Gibco). For Embryoid body (EB) generation, iPSC colonies were dissociated and 2.5 million cells/well were plated into low adherence AggreWell™ 800 plate (34850, Stem Cell Technology) using mTeSR Plus media containing 50 ng/mL human bone morphogenetic protein 4 (BMP-4) (130-111-167, Miltenyi Biotec), 50 ng/mL human vascular endothelial growth factor (VEGF) (130-109-396, Miltenyi Biotec), 20 ng/mL human stem cell factor (SCF) (PT-25.0010, R&D), and 10 μM ROCK inhibitor (100-1044, Stem Cell Technology). Plated cells were centrifuged at 580 x *g* for 3 min without brake, and medium change was performed each day for 4 days. For myeloid maturation, EBs were gently swirled up from the AggreWell™800 plate and transferred onto a 40 μM strainer, washed once with wash Buffer (DMEM-F12 + 0.1% BSA), and distributed evenly into 6-well plates (Greiner). EBs were maintained in X-VIVO 15 media (02-053Q, Lonza) supplemented with 1 % penicillin/streptomy-cin (Gibco), 1X GlutaMAX (35050061, Gibco), 55 μM 2-mercaptoethanol (21985023, Gibco), 100 ng/mL human macrophage colony stimulating factor (M-CSF) (130-096-492, Miltenyi Biotec), and 25 ng/mL human interleukin- 3 (IL-3) (130-095-070, Miltenyi Biotec). EBs were fed weekly. Microglia progenitors were collected during medium change and ultimately differentiated into iPSC-derived microglia by culture for 2 weeks in 50 % DMEM/F-12 and 50 % Neuro-basal Medium (10888022, Gibco), supplemented with 1X GlutaMAX, 55 μM 2-mercaptoethanol, 1X B27 Supplement with Vitamin A (Gibco), 100 ng/mL IL-34 (577904, Biolegend), and 20 ng/mL M-CSF (130-096-492, Miltenyi Biotec) at a density of 100,000 cells/mL. Medium change was performed 3 times per week.

### hMDMs isolation, macrophage differentiation, and inflamma-some activation

Peripheral blood mononuclear cells (PBMCs) were isolated from whole blood by density centrifugation with Ficoll 1.077 g/mL (GE17-1440-02, Sigma Aldrich). Briefly, EDTA-anticoagulated blood samples were diluted 1:1 with PBS (D8537, Sigma Aldrich) and subsequently added onto Ficoll and centrifuge at 500 x *g* for 25 min at room temperature without brake. Afterwards, the PBMC layer was carefully collected and transferred into another tube and washed twice with PBS. In order to isolate the monocytes from the PBMCs portion, $30 \times 10^6$ cells were seeded in 10 cm culture-treated dishes (Greiner bio-one) in 10 mL of Monocyte Attachment Medium (C-28051, Promocell) to promote monocyte adhesion. After 1.5 h of incubation at 37 °C and 5 % CO$_2$, the supernatant was aspirated from the dishes and washed 3 times with PBS to eliminate non-adherent cells. Then, complete RPMI medium (10 % FBS, 1% L-glutamine, 1 % P/S) was added, and monocytes were differentiated into macrophages with 50 ng/mL of hGM-CSF (NDC 0024-5843-05, Sargramostin, Leukine/Sanofi). The cells were incubated for 7 days, refreshing the culture at day 4 with 5 mL of complete RPMI medium and 50 ng/mL of hGM-CSF. Afterwards, the macrophages were washed twice with PBS and subsequently detached adding 10 mL of Macrophages Detachment Medium (C-41330, Promocell). The dishes were incubated for 40 min at 4 °C and 20 min at room temperature, and then, 10 mL of complete RPMI medium was added to finally detach the cells carefully pipetting up and down. The collected cells were washed twice complete RPMI medium, and the cell concentration determined.

The macrophages obtained were seeded in coverslips at $0.3 \times 10^6$ cells/well in 500 μL or in 96-well plates at $0.1 \times 10^6$ cells/well in 200 μL of complete RPMI medium and incubated for 24 h at 37 °C and 5 % CO$_2$ to facilitate their adhesion. For NLRP3 inflammasome activation, hMDMs were primed with 10 ng/mL of LPS-EK for 3 h and stimulated or not with 10 μM nigericin for 1 h. For microtubule polymerization

disruption, LPS-primed cells were pretreated for 30 min with colchicine or nocodazole at 10 μM. Then, cells were treated with nigericin as indicated above.

## Primary human neutrophil isolation and stimulation

Neutrophils of healthy human donors were isolated and their activation status analyzed as described in ref. 59. or using the MACSxpress whole blood neutrophil isolation kit (130-104-434, Miltenyi Biotec) following manufacturer instructions. For the first option, whole blood (EDTA-anticoagulated) was diluted in PBS, loaded on Ficoll and centrifuged for 25 min at $500 \times g$ at 21 °C without brake. The erythrocytegranulocyte pellet was taken and erythrocyte lysis (using 1X Ammonium-Chloride-Potassium (ACK) lysis buffer, ThermoFisher Scientific A1049201) was performed twice (for 20 and 10 min) at 4 °C on a roller shaker. The remaining cell pellet was carefully resuspended in complete culture medium RPMI and $1.6 \times 10^6$ cells/mL were seeded for 24-well plate and $0.4 \times 10^6$ cells/mL for 96-well plate. After seeding, the cells were rested (30 min) and primed with 10 ng/mL of LPS-EK for 3 h, then left incubated or not with nocodazole or colchicine, both at 10 μM for 30 min and finally stimulated or not with 10 μM nigericin for 2 h or for 1 hour in the comparative experiments using macrophages derived from the same blood donor. After PMN isolation and stimulation, the purity and activation status of neutrophils was determined by flow cytometry, using specific CD15, CD66b, CD14 and CD62L antibodies. 200 uL of the cell suspension was transferred into a 96-well plate (U-shape) and spun down for 5 min at $450 \times g$, 4 °C. FcR block was performed using pooled human serum diluted 1:10 in FACS buffer (PBS, 1 mM EDTA, 2 % FBS heat inactivated) for 15 min at 4 °C. After washing, the samples were stained for 20–30 min at 4 °C in the dark. Thereafter, fixation buffer (4 % PFA in PBS) was added to the cell pellets for 10 min at RT in the dark. After an additional washing step, the cell pellets were resuspended in 150 μL FACS buffer. Measurements were performed on a FACS Canto II from BD Bioscience, Diva software. Analysis was performed using FlowJo V10 analysis software

## Cloning, expression, and purification of human NLRP3

Full length human NLRP3 (3-1036, UniProt accession code Q96P20) WT, full length mouse Nlrp3 (1-1033, Uniprot accession code Q8R4B8) WT) or human mutants with the linker region (exon 3 sequence) deleted (Δ95-134) or deleted and reconstituted with a small, flexible linker (Δ95-134 + linker (GSGAGG)), codon-optimized for *Spodoptera frugiperda*, were cloned in an in-house modified pACE-Bac1 vector containing a N-terminal MBP-tag, followed by a Tobacco etch virus (TEV) protease cleavage site.

For recombinant protein expression of the different NLRP3 proteins, 500 mL of *Sf9* insect cells were infected with 3 % v/v viral stock of the second virus passage. Expression cultures were incubated with gentle shaking for 48 h at 27 °C, and were subsequently harvested by centrifugation at 1000 x *g* (JLA-8.1000 rotor, Beckman-Coulter) for 20 min. Cell pellets were washed with PBS and either subsequently used for protein purification or flash-frozen in liquid nitrogen for storage at −80 °C. For protein purification, cell pellets were solubilized in lysis buffer (50 mM HEPES pH 7.5, 150 mM NaCl, 0.5 mM TCEP, 10 mM MgCl$_2$, 1 mM ADP), supplemented with 1 mM phenylmethylsulfonyl-fluoride (PMSF), followed by sonication (10 sec on, 5 s off for 4 min at 40% intensity) on ice. Cell lysates were centrifuged at $75,000 \times g$ (JA-25.50 rotor, Beckman-Coulter) and 4 °C for 1 h and the supernatants were filtered with a 45 μm syringe filter, prior to application onto a lysis-buffer equilibrated 5 mL MBP-trap column (GE Healthcare) attached to an ÄKTA-Start FPLC system. The column was subsequently washed with 10 CVs of lysis buffer, followed by elution of the proteins with 5 CVs of lysis buffer supplemented with 15 mM maltose. Proteins were further purified by size exclusion chromatography (SEC) on a Superose 6 increase 10/300 GL column (GE Healthcare) equilibrated with lysis buffer. Elution fractions were analyzed by Coomassie- staining SDS-PAGE and negative stain electron microscopy (EM). Proteins used for lipid strip binding assays were purified as described above.

## Negative stain EM

For negative stain EM, 4 μL of the indicated SEC elution fractions were applied onto glow-discharged carbon-coated copper grids (200 nm mesh, PLANO). Samples were incubated on the grid for 1 min, before excess sample was blotted away with a filter paper. Grids were subsequently washed by dipping the sample side into three individual 20 μL drops of lysis buffer, alternated by a blotting step in between. Samples were negative stained by dipping the sample side into a 20 μL drop of 2 % uranyl acetate and incubation for 30 s, before excess stain was blotted away with a filter paper. Samples were air-dried and subsequently imaged with a Jeol JEM-2200FS transmission electron microscope operated at 200 kV.

## SEC-coupled multi-angle light scattering (SEC-MALS)

For SEC-MALS of the MBP-NLRP3 (Δ95-134) protein, elution fractions (C7-D4) corresponding to the second peak of the SEC purification were pooled and concentrated to 1.3 mg/mL. The concentrated protein sample was centrifuged at 10,000 x *g* and 4 °C for 10 min before the supernatant was loaded onto a lysis buffer equilibrated Superose 6 increase 10/300 GL column (GE Healthcare), connected to a 1260 Bioinert Infinity LC system and equipped with a multi-angle light scattering detector (miniDawn 3141MD3, Wyatt) and a refractive index detector (Optilab rEX 650, Wyatt). Data were collected every 0.5 s applying a flow-rate of 0.5 mL min$^{-1}$. Data analysis was performed using the Astra 8 software (Wyatt).

## Immunofluorescence analysis

Cells were imaged using a Plan-Apochromat 63×/1.4 oil objective on a Zeiss LSM800 confocal system equipped with an Airyscan module and controlled by the Zen blue software. In brief, cells plated in coverslips (12 mm) were fixed in 4 % PFA in PBS (420801, Biolegend) at 37 °C for 15 min, then washed in PBS 3 times with 5 min interval between washes. Afterwards, cells were permeabilized for 5 min in 0.05 % saponin in PBS for 5 min. Cells were then blocked in blocking buffer (2 % bovine serum albumin (BSA), 0.05 % saponin in PBS) for 1 h. Staining with primary antibodies (2 h to overnight) and secondary antibodies (1 h) was performed in blocking buffer. To remove unbound antibodies, washing steps were performed. Staining of the nuclei was performed with Hoechst 33342. The primary and secondary antibodies used for immunofluorescence in this study are detailed in the Supplementary Table 4 and the Reporting Summary.

## Live cell imaging

Live cell imaging experiments were performed in an LSM 800 (Zeiss) microscope equipped with a 63×1.42 NA Objective (Zeiss). Environmental control was provided by the incubation chamber (controlled temperature, humidity, CO$_2$), a heated stage (Heating insert P Lab-TekTM S1, Zeiss) and an objective heater (Zeiss). Briefly, PMA-differentiated THP-1 Null2 *NLRP3* KO cells reconstituted with WT- or Δexon3-mNG NLRP3 were seeded in 4-chamber glass-bottom dishes (Ibidi). After priming with LPS-EK for 4 h, cells were stimulated while imaging with nigericin (10 μM) or imiquimod (100 μM) in phenol red-deficient RPMI 1640 imaging media (118350-030, Sigma Aldrich). In order to stain cell death, SYTOX™ Orange Nucleic Acid Stain (S11369, Invitrogen, dilution 1/50,000) was added to the media 10 min prior imaging. To stain the microtubule network, SiR-Tubulin incubation was performed for 4 h (Cytoskeleton, with 2 μM SiR-Tubulin and 10 μM verapamil, NC0958386, Thermo Fisher Scientific). Videos were taken for 2 h at intervals of 48-120 s. For microtubule staining in neutrophils, the cells were seeded into PLL-coated 4-chamber dishes at a density of $1 \times 10^6$ cells per well in RPMI + 10 % FBS supplemented with 2 μM SIR-

Tubulin and 10 μM verapamil. Cells were treated as explained in neutrophils stimulation section. Then, nuclei were then stained with 10 μM Hoechst 33342 in PBS for 10 min, and then after washing with PBS, imaging was performed in fresh phenol red-free RPMI + 10 % FBS using the same microscope setup. Analysis of live cell imaging and generation of the different galleries were performed using Fiji[60]. Individual cells were manually tracked (10-20 cells approx. per condition and per experiment), and each frame was added to the ROI manager. Galleries of tracked cells were generated with the automated macro "Measure-Tracks_Gallery" (https://github.com/koschink/Phafin2).

### Size exclusion chromatography analysis of lysates

The different THP-1 cell lines were LPS-primed as described, lysates prepared by sonication and 150 μL lysate separated on a calibrated Superose 6 30/100 analytical column (Cytiva) at 0.25 mL/min using a 25 mM Tris HCl pH 7.4 150 mM NaCl running buffer and collecting 250 μL fractions on an ÄKTA Pure system (Cytiva). MW standards and fractions are shown below. Identical fractions were then separated by 8 % SDS-PAGE and immunoblotting for β-actin and NLRP3 D4D8T antibody, see Supplementary Table 4. Additionally, a fraction of the THP-1 cell lysates obtained by sonication was directly analysed by immunoblotting. Gel filtration standards used were: thyroglobulin (670 kDa, fraction D2), γ-globulin (158 kDa, fraction E2/3), ovalbumin (44 kDa, fraction E7/8), myoglobin (17 kDa, fraction E12) and vitamin B12 (1,35 kDa, fraction F8).

### Immunoblot

After treatments, cell culture supernatants and cell lysates were collected for immunoblot analysis. Cells were lysed for 30 min on ice cold 1× RIPA buffer (50 mM Trizma base pH 7.4, 150 mM NaCl, 1 mM EDTA, 1 % Triton X-100, 0.1 % SDS, 0.5 % sodium deoxycholate, 10 % glycerol) supplemented with one tablet of protease inhibitor cocktail (Roche) for every 10 mL of RIPA. Cell lysates were centrifuged at 16000 x $g$ for 15 min at 4 °C. For the supernatants, the proteins were precipitated and concentrated using methanol–chloroform precipitation and finally resuspended in RIPA buffer. The immunoblots were prepared using Tris-glycine SDS–PAGE and transferred to nitrocellulose membranes (1620115, Bio-Rad) by electroblotting. Membranes were blocked with Tris-buffered saline with 0.1 % Tween 20 (TBS-T) + 5 % BSA (blocking buffer) and probed overnight at 4 °C with primary antibody. After three washes with TBS-T, membranes were incubated with HRP-conjugated secondary antibody in blocking buffer for 1 h at room temperature. Membranes were washed again with TBS-T three times. Proteins were visualized using ECL substrate with chemiluminescent detection (LI-COR Odyssey). The primary and secondary antibodies used for immunoblotting in this study are detailed in the Reporting Summary and in Supplementary table 4.

### In vitro lipid strip assay

Lipid binding assay with purified human NLRP3 variants was performed using PIP strips (10 PK P-6001, Echelon Biosciences) according to manufacturer instructions. In short, the PIP strip membranes were blocked using 3 % BSA in TBS-T for 1 h followed by incubation with 5 mL of human or mouse TEV-digested WT or NLRP3Δexon3 proteins (7.5 μg in total) diluted in 3 % BSA in TBS-T for 2 h. Next, the membranes were washed in TBS-T for 15 min and incubated with the primary antibody NLRP3 Cryo2 1:1000 in TBS-T 3 % BSA for 1 h. Then, the strips were washed in TBS-T for 15 min and incubated with the secondary anti-mouse IgG-HRP antibody at 1:10,000 dilution in TBS-T 3 % BSA for 1 h. Then, the strips were washed in TBS-T for 30 min and the bound proteins were visualized using ECL substrate with chemiluminescent detection (LI-COR Odyssey). All steps performed at room temperature. All samples were analysed at the same time under the same conditions.

### Cytokine measurement

Human IL-1β (437015, Biolegend), IL-18 (DY318-05, R&D Systems) and human TNF-α (88-7346-22, Invitrogen) were measured by triplicate ELISA according to manufacturer's instructions.

### Lactate dehydrogenase assay (LDH)

To measure cell death, the LDH present in cell-free supernatants was detected in triplicate using the Cytotoxicity Detection kit (11644793001, Roche) according to manufacturer instructions, the reaction was read in a Synergy Mx (BioTek) plate reader at 492 nm and corrected at 620 nm.

### SYTOX Orange assay for plate reader measurement

In order to assess cell death, the signal of SYTOX™ Orange Nucleic Acid Stain was measured in the plate reader in real time in living cells. In brief, THP-1 cell lines were prepared for inflammasome activation in 96-well black plates (Thermo Fisher, 165305) in the presence of SYTOX™ orange (S11369, Invitrogen, dilution 1/50,000). Kinetics was measured by detecting SYTOX signal every minute with a filter 530/25,590/35 using the Synergy 2 plate reader (BioTek) at 37 °C in 5 % $CO_2$.

### Antisense oligonucleotides and exon skipping protocol

Antisense oligonucleotides targeting exon 3 were designed following[61]. In this regard, potential target sites for efficient exon skipping were identified using the server Human Splicing Finder (Genomnis SAS Company, France)[62]. Two potential regions were selected based on the prediction of possible exonic splice enhancer sites (ESEs). The predicted mRNA structure by mFoldWeb Server[63] showed that one of those regions was partially open, a necessary requisite for a potential target region. The final designed oligonucleotide was named AON exon3 (5′ CACTCCTCTTCAATGCTGTCTTCCT 3′). Control antisense oligonucleotide (Control AON), AON targeting exon 5[26], AON targeting exon 3, and Endoporter were obtained from gene-tools.com. THP-1 WT cells were treated overnight with PMA (100 ng/mL) for differentiation. Media was changed the day after for complete RPMI. After 48 h, transfection was performed with premixed morpholinos and Endoporter to a final concentration of 10 μM and 6 μM, respectively. Of note, morpholinos were heated up to 65 °C in order to recover full activity before use. 40 h after transfection, cells were LPS-primed and treated for inflammasome activation.

### RNA isolation and *NLRP3* exon-specific RT-qPCR

RNA was isolated using the RNeasy kit (74104, Qiagen), according to the manufacturer's instructions and quantified on Nanodrop 1000 (Thermo Fisher Scientific). Reverse transcription of RNA (800-1000 ng) to generate cDNA was performed using High-Capacity cDNA Reverse Transcription Kit (4368814, Applied Biosystems) following manufacturer's instructions. In order to assess the relative expression of the exon 3 and exon 7, cDNA abundance was measured by qPCR in a QuantStudio6 cycler using TaqMan$^{TM}$ system following manufacturer's instructions. The TaqMan™ Gene Expression Assays (13428456, Thermo Fisher Scientific) used were the following: for TBP (housekeeper) Hs00427620_m1, for NLRP3 exon 2–3, ID Hs00918082_m1; for NLRP3 exon 6-10, ID Hs00918080_g1. Analysis was performed using QuantStudio Real-Time-PCR software.

### RNA sequencing and transcriptome analysis of Δexon 3 THP-1 single clones

Total RNA was extracted from THP-1 WT and the five Δexon3 clonal cell lines (D1 to D5) under untreated and LPS-stimulated conditions. RNA integrity was assessed prior to library preparation (RIN = 10.0 for all samples). Libraries were prepared using poly(A) enrichment and sequenced to a depth of 60–72 million paired-end $2 \times 10^5$ bp reads per sample on an Illumina NovaSeq 6000 platform. Raw reads were processed using nf-core/rnaseq v3.16.1[64]. Reads were aligned to the

Ensembl Homo_sapiens. GRCh38.113 human reference genome using STAR. Transcript-level quantification was performed with Salmon and summarized to gene-level counts using the pipeline's default settings. Default nf-core quality control modules were applied (FastQC, MultiQC). NLRP3 splice junction visualization was performed using ggsashimi 1.1.5 on STAR-aligned BAM files[65]. For Δexon3 clones, tracks were aggregated using the mean_j option to display mean coverage and junction support across clones.

## Image analysis

Image processing and quantification were performed in Fiji software[60]. For colocalization analysis, Pearson's correlation with Costes automatic threshold was applied by using JacoP Plugin[66]. Quantification of NLRP3 specks associated with TGN38 was performed by segmentation of NLRP3 specks with the automatic threshold *Renyi's entropy* and regions of interest (ROIs) were generated with the plugin Analyze particles. Defined ROIs were superimposed onto the channel of interest (TGN38) and mean intensity was measured. The showed images corresponded from a single focal z-plane where we performed colocalization analysis. Number of ASC specks and nuclei were determined by using the plugin Analyze particles and the ratio between them was calculated. The percentage of TGN and MTOC associated specks was performed manually in $3 \times 3$ TILES (63x). Each bar in the graphs represents the mean $\pm$ SD, and each datapoint represents the average of 3 tiles ($3 \times 3$ files of view, FOV) (200 cells approx.). For live cell imaging experiments, timing of speck formation in MTOC-dependent or -independent specks was determined manually. Images show 30 frames from a time-lapse sequence, spaced 2.4 min apart. The distance travelled by WT or Δexon3 specks was determined using Mtrack2 plugin[67] from Fiji. In the endogenously NLRP3 Δexon3 THP-1 single clones, maximum-intensity projections were analyzed in Fiji/ImageJ using a custom Jython script (available on GitHub https://github.com/Chlorinetrifluoride/xulubi) to segment ASC specks, nuclei, and centrosomes (MTOCs), with MTOCs identified by pericentrin staining. After background subtraction and automated threshold-based segmentation, object centroids were extracted, and ASC specks were classified as MTOC-associated based on their proximity to a pericentrin-defined MTOC and to an active nucleus; annotated overlays were exported for quality control. For each experiment, images were randomly acquired with the same settings (laser power, detector gain) and below pixel saturation.

## Screening public transcriptomes for *NLRP3* transcripts missing exon3

To search for a direct NLRP3 exon2-exon4 junction in RNA-Seq read data, we concatenated the last 20 bp of exon 2 with the first 20 bp of exon 4. The resulting sequence AAGAGATGAGCCGAAGTGGGATTACCGTAAGAAGTACAGA was queried against all 186 human RNA-Seq samples in seven NCBI BioProjects (accession IDs: PRJNA378936; PRJNA674655; PRJNA722048; PRJNA789541; PRJNA901389; PRJNA941263; PRJNA978574) using the NCBI SRA nucleotide BLAST service with default search parameters (v2.14.1, linear gap cost with +1 match and −2 mismatch score, word size 28, expect threshold 0.05). No matching reads were found in any of the samples.

## Statistics and reproducibility

Experimental data were analysed using GraphPad Prism 8. The exact value of n is indicated in the figure legends. Normal distribution in each group was analysed using the Shapiro-Wilk test first for the subsequent choice of the parametric (ANOVA when comparing multiple groups or Student's t-test when comparing two groups for normally distributed data) or non-parametric (Mann-Whitney U) test as indicated in the figure legends. Data are always presented as mean $\pm$ SD, *p*-values ($\alpha = 0.05$) were calculated as indicated in the figure legends using GraphPad Prism, and *p*-values $< 0.05$ were considered statistically significant.

## Reporting summary

Further information on research design is available in the Nature Portfolio Reporting Summary linked to this article.

## Data availability

Further information and reasonable requests for resources and reagents should be directed to and will be fulfilled by the corresponding authors Ana Tapia-Abellán (ana.tapia@um.es) and Alexander N.R. Weber (alexander.weber@uni-tuebingen.de).

All materials and data generated during this study are included in this article and its Supplementary Information files or available from the authors as are unique reagents used in this Article. The uncropped western blot, raw numbers for charts and graphs are available in the Source Data file whenever possible. Please contact the lead contact for unique material requests. Any material that can be shared will be released via a material transfer agreement. Source data are provided with this paper.

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

## Acknowledgements

We thank Gloria López-Castejón and Fátima Martin-Sánchez (Institute of Immunology and Inflammation, University of Manchester, UK) for providing THP-1 cells null2 and null2 *NLRP3* KO, Thomas Zillinger (Bonn, Germany) for regular THP-1 cells our students Kim Hebel, Garvit Sharma, Nuse Afahaene and Kenyatta Doumanas (all University of Tübingen) for their help with ELISAs, tissue culture or Python scripting, respectively, Pablo Pelegrin (University of Murcia) for the pcDNA-YFP-NLRP3 Δexon3-LUC plasmid, Kay Oliver Schink (University of Oslo) for pENTR20-mNG plasmid and for sharing very useful Fiji scripts for the analysis of our microscopy data, Clare Bryant and Joseph Boyle (University of Cambridge) for help with CRISPR-Cas9 editing, Michael Schindler (Molecular Virology, University of Tübingen) for the TGN46 plasmid, Vinicius Nunez Cordero Leal for PBMC isolation from selected donor samples, Markus W. Löffler and Laura Velasco Paredes for biobank support and acquiring blood samples, and Kia Wee Tan (Uppsala University) for his useful advice and comments on cellular trafficking. We would also like to sincerely thank the German and Spanish blood donors who participated in this study. Funding The study was supported by the grants from the internal support program of the Medical Faculty, University of Tübingen, Fortüne-Antrag Nr. 2615-0-0 and Nr. 3023-0-0 (to M.M.-T and A.T.-A), the Volkswagenstiftung Momentum grant "InnatelyHuman" (to X.L., A.S. and A.N.R.W.), the DFG (Deutsche Forschungsgemeinschaft/German Research Foundation) for support via the Clusters of Excellence "iFIT-Image Guided and Functionally Instructed Tumor Therapies" (EXC-2180, to A.N.R.W and A.T.-A) and "CMFI- Controlling Microbes to Fight Infections" (EXC-2124, to J.G and A.T-A) and via grants CRC TR156 "The skin as an immune sensor and effector organ – Orchestrating local and systemic immunity" (to A.H., F.B. and A.N.R.W., specifically INST 35/1259-3), and We-4195/18-1 (to A.N.R.W.). Further, we gratefully acknowledge the funding from Fundación Séneca 22413/SF/23 Spain and RYC2023-043193-I funded by MCIU/AEI/10.13039/501100011033 and by the ESF+ (AT-A). L.E. is supported by a PhD scholarship that is funded by the German Academic Scholarship Foundation and the International Max-Planck Research School for the Mechanism of Mental Function and Dysfuntion. M.G. is funded by the European Research Council (ERC Advanced Grant NalpACT), by Germany's Excellence Strategy-EXC2151-390873048, and by a grant of the DFG (GE 976/16-1).

## Author contributions

M.M.-T., A.N.R.W. and A.T.-A conceived the study. M.M.-T., I.V.H., L.F., J.T., D.K.-V., M.G., A.N.R.W and A.T.-A designed the methodology and key resources of the study. M.M.-T., I.V.H., G.L., L.F., A.H., X.L., J.T., L.E., A.S., J.G., F.B., J.S.M. and A.T.-A., performed experiments and analyzed the data. M-M-T, A.S., A.N.R.W and A.T.-A contributed to the visualization of the work. M.M.-T., A.N.R.W and A.T.-A curated the data and wrote the original draft of the manuscript. M.M.-T., I.V.H., D.K.-V., M.G., A.N.R.W., and A.T.-A. reviewed and edited the manuscript. M.M.-T., M.G., A.N.R.W., and A.T.-A. supervised the project. A.N.R.-W. and A.T.-A. administered the project and M.G., A.N.R.W. and A.T.-A. acquired funding.

## Funding

## Competing interests

The authors declare no competing interests.
