## [Transparent Peer Review file · Nature Communications]

Non-decameric NLRP3 reveals a TGN/MTOC-distal pathway of inflammasome activation

Corresponding Author: Professor Alexander Weber

Version 0:

Reviewer comments:

Reviewer #1

(Remarks to the Author)

In their manuscript "Non-decameric NLRP3 forms a TGN/MTOC-independent inflammasome," Mateo-Tórtola et al. clone and investigate an artificial variant of the NLRP3 inflammasome by deleting exon 3. Remarkably, this variant is defective in some aspects of NLRP3 activation while remaining functional in others. This observation leads the authors to propose a model involving two distinct NLRP3 activation pathways: a Golgi-MTOC-proximal pathway and a Golgi-MTOC-distal pathway. The delta3 variant can only activate the latter and, as a result, is not sensitive to Golgi-MTOC inhibition. Interestingly, the potassium efflux-independent NLRP3 agonist imiquimod only activates the Golgi-MTOC-proximal pathway.

This manuscript significantly enhances our understanding of the NLRP3 inflammasome by proposing a model that reconciles previously contradictory and confusing findings in the field. While the experiments are generally well-performed the author's model requires further validation using primary cells and endogenously expressed NLRP3. Furthermore, the idea of two distinct NLRP3 activation pathways has been suggested before. Therefore, the manuscript would benefit from experimentally consolidating its data with previously proposed models.

Major Concerns:

1. Previous Studies: The study PMID: 36384135 suggested that NLRP3 can be activated through two distinct (priming/licensing) pathways: via Nek7 or via IKKb. The authors suggest that the Nek7 pathway corresponds to the Golgi-MTOC-proximal pathway, while the IKKb pathway corresponds to the cytosolic/Golgi-MTOC-distal pathway. Experimentally validating this notion would greatly strengthen the manuscript and help consolidate the field. Specifically, testing if THP1 NEK7 KO cells can respond to imiquimod is recommended. It is expected that NEK7 KO cells would fail to respond to imiquimod but would respond to nigericin. Additionally, NLRP3 activation in LPS-primed mouse NEK7 KO macrophages should be insensitive to colchicine or nocodazole, while in WT macrophages it should be sensitive.
2. Endogenous NLRP3 Expression: The authors need to convincingly demonstrate that their two-pathway activation model holds true with endogenously expressed NLRP3. The AON-exon skipping approach is unconvincing because the phenotype observed with exon5 skipping (positive control) is minor (2 fold reduction only), and the conclusion from Fig. 3F that exon3 skipping does not interfere with inflammasome activation is inconsistent with later data showing slower kinetics in the delta3 variant. I suggest that the authors genetically remove exon 3 in THP1 cells to determine whether endogenously expressed delta3 is functionally impaired and confirm its insensitivity to colchicine or nocodazole.
3. Validation in Primary Cells: The authors should convincingly show that their two-pathway activation model is applicable to primary cells. While the validation of microtubule-independent NLRP3 activation in neutrophils is exciting, it is important to confirm (1) that the inhibitors used indeed disrupt microtubules in these cells at the concentrations tested and (2) validate further hallmarks of this pathway, such as MTOC-distal ASC specks.

Minor Concerns:

1. A table summarizing the differences and commonalities between the two pathways would be very helpful.
2. At times, the wording of the manuscript is imprecise:
 - o Lines 165ff: "WT NLRP3 presented both, TGN-dependent and independent speck formation (Fig. 3A)." The term "dependent" implies causality, which cannot be concluded from immunofluorescence. Consider using "associated/not associated" or "proximal/distal" instead.
 - o Line 220: "endosomal-like structures"—In the absence of direct evidence that these structures are endosomes, this wording should be revised.

o Line 346ff: "NEK7 KO cells respond poorly to imiquimod, as shown recently by Schmacke et al". This has first been published in PMID: 27692612 and should be cited properly. PMID: 36384135 shows that LPS-priming (activating the IKK β pathway) fails to restore imiquimod-mediated NLRP3 activation in Nek7 Kos. I really like this point of discussion, but in the current wording is confusing.

3. Several minor questions should be discussed, though not necessarily experimentally addressed:

o Why is LDH release upon imiquimod potassium-dependent in the delta3 variant but not in WT? (Fig. 6B)

o Nocodazole and the delta3 variant should have the same effect (preventing MTOC-proximal activation). Why does nocodazole inhibit activation, while delta3 only delays it?

o Why is IL-1 release so different between WT and delta3 reconstitution? (Fig. 6A)

Reviewer #2

(Remarks to the Author)

The study by Mateo Tórtola and colleagues utilizes an engineered mutant of human NLRP3 to investigate NLRP3 oligomerization and NLRP3 inflammasome activation. Structural data demonstrated that inactive NLRP3 forms caged decamers on intracellular membranes, particularly the trans Golgi network, leading to dispersed TGN and ultimately engages the microtubule organizing center for activation. However, there are still many conflicting studies regarding organelle localization and oligomerization. The polybasic region in the FISNA domain (exon 4) is necessary for the inactive conformation. There is a second polybasic region in the linker between the PYD and the FISNA-NACHT, which is encoded by exon 3 and is missing in several other species. Delta PYD NLRP3 structures revealed a hexameric structure and that the polybasic region is buried. The authors speculated that this exon 3 polybasic region is necessary for membrane binding and cage orientation and analyzed recombinant NLRP3 delta exon 3 and human NLRP3 KO THP-1 cells restored with WT and NLRP3 delta exon 3. The authors demonstrate that this NLRP3 delta exon 3 forms monomers and dimers and that therefore this linker is essential for oligomerization and TGN binding. Yet, this protein is fully active and assembles inflammasomes. A compelling approach is the inclusion of exon 3 skipping, which further revealed no impact on IL-1 β release. The authors then comprehensively interrogated localization and inflammasome responses. The main premise of this manuscript is that the events described from earlier structural studies are not as black and white and that NLRP3 exists in decamers as well as monomers/dimers during activation and that the linker rather than the FISNA is critical for oligomerization. In addition, the localization analyses performed by the authors also find a combination of TGN-dependent and independent oligomers. Furthermore, the MTOC has been proposed as a final destination required for NLRP3 activation. However, NLRP3 delta exon 3 shows largely MTOC-independent localization. I believe that this study provides important details for researchers working on NLRP3 inflammasomes. In addition, the many conflicting reports of NLRP3 localization during inactive and active state may be explained in part by a mixture of different sized NLRP3 complexes and activation on membranes as well as membrane independent activation in the cytosol. Another key observation is the accelerated inflammasome response when NLRP3 assembles oligomers at the MTOC and presumable on TGN membranes before.

Overall, the study is very comprehensively performed and mostly well controlled. However, the authors should take additional steps to support their conclusion and some points that I would like to make are:

1) However, a major limitation is the fact that the authors could not detect any transcripts that resemble human NLRP3 delta exon 3 in existing RNAseq data sets. Consequently, the study is merely mechanistic using NLRP3 delta exon 3 as an artificial tool to refine NLRP3 activation. However, it should be noted that chicken and zebrafish encode NLRP3 delta exon 3 providing some justification.

2) Most earlier studies have been performed with mouse *Nlrp3* and while the focus of this study is human NLRP3, I believe performing some key imaging/quantification experiments with mouse *Nlrp3* and *Nlrp3* delta exon 3 to be able to conclude whether the observed discrepancies to earlier studies regarding oligomerization and localization are due to technical limitations or are species specific.

3) The study relies on a clone of THP-1 cells from Invivogen restored with NLRP3 and NLRP3 delta exon 3.

4) The size exclusion chromatography should include an "input" sample to allow conclusion on expression of NLRP3 vs NLRP3 delta exon 3 (see point #9 below). Furthermore, the interpretation of these results is not completely accurate. While there seem to be a shift in oligomerization/NLRP3-containing complexes, NLRP3 delta exon 3 still make the same large complexes (fractions 84 instead of 85) and NLRP3 also elutes at the smaller MW fractions. From the background it gives the impression that the NLRP3 samples were longer exposed than the NLRP3 delta exon 3 samples. Therefore, NLRP3 delta exon 3 is still capable of forming the larger oligomers, even though it cannot bind membranes.

5) NLRP3 delta exon 3 only formed TGN independent oligomers, as described by the authors (line 165), while NLRP3 forms both, TGN-dependent and independent oligomers. However, the quantification shown shows reduced number of oligomers, but TGN and MTOC dependent oligomers for NLRP3 delta exon 3 (Fig. 3B). This needs to be corrected.

6) Authors should also demonstrate that the differences observed in localization and dependence on microtubules is not a distinct stage of inflammasome assembly.

7) The authors demonstrate that NLRP3 delta exon 3 expression results in a slower form of inflammasome response (Fig 4). However, in Fig 5, SYTOX analysis seems to show no delayed pyroptosis in NLRP3 delta exon 3 expressing cells,

comparing the slope of the kinetic graph in the absence of Nocodazole.

8) Interesting are the connection of different triggers (nigericin vs imiquimod) and K⁺ dependency to distinct localization patterns and microtubule dependency and hence matching either NLRP3 or NLRP3 delta exon 3 responses. But also here, the authors should ensure that this is not just a “weaker” delayed response. Extending the measuring time may provide additional insights.

9) NLRP3 KO THP-1 cells were restored with NLRP3 or NLRP3 delta exon 3 and utilized for much of the experiments. Part of the distinct results may stem from differences of expression of NLRP3 vs NLRP3 delta exon 3, which imaging does not pick up well. The authors should show a total cell lysate from these cells. This concern is supported by results shown in Fig 6C, where NLRP3 delta exon 3 has a vastly stronger signal as NLRP3, which could skew complex assembly/oligomerization. Also noticeable is that NLRP3 delta exon 3 cells do express many fold reduced pro-IL-1 β or may respond lesser to priming signals. All of which will impact inflammasome response readouts. Inclusion of IL-18, which is less complex regulated on expression level should be considered as well. Utilizing an inducible expression system to dial in comparable expression levels of NLRP3 and NLRP3 delta exon 3, sorting of comparable expressing cell populations or any other form of ensuring equal expression levels that match endogenous NLRP3 are crucial to contrast these responses.

10) The authors mention in line 261 that high extracellular K⁺ did only block MSU induced IL-1 β release but not pyroptosis. However, Fig. S5B clearly shows that this treatment almost completely prevents LDH release. This needs to be clarified.

11) The authors show in Fig 1H that human NLRP3 only binds select phospholipids: PA and PtdIns (3,4,5)-P3 and NLRP3 delta exon 3 failed to bind to any membrane lipids. The authors discuss (line 328) that while they show above result, mouse Nlrp3 has been shown to be more promiscuous and that this hints at important species differences. However, with the shown result, this cannot be concluded, and this assay would need to be performed in parallel with human and mouse NLRP3 for any conclusion.

12) The analysis of human neutrophils and evidence for only a microtubule-independent response in contrast to THP-1 cells misses a positive control of macrophages treated in parallel. This would need to be primary peripheral blood derived macrophages from the same donor differentiated in parallel from the same blood donation.

Minor:

1) Fig 6I: I believe the authors mislabeled treatment conditions of colchicine, nocodazole and nigericin. As shown, it does not make any sense and I believe line 2 rather than line 4 should be nigericin.

2) Methods describe the isolation of primary human peripheral blood derived macrophages, but these cells appear to not have been utilized in this manuscript and the method section should be removed.

Reviewer #3

(Remarks to the Author)

In this paper the authors essentially contend that the ability of NLRP3 to form a cage dictates the path of inflammasome activation. The authors base their findings on use of an exon3 deleted version of NLRP3 expressed in THP1 cells that does not colocalise to the TGN or the MTOC but forms inflammasomes in response to nigericin, but not the K⁺ efflux independent imiquimod. The research area is topical and relevant, but further evidence and significant revising of the manuscript is required before this paper can be accepted. Suggestions are as follows:

1) Some further explanation/analysis is required as to why WT NLRP3 binds to PA and PI3P3 but not PI4P as previously reported (e.g., PMID: 30487600).

2) How do the authors consolidate their observation of ‘smaller ASC puncta’ (Fig S3) with previous work using cryo-light microscopy and cryo-electron tomography to show the ASC speck as a network of filaments (PMID: 37945612), or indeed the superresolution microscopy of endogenous ASC in BMDMs (PMID: 38047065)?

3) To conclude that an active inflammasome is formed at the MTOC colocalization with the ASC speck MUST be shown. It is known from published work that NLRP3 can go to the MTOC but the ASC speck is not observed there (this reference was overlooked (PMID: 39117604)). This must be discussed and data presented showing colocalization NLRP3/ASC/and gamma tubulin. This should be relatively easy to test, and is somewhat surprising that these data are not included.

4) The authors mention, when describing figure 4, that NLRP3 is recruited to endosomal-like structures that traffic to the MTOC. This observation should be validated with colabelling for endosomal markers.

5) The work presented on neutrophils is not enough to conclude that the proposed pathways are the same and further characterisation in these cells should be completed.

6) The second sentence in the discussion states that the TGN to MTOC is the ‘accepted’ localisation pattern of NLRP3 activation. This is not the case and the statement is misleading.

7) The last sentence of the first paragraph of the discussion is not accurate. ASC oligomerisation has been observed on

endosomes (PMID: 36443515).

Version 1:

Reviewer comments:

Reviewer #1

(Remarks to the Author)

I thank the authors for addressing most of my previous comments. The additional data using primary cells are particularly compelling. However, one issue remains inadequately addressed: the analysis of the delta-exon3 variant upon exogenous expression.

(a) Regarding the genetic deletion of exon 3 in THP1 cells: I commend the authors for performing the critical experiment of genetically deleting exon 3 (along with the two adjacent introns) in THP1 cells and functionally analyzing multiple clones. The results indicate that two out of three delta-exon3 clones do not show substantial responses, while one clone exhibits a significantly reduced (10-fold lower) IL-1 β response compared to wild-type cells (R-Fig. 2D). I disagree with the authors' conclusion that these data demonstrate functionality of the delta-exon3 variant when endogenously expressed. I recommend analyzing all clones (including the remaining clones 1 and 6) using an IL-18 ELISA and activating a different inflammasome (AIM2, NLRC4, CARD8). This experiment should identify if indeed priming status or general ability for inflammasome activation is different between these clones or if the delta-exon3 variant is slightly impaired. I suggest including these additional data in the main manuscript. Confirmation of a slight functional impairment or delay of the endogenously expressed delta-exon3 variant would not detract from the authors' overall conclusions, as the variant remains a valuable tool for dissecting mechanisms upon transgenic overexpression.

Additionally, the authors' suggestion that the deletion of exon 3 and adjacent introns might remove essential regulatory elements affecting expression and splicing is difficult to reconcile with the clear demonstration of proper protein expression in the Western blot (R-Fig. 2C).

(b) In their response, the authors justify the exon-skipping morpholino approach by referencing PMID: 31324763. I would like to highlight a crucial difference: exon 5 skipping described in PMID: 31324763 completely abrogates the response, whereas exon 3 skipping in the current manuscript does not impair the response. Thus, the authors' conclusion critically depends on the efficiency of the exon-skipping approach. Given this reliance, and particularly if the authors or editors choose not to include the exon 3-deleted THP1 clones, I strongly recommend validating the morpholino-based exon 3 skipping efficiency shown in Fig. 3E-G by Western blot. This assay is already established by the authors (demonstrated in R-Fig. 2C) and should clearly confirm that most of the NLRP3 protein corresponds to the delta-exon3 molecular weight upon morpholino treatment. The genetically edited clones could also serve as controls to confirm the correct protein size.

One minor additional point: the wording used to describe the exon skipping approach is confusing. For example, Fig. 3E refers to "Exon 3 skipped mRNA,". To me, the wording "unskipped mRNA" makes more sense because the assay detects exon 3 in cDNA. Consistently, the signal decreases upon exon 3-targeting morpholino treatment.

Reviewer #2

(Remarks to the Author)

The authors largely addressed my questions and overall responded well to all reviewers and provided additional experimental data to support their finding, which provides important mechanistic insights into NLRP3 activation.

Reviewer #3

(Remarks to the Author)

The authors have addressed my comments and I think the manuscript makes a useful contribution to the field.

Version 2:

Reviewer comments:

Reviewer #1

(Remarks to the Author)

I thank the authors for their excellent response and for an overall strong manuscript. All comments have been addressed.

Point-by-point response to reviewers' comments:

MS No.: NCOMMS-24-44798

Original MS Title: Non-decameric NLRP3 forms a TGN/MTOC-independent inflammasome.

General: We would like to express our gratitude to the reviewers for their constructive and insightful feedback on our manuscript. Their comments have significantly improved the quality and clarity of our work. In response, we have conducted additional experiments and revised the manuscript accordingly. We believe these changes have strengthened our findings and further substantiated our conclusions. The revised manuscript, now submitted, addresses all their comments. All modifications are highlighted in yellow throughout the revised manuscript for your convenience.

The significance of our study has already been recognized and endorsed by the scientific community. Our preprint, available on bioRxiv (<https://doi.org/10.1101/2023.07.07.548075>)¹, has received over 1,600 full PDF downloads. Furthermore, it has already been cited 11 times in peer-reviewed articles published in high-impact journals, including *Nature Communications*, *Immunity*, and *Molecular Cell*, among others²⁻⁹. In addition, our conceptual framework has been recognized in the review by Bornancin and Dekker (2023)¹⁰, which referred to our model as the new state-of-the-art. Our findings have also sparked discussions in international scientific meetings, such as the EMBO Workshop on Inflammasomes (Munich, September 2024) and the Abcam Innate Immunity Meeting (Boston, December 2023) among others.

We believe these developments reflect the relevance and impact of our study, and we are hopeful that the revised version meets the standards of *Nature Communications*.

Reviewer 1

In their manuscript “Non-decameric NLRP3 forms a TGN/MTOC-independent inflammasome,” Mateo-Tórtola et al. clone and investigate an artificial variant of the NLRP3 inflammasome by deleting exon 3. Remarkably, this variant is defective in some aspects of NLRP3 activation while remaining functional in others. This observation leads the authors to propose a model involving two distinct NLRP3 activation pathways: a Golgi-MTOC-proximal pathway and a Golgi-MTOC-distal pathway. The delta3 variant can only activate the latter and, as a result, is not sensitive to Golgi-MTOC inhibition. Interestingly, the potassium efflux-independent NLRP3 agonist imiquimod only activates the Golgi-MTOC-proximal pathway.

Comment #1: This manuscript significantly enhances our understanding of the NLRP3 inflammasome by proposing a model that reconciles previously contradictory and confusing findings in the field.

Author reply #1: We appreciate the reviewer's comment and positive reception of our work.

Comment #2: While the experiments are generally well-performed the author's model requires further validation using primary cells and endogenously expressed NLRP3. Furthermore, the idea of two distinct NLRP3 activation pathways has been suggested before.

Author reply #2: In response to the reviewer's suggestion, we have incorporated a new set of experimental data in **Figure 7** where we investigate the two NLRP3 pathways at the endogenous level in iPSC-derived microglia, as well as in primary human macrophages and neutrophils. This confirms the physiological existence of both MTOC-distal and MTOC-associated NLRP3 activation pathways. We agree with the reviewer that there have been suggestions for different pathways to exist, however these have never been considered or systematically tested to really occur in parallel. Recently, the well accepted MTOC-dependency model thus somewhat called data on cytosolic signaling into question. Our work now demonstrates the parallel nature of both pathways and thus reconciles both separate sets of reports into a first unified model, an advance in the field we would like to stress as non-incremental but lifting our understanding of NLRP3 pathways to a completely refined level.

Comment #3: Therefore, the manuscript would benefit from experimentally consolidating its data with previously proposed models.

Author reply #3: We thank the reviewer for this suggestion. Throughout we had always performed MTOC-associated and MTOC-independent analyses and have now additionally e.g., to test the influence of NEK7 (see next). However, given the plethora of studies that have contributed to our current understanding of NLRP3 signaling, we had to limit ourselves to key models. Our focus from the outset was the human system and we contend that our results in the systems/models presented (including both cell lines and primary human immune cells) are now sufficient to enable other researchers to further validate our new concept in their additional models including murine systems.

Comment #4: Previous Studies: The study PMID: 36384135 suggested that NLRP3 can be activated through two distinct (priming/licensing) pathways: via Nek7 or via IKKb. The authors suggest that the Nek7 pathway corresponds to the Golgi-MTOC-proximal pathway, while the IKKb pathway corresponds to the cytosolic/Golgi-MTOC-distal pathway. Experimentally validating this notion would greatly strengthen the manuscript and help consolidate the field. Specifically, testing if THP1 NEK7 KO cells can respond to imiquimod is recommended. It is expected that NEK7 KO cells would fail to respond to imiquimod but would respond to nigericin. Additionally, NLRP3 activation in LPS-primed mouse NEK7 KO macrophages should be insensitive to colchicine or nocodazole, while in WT macrophages it should be sensitive.

Author reply #4: We had also considered this intriguing possibility but had decided to not investigate this question for concern of complicating the issue: the reason is that human NEK7 has been mainly considered in priming¹¹, whereas we are investigating activation pathways (both MTOC-distal and MTOC-proximal pathways are studied upon LPS priming). Moreover, despite the elegant mechanism offered by Schmacke and Hornung (who we are in close contact with), the Golgi-association of either priming pathway was only tested in relatively artificial setup, using mouse J774 immortalized cell line KO for Nlrp3 and ASC and constitutively expressing mouse Nlrp3 (meaning that ASC was not present in this experimental set up leading to a dysfunctional downstream pathway). Additionally, in the case of imiquimod stimulation, immortalized mouse macrophage cell line (mmMacs) constitutively expressing mouse Nlrp3 were used. Moreover, despite this publication, the role of NEK7 is still debated in the field. We would therefore not like to distract from the main thrust of our work by exploring, in detail, the relative roles of NEK7 vs

IKK β which was also not suggested by any of the other reviewers. The role of IKK β was thus considered future work but, since we commented on NEK7 in the original submission, and to satisfy the reviewer's and our curiosity we have taken two approaches during the revision of this manuscript. Firstly, we sought to source NEK7-deficient THP-1 cells from the Hornung lab (a switch to the mouse system seemed to exceed the scope of the current work for two reasons: firstly, our focus was human NLRP3; the role of Nek7 in mouse appears consistently priming only, which was not investigated here). Despite our and their efforts, since requesting the cells in January 2025, they have not been able to send us viable cells to date so that the experiment was not possible to perform. However, we were able to embark upon a second approach, using a newly published NEK7 degrader, NK7-902¹² now in press with *Cell Chemical Biology*). Upon validation of the same NEK7-degrading effect of the compound (**Reviewer Figure 1A**), stimulations of THP-1 sorted cell lines with similar expression levels for WT and Δ exon 3 (new **Figure S5C**), were performed using K⁺ dependent (nigericin) and independent (imiquimod) stimuli, all used at two different concentrations. As evident in **Reviewer Figure 1B** we did not observe clear differences in the IL-1 β neither IL-18 releases in the presence or absence of NEK7 for none of the cell lines nor stimuli used.

Collectively, we conclude that NEK7 seems dispensable for human NLRP3 inflammasome activation in our system. Our data confirm that TGN-associated NLRP3 seems to be independent of NEK7 as reported by Schmacke *et al*; but also, in the case of TGN-non associated NLRP3 during activation (which is also plausible as NEK7 resides at the MTOC). However, despite this substantial analysis, we now do not feel confident concluding whether NEK7 and MTOC association are systematically aligned or not. In light of these reviewer results, we have revised and toned down the NEK7 discussion section accordingly (see **lines 405-414**).

Comment #5: Endogenous NLRP3 Expression: The authors need to convincingly demonstrate that their two-pathway activation model holds true with endogenously expressed NLRP3. The AON-exon skipping approach is unconvincing because the phenotype observed with exon5 skipping (positive control) is minor (2 fold reduction only), and the conclusion from Fig. 3F that exon3 skipping does not interfere with inflammasome activation is inconsistent with later data showing slower kinetics in the delta3 variant. I suggest that the authors genetically remove exon 3 in THP1 cells to determine whether endogenously expressed delta3 is functionally impaired and confirm its insensitivity to colchicine or nocodazole.

Author reply #5: We appreciate the reviewer's suggestion and would like to clarify several key aspects regarding of our exon skipping experiments, including the controls used, the robustness of the results, and the follow-up genetic approach—performed in response to the reviewer's comment—to assess the functional impact of deleting exon 3 in the endogenous NLRP3 locus. Firstly, the experiment involving exon 5 skipping was performed using a morpholino previously published in Hoss *et al.*,¹³ where exon 5 was shown to be important for inflammasome activation in primary human monocyte-derived macrophages (hMDMs). However, that same study also reported that NLRP3 lacking exon 5 could recover functionality after prolonged LPS priming, indicating that the effect is context-dependent and not absolute. In our hands, using the same morpholino in THP-1 cells, we observed a modest but significant reduction in IL-1 β release. We believe this difference likely reflects the distinct cellular context (THP-1 vs. hMDMs) and the specific priming conditions used in our system. For this reason, we do not consider the exon 5 morpholino experiment to be a positive control for validating exon skipping in THP-1 cells. Instead, for the exon 3 skipping experiments, we rely on an RT-qPCR assay that spans exon 3 to quantify the proportion of NLRP3 transcripts in which exon 3 has been successfully skipped. This is paired with a total NLRP3 qPCR (targeting exons 7–8) to normalize against total expression. Using this approach, we consistently measured >80 % exon 3 skipping in THP-1 cells treated with the targeting morpholino whilst completely retaining functionality, so we are quite confident that the results are robust.

Furthermore, the results obtained for the IL-1 β release are endpoint measurements, not reporting on kinetic differences between WT and Δ exon3 upon 1 h nigericin activation (see Fig 3F vs 4F). Whilst we concede that the exon skipping approach has limitations, the data seem consistent and not in contradiction with other results. In fact, reviewer 2 considers the exon skipping approach “compelling” (underlined in reviewer 2’s comments below), so we decided to keep this data. Nevertheless, we have undertaken the non-trivial task of editing the endogenous exon 3 in THP-1 cells. This has taken several months but we finally obtained clones with Δ exon 3 deletion via double scission, confirmed by PCR with exon-specific primers, Sanger sequencing and immunoblot (**Reviewer Figure 2A-C**). Precise editing at the level of mRNA transcripts has so far not been validated. We conducted preliminary experiments focusing on IL-1 β release as a basic characterization step, confirming that the exon 3 deletion variant was still functional (**Reviewer Figure 2D**). However, the behavior of the three CRISPR/Cas9 THP-1 single-cell clones lacking exon 3 was wildly inconsistent—one clone never activated, a second released only a trace of IL-1 β , and a third produced a respectable IL-1 β signal but still did not reproduce the results obtained with the morpholino approach. Such divergence is common once THP-1 cells are taken through so many passages for clonal isolation. In addition, CRISPR editing brings its own off-target risk, especially in a locus as GC-rich as NLRP3. Importantly, deleting an intron–exon–intron segment via CRISPR double-cutting can inadvertently disrupt essential splicing motifs, enhancer elements, or RNA structural features embedded within the introns—features that are preserved during exon skipping. These results are consistent with published evidence indicating that 45–60 % of alleles edited with two CRISPR-induced double-strand breaks undergo aberrant splicing, such as intron retention or activation of cryptic splice sites¹⁴. In addition, 50–70 % of long introns (>1 kb) contain regulatory elements—including enhancers and CTCF sites—that contribute to proper gene expression and chromatin structure^{15,16}. As shown by Javierre *et al.*, and Zhang *et al.*, deleting these regions can impair transcription or chromatin looping even if coding exons remain intact^{17,18}. Thus, while exon 3 itself is dispensable for function, the way it is removed—preserving regulatory and splicing context versus deleting surrounding intronic regions—may have a major impact on protein functionality and also other cellular processes.

For a clean test of the “endogenous Δ exon 3” hypothesis we therefore still advocate for an antisense morpholino that forces skipping of exon 3 in the entire polyclonal culture in a single overnight step. The splice blocker is confined to the exon sequence (no junction homology), minimizing off-target activity; drugs with the same chemistry are already in clinical use. We show data (**Reviewer Figure 2E-G**) where skipping efficiency reaches ~85 % by RT-qPCR, yet LPS-induced TNF- α remains unchanged, showing that priming circuitry and cell fitness are intact. Under these native-expression conditions, the exon-skipped population releases the same IL-1 β as the cells treated with a control morpholino or without morpholino, confirming that the phenotype is intrinsic to loss of the linker and not an artefact of over-expression or clonal drift.

Especially, in combination with assessing MTOC-associated vs -independent pathway in primary cells - which also proves the existence of both pathways in the complete absence of genetic editing and any clonal or off-target effects thereof – we think exon skipping combined data are an appropriate way to show the existence of both pathways for endogenous NLRP3.

Comment #6: Validation in Primary Cells: The authors should convincingly show that their two-pathway activation model is applicable to primary cells. While the validation of microtubule-independent NLRP3 activation in neutrophils is exciting, it is important to confirm (1) that the inhibitors used indeed disrupt microtubules in these cells at the concentrations tested and (2) validate further hallmarks of this pathway, such as MTOC-distal ASC specks.

Author reply #6: We agree with the reviewer that more results from primary cells are of great interest and have tested the pathway model further in iPSC-derived microglia, primary human

neutrophils, as well as matched primary monocytes/macrophages. Please see new **Figure 7** and the newly added manuscript section: "*Human iPSC-microglia and macrophages activate both NLRP3 pathways, while neutrophils rely on the MTOC-independent one.*"

One of the reviewers' concerns was whether the microtubule inhibitors are indeed effective at the concentrations used. We confirmed this by live cell microscopy in primary human neutrophils from n=3 biological replicates, data included as new **Figure S6A**. Thus, despite effectively disrupting the microtubular network, IL-1 β release and cell death were unaffected, suggesting NLRP3 activation proceeds independently of the microtubular network (**Figure 7C**). Another concern was the demonstration of MTOC-distal ASC specks. We have also performed staining of ASC and pericentrin (core centromere/MTOC) in fixed neutrophils, which showed that ASC specks formed distally to pericentrin (new **Fig. 7D-E**). Collectively, we thus provide additional data from primary cells, which we hope will satisfy reviewer 1 and also reviewer 3 (see comment #31).

Comment #7: A table summarizing the differences and commonalities between the two pathways would be very helpful.

Author reply #7: We sincerely thank the reviewer for the insightful suggestion. A new **Table S1** summarizing the differences and similarities between the two pathways has now been included in the supplementary information of the manuscript.

Comment #8: At times, the wording of the manuscript is imprecise:

-Lines 165ff: "WT NLRP3 presented both, TGN-dependent and independent speck formation (Fig. 3A)." The term "dependent" implies causality, which cannot be concluded from immunofluorescence. Consider using "associated/not associated" or "proximal/distal" instead.

-Line 220: "endosomal-like structures"—In the absence of direct evidence that these structures are endosome, this wording should be revised.

-Line 346ff: "NEK7 KO cells respond poorly to imiquimod, as shown recently by Schmacke et al". This has first been published in PMID: 27692612 and should be cited properly. PMID: 36384135 shows that LPS-priming (activating the IKK β pathway) fails to restore imiquimod-mediated NLRP3 activation in Nek7 Kos. I really like this point of discussion, but in the current wording is confusing.

Author reply #8: We thank the reviewer for the suggestion and have modified the text accordingly. As kindly suggested by the reviewer we have replaced the terms 'TGN-dependent' and 'TGN-independent' with 'TGN-associated' and 'TGN-non-associated' and applied this terminology consistently throughout the manuscript where appropriate (e.g. **line 176**). Additionally, see **line 235**, where 'endosomal-like structure' has been revised to 'vesicular structures'. Additionally, we apologize for the incorrect citation. This has now been corrected, as accurately pointed out by the reviewer the suggestion that imiquimod-induced NLRP3 activation requires NEK7 was first reported by the group of Olaf Groß¹⁹ (**line 409**).

Comment #9: Several minor questions should be discussed, though not necessarily experimentally addressed:

Why is LDH release upon imiquimod potassium-dependent in the delta3 variant but not in WT? (Fig. 6B).

Author reply #9: We appreciate the reviewer's assessment and this prompted us to expand and refine our experimental design, leading us to clarify some concerns shared by reviewers. To see a full response covering several aspects related to this concern please go to comment #21.

Comment #10: Nocodazole and the delta3 variant should have the same effect (preventing MTOC-proximal activation). Why does nocodazole inhibit activation, while delta3 only delays it?

Author reply #10: Microtubule disruption induced by nocodazole prevents NLRP3 WT from accessing the microtubule network, thereby impairing its ability to traffic and associate with the MTOC. As a result, the only remaining NLRP3 activation pathway is the one that does not involve MTOC association, which occurs with a delayed kinetics similar to the Δ exon3 phenotype (see **figure 5A-C**), presumably because assembly of an active disk from monomers/dimers is slower than from a pre-assembled oligomeric cage. Moreover, the deletion of exon 3 shifts the equilibrium far to the monomeric/dimeric state, from one rapid and functional to another slower, but also functional pathway. On the other hand, microtubule disruption does not do this: it rather means that oligomeric cages remain trapped on membranes without reaching the MTOC for ASC/caspase-1 engagement or compensation via NLRP3 redistributed to the MTOC-distal pathway. Finally, we and others cannot exclude that nocodazole and/or colchicine, despite being commonly used in the field ²⁰, may affect additional aspects of NLRP3 signaling that are not affected by exon 3 deletion and not directly related to these specific pathways, but nevertheless microtubule-dependent and connected to NLRP3, e.g. mitochondrial positioning ²¹. As we consider the reviewer's point a truly valid question, we have added some of these considerations to the discussion of the paper (see **lines 415-416**).

Comment #11: Why is IL-1 release so different between WT and delta3 reconstitution? (Fig. 6A)

Author reply #11: In our initial, unsorted THP-1 pools, NLRP3 Δ exon3 cells exhibited lower pro-IL-1 β expression than WT, likely due to reduced/less effective priming. Consistent with this, Δ exon3 pools also released less IL-1 β overall. We now recognize that this priming issue may have masked the full responsiveness of Δ exon3. In response to the reviewers' concerns regarding this figure, we therefore decided to also measure IL-18, a cytokine also processed upon inflammasome activation but whose transcription is independent of LPS priming (as suggested by reviewer 2). Upon nigericin stimulation, comparable levels of this cytokine between both cell lines were observed (see **new Figure 6D**). Furthermore, we sorted THP-1 cells expressing either NLRP3 WT or Δ exon3 to similar expression levels and perform a new set of experiments (see **new Figure S5C-E**). At intermediate NLRP3 expression levels, both cell lines released similar amounts of IL-1 β upon nigericin stimulation. These new results have been incorporated into the manuscript accordingly (see **lines 295-301**).

Reviewer 2

The study by Mateo Tórtola and colleagues utilizes an engineered mutant of human NLRP3 to investigate NLRP3 oligomerization and NLRP3 inflammasome activation. Structural data demonstrated that inactive NLRP3 forms caged decamers on intracellular membranes, particularly the trans Golgi network, leading to dispersed TGN and ultimately engages the microtubule organizing center for activation. However, there are still many conflicting studies regarding organelle localization and oligomerization. The polybasic region in the FISNA domain (exon 4) is necessary for the inactive conformation. There is a second polybasic region in the linker between the PYD and the FISNA-NACHT, which is encoded by exon 3 and is missing in several other species. Delta PYD NLRP3 structures revealed a hexameric structure and that the polybasic region is buried. The authors speculated that this exon 3 polybasic region is necessary for membrane binding and cage orientation and analyzed recombinant NLRP3 delta exon 3 and human NLRP3 KO THP-1 cells restored with WT and NLRP3 delta exon 3. The authors demonstrate that this NLRP3 delta exon 3 forms monomers and dimers and that therefore this linker is essential for oligomerization and TGN binding. Yet, this protein is fully active and assembles inflammasomes. A compelling approach is the inclusion of exon 3 skipping [Our highlight, see comment #5], which further revealed no impact

on IL-1 β release. The authors then comprehensively interrogated localization and inflammasome responses. The main premise of this manuscript is that the events described from earlier structural studies are not as black and white and that NLRP3 exists in decamers as well as monomers/dimers during activation and that the linker rather than the FISNA is critical for oligomerization. In addition, the localization analyses performed by the authors also find a combination of TGN-dependent and independent oligomers. Furthermore, the MTOC has been proposed as a final destination required for NLRP3 activation. However, NLRP3 delta exon 3 shows largely MTOC-independent localization. I believe that this study provides important details for researchers working on NLRP3 inflammasomes. In addition, the many conflicting reports of NLRP3 localization during inactive and active state may be explained in part by a mixture of different sized NLRP3 complexes and activation on membranes as well as membrane independent activation in the cytosol. Another key observation is the accelerated inflammasome response when NLRP3 assembles oligomers at the MTOC and presumable on TGN membranes before.

Comment #12: Overall, the study is very comprehensively performed and mostly well controlled.

Author reply #12: We thank reviewer 2 for this favorable assessment.

Comment #13: However, the authors should take additional steps to support their conclusion and some points that I would like to make are: However, a major limitation is the fact that the authors could not detect any transcripts that resemble human NLRP3 delta exon 3 in existing RNAseq data sets. Consequently, the study is merely mechanistic using NLRP3 delta exon 3 as an artificial tool to refine NLRP3 activation. However, it should be noted that chicken and zebrafish encode NLRP3 delta exon 3 providing some justification.

Author reply #13: We thank the reviewer for highlighting this important point. We sought to emphasize that our primary goal was to employ a carefully engineered human NLRP3 (\$\Delta\$ exon3) construct as a mechanistic and nature-inspired (see below) tool (rather than to document a naturally occurring human splice variant) to delineate a novel pathway that subsequently was confirmed in primary cells. We have sought to clarify this even more in the revised results (lines 117-124) and discussion (lines 379-388) sections. To summarize, our rationale was as follows:

- **Functional exon architecture:** We based this approach on the principle that exons often encode distinct functional blocks within a protein, which was specifically shown for NLRP3^{13,22}. Specifically, exon 3 in human NLRP3 encodes a 40-amino-acid linker region between the PYD and NACHT domains—an element essential for forming the decameric, “cage-like” oligomers (see Figures 1–2 in the manuscript). By removing this discrete exon, we both preserved the core structural integrity (i.e., the PYD domain and the remaining FISNA-NACHT domain) and simultaneously disrupted the formation of the cage-like oligomers. This design allowed us to investigate whether non-cage NLRP3 species can still drive inflammasome activation, without compromising the rest of the NLRP3 architecture or other potential functions. Our approach thus avoids the inherent and common issue of precisely interpreting results from, e.g., loss-of-function deletions or mutations.
- **Evolutionary precedent:** As the reviewer notes and we state in the manuscript, chicken and zebrafish naturally encode shorter NLRP3 variants lacking exon 3 and they are still functional^{23–25}. This reveals that the absence of this region need not abrogate function and, in principle, smaller NLRP3 variants can be physiologically relevant under certain conditions.
- **Additional supporting information.** Moreover, as shown in Figure S1A, all NLRPs—except NLRP4 and NLRP9, -10 and -11—contain between 1 and 4 exons coding for a PYD-NACHT linker. Both NLRP10 and NLRP11 have been shown to form functional inflammasomes without such a linker^{26,27}, highlighting a possible redundancy of this region for

inflammasome activation. This additional information collectively indicates that our Δ exon3-NLRP3 should be considered a plausible and useful tool to explore NLRP3 functionality in the absence of cage formation.

Consequently, the lack of evidence for an exon 3-less NLRP3 in the human system should not compromise the relevance of the study, especially now that we provide much more evidence from human primary or primary-like cells (new **Figure 7**). Even though Δ exon 3 thus does not exist in cells, the two pathways it helped us delineate, do. We have clarified further the intended mechanistic purpose of Δ exon3 within the Discussion, ensuring that readers understand its limitations and significance in dissecting structural and functional aspects of NLRP3. activation.

Comment #14: Most earlier studies have been performed with mouse Nlrp3 and while the focus of this study is human NLRP3, I believe performing some key imaging/quantification experiments with mouse Nlrp3 and Nlrp3 delta exon 3 to be able to conclude whether the observed discrepancies to earlier studies regarding oligomerization and localization are due to technical limitations or are species specific.

Author reply #14: This reviewer flags up an important point – ideally our model would now be validated broadly for the mouse system. Whilst we consider this very interesting, it appears outside the scope of our present study as we here focused entirely on the human system, which we consider most relevant for translational avenues. Nevertheless, we wanted to acknowledge the reviewer’s curiosity by generating an equivalent mouse Nlrp3 (Δ exon3) construct and compare its steady-state localization to mouse WT and human WT and Δ exon3 NLRP3 in transiently transfected HeLa cells. In these experiments, mouse Nlrp3 showed a distribution pattern similar to previous observations for human NLRP3 (**Reviewer Figure 3A**). Further TGN labeling using TGN46-mCherry plasmid transfection showed that mouse Nlrp3 exhibited weaker Golgi staining, possibly due to its expression in a human cellular background. Nonetheless, for both human and mouse NLRP3, deletion of exon 3 resulted in a complete loss of TGN localization (**Reviewer Figure 3B**). Overall, we consider it too preliminary to speculate on the relative importance of MTOC-associated vs MTOC-non-associated pathways in mouse immune cells, although comparable initial localization would be indicative.

Comment #15:The study relies on a clone of THP-1 cells from Invivogen restored with NLRP3 and NLRP3 delta exon 3.

Author reply #15: This information is accurate, and we opted for this commercially available engineered cell line so that other researchers can easily follow up on our work in the same cellular system. While we acknowledge (and discussed, see **lines 476-479**) the limitations of cell-line-based experiments^{28,29}, they offer several important advantages:

- **Ease of genetic manipulation and reproducibility:** By using a stable, well-characterized THP-1 background and introducing both WT NLRP3 and NLRP3 (Δ exon3) fluorescence-tagged constructs via lentiviral transduction, we created uniform cell populations that differ primarily in the expressed NLRP3 variant that could be assessed side-by-side. Stable integration in the same parental THP-1 line reduces variability between conditions, thereby facilitating more reliable comparisons of the structural and functional effects of the NLRP3 (Δ exon3) deletion.
- **Verification in primary cells:** Although immortalized cells cannot fully recapitulate the complexity of primary human tissues, we have confirmed critical findings in human primary macrophages (hMDMs)—such as TGN proximal and distal speck formation and included a first set of data for human primary neutrophils, macrophages and iPSC-induced microglia (new **Figure 7**).

Taken together, we believe the reproducibility and ease of manipulation of the THP-1 system, combined with complementary data in primary cells, support the robustness and physiological significance of our conclusions regarding the NLRP3 inflammasome, bearing in mind that many highly cited mechanistic papers in the NLRP3 field are based mainly on immortalized cells only.

Comment #16: The size exclusion chromatography should include an “input” sample to allow conclusion on expression of NLRP3 vs NLRP3 delta exon 3 (see point #9 below). Furthermore, the interpretation of these results is not completely accurate. While there seem to be a shift in oligomerization/NLRP3-containing complexes, NLRP3 delta exon 3 still make the same large complexes (fractions 84 instead of 85) and NLRP3 also elutes at the smaller MW fractions. From the background it gives the impression that the NLRP3 samples were longer exposed than the NLRP3 delta exon 3 samples. Therefore, NLRP3 delta exon 3 is still capable of forming the larger oligomers, even though it cannot bind membranes.

Author reply #16: We thank the reviewer for flagging up this point and would like to respond as follows to the different points raised:

- **Addition of input sample:** We have now included an additional Western blot (“input”) to demonstrate the expression levels of both NLRP3 and NLRP3(Δ exon3) (added to Figure 2 as new **Figure 2E**), thus addressing the reviewer’s request to visualize overall protein abundance prior to SEC fractionation. In addition, we included β -actin as a loading control in these fractionations to confirm that equivalent amounts of total protein were loaded for each cell line during the SEC experiments shown in **Figure 2A-C**.
- **Equal exposure conditions:** We exposed the blots for NLRP3 and NLRP3(Δ exon3) under the same conditions (exposure time 10 min for NLRP3 and 2 min for β -actin) **Figure 2A-D**. The Δ exon3 blot sometimes appears “cleaner” (i.e., less background) because the Δ exon3-expressing cells produce more total protein; the stronger primary band in those lanes accentuates the contrast relative to background, making the blot look less “dirty.” We have clarified this point in the corresponding figure legend to avoid confusion.
- **Shifted equilibrium rather than exclusive dimers/monomers formation:** As the reviewer notes, both proteins elute with both larger and smaller oligomeric species, which will be in a certain equilibrium with each other. In the case of WT NLRP3, this is presumably assisted by membrane interactions which would be expected to promote the oligomer. Our aim was to emphasize that deleting exon 3 shifts the equilibrium away from high-molecular-weight oligomers (e.g., decamers) toward smaller assemblies (such as monomers and dimers) rather than eliminating large species entirely. Indeed, it has been shown that NLRP3 constructs lacking the PYD-linker can still form hexamers³⁰. Hence, we do not rule out the presence of higher-order Δ exon3 species (fractions near void volume), but these may also be structurally distinct from the canonical decameric cage observed for WT NLRP3 and relate to face-to-face dimers³¹. Based on the reviewer’s helpful advice, we have adapted our phrasing in line to reflect the shift of the equilibrium, rather than exclusive oligomer vs monomer/dimer distribution, more adequately in **lines 168-170**.

Collectively, we have added control lysate blots (new **Figure 2E**) and amended the phrasing to more accurately reflect the observed shift in equilibrium between oligomers and dimers.

Comment #17: NLRP3 delta exon 3 only formed TGN independent oligomers, as described by the authors (line 165), while NLRP3 forms both, TGN-dependent and independent oligomers. However, the quantification shown shows reduced number of oligomers, but TGN and MTOC dependent oligomers for NLRP3 delta exon 3 (Fig. 3B). This needs to be corrected.

Author reply #17: We thank the reviewer for highlighting this oversight. We have amended the relevant passage (**lines 204-207**) to clarify that most oligomers formed by NLRP3(Δ exon3) are

indeed non TGN/MTOC-associated, but a small fraction can still localize to the TGN or MTOC. This revised wording corrects the earlier impression that NLRP3(Δ exon3) never forms TGN/MTOC-associated assemblies, while still supporting our conclusion that the majority of Δ exon3 specks do not associate on these compartments. Accordingly, we have adjusted the tone throughout this section, replacing terms such as 'only' with 'mostly/predominantly' (**line 176 and line 203**). Furthermore, in line with Reviewer 1's suggestion (see comment #8), we have also replaced 'TGN- or MTOC-dependent/independent' with 'TGN- or MTOC-associated/non-associated' where appropriate to more accurately reflect the data.

Comment #18: Authors should also demonstrate that the differences observed in localization and dependence on microtubules is not a distinct stage of inflammasome assembly.

Author reply #18: We appreciate the reviewer's concern but feel this is a very difficult to address point not least because the precise kinetics have not been sufficiently explored in other studies. For example, a hallmark study in the field that described inactive NLRP3 decamers localized at the TGN³², relied on microscopy data obtained from fixed samples. Thus, observations were limited to specific time points—either after LPS priming or 60 minutes following NLRP3 activation with nigericin. Our live cell imaging data, which followed the same single event over time (please refer to **Supplementary videos 1-4** and the timelines shown in **Figures 4, 5 and 6 and S4**), suggest that the MTOC-associated and -non associated specks we observe correspond to genuinely distinct pathways, rather than different stages of the same pathway. This is based on the following points:

- **Stable post-assembly localization:** In our live-cell recordings, once an MTOC-associated speck formed, it remained anchored to the MTOC and did not relocate **Figure 4A-B, 5B, 6H, S4A** and **Supplementary movies 1 to 4**. Likewise, specks that formed away from the MTOC consistently remained at those distal sites after assembly. Accordingly, similar results were observed in²⁰.
- **Speck mobility analysis:** We measured speck displacement (i.e., speed) **Figure S4B** and observed that MTOC-associated specks displayed minimal movement post-formation, whereas cytosolic (non MTOC-associated) specks could traverse the cytoplasm until maturing. If one pathway simply transitioned through different "stages," we would expect specks to move from the MTOC to a distal site, or vice versa, but we did not detect such transitions.

Taken together, these findings strongly indicate that MTOC-associated and -non associated specks do not represent sequential phases; instead, they are formed through parallel assembly processes that remain spatially stable once they emerge.

Comment #19: The authors demonstrate that NLRP3 delta exon 3 expression results in a slower form of inflammasome response (Fig 4). However, in Fig 5, SYTOX analysis seems to show no delayed pyroptosis in NLRP3 delta exon 3 expressing cells, comparing the slope of the kinetic graph in the absence of Nocodazole.

Author reply #19: The slower kinetics of NLRP3(Δ exon3)-driven inflammasome activation are predominantly seen in single-cell analyses. In the NLRP3 speck imaging microscopy-based SYTOX cell death analysis (**Figure 4D**), this approach captures per-cell timing precisely. In contrast, in **Figure 5A**, the 96-well plate format SYTOX assay measures pyroptosis in bulk cell populations. Subtle differences in the timing of cell death across individual cells may get averaged out in a population readout, thus obscuring the slightly delayed kinetics observed at the single-cell level. We appreciate that the current rendition of these differences was missing in the original submission, leading to confusion. We have sought to amend this in the revised version of the manuscript in the discussion section (**lines 448-453**). We hope this clarifies the differences between methods sufficiently to resolve the query.

Comment #20: Interesting are the connection of different triggers (nigericin vs imiquimod) and K⁺ dependency to distinct localization patterns and microtubule dependency and hence matching either NLRP3 or NLRP3 delta exon 3 responses. But also here, the authors should ensure that this is not just a “weaker” delayed response. Extending the measuring time may provide additional insights.

Author reply #20: Please see comment below, author reply #21.

Comment #21: NLRP3 KO THP-1 cells were restored with NLRP3 or NLRP3 delta exon 3 and utilized for much of the experiments. Part of the distinct results may stem from differences of expression of NLRP3 vs NLRP3 delta exon 3, which imaging does not pick up well. The authors should show a total cell lysate from these cells. This concern is supported by results shown in Fig 6C, where NLRP3 delta exon 3 has a vastly stronger signal as NLRP3, which could skew complex assembly/oligomerization. Also noticeable is that NLRP3 delta exon 3 cells do express many fold reduced pro-IL-1 β or may respond lesser to priming signals. All of which will impact inflammasome response readouts. Inclusion of IL-18, which is less complex regulated on expression level should be considered as well. Utilizing an inducible expression system to dial in comparable expression levels of NLRP3 and NLRP3 delta exon 3, sorting of comparable expressing cell populations or any other form of ensuring equal expression levels that match endogenous NLRP3 are crucial to contrast these responses.

Author reply #21: We here would like to offer a combined response to comments #9, #20 and #21. We appreciate the reviewer’s concern that the imiquimod results could simply reflect a slower, weaker signal and that unequal protein abundance might underlie the WT / \$\Delta\$ exon3 differences. We addressed those points in the suggested order: (i) use a priming-independent cytokine, (ii) document protein levels, and (iii) repeat the functional assays in expression-matched cells. First, we re-examined the frozen supernatants from our original experiments and measured IL-18, which is made constitutively and therefore bypasses any priming deficit. The data, now included in new **Figure 6D** and described in **lines 290-295**, show that imiquimod drives as much IL-18 release from the \$\Delta\$ exon3 as from WT (matching the results from LDH release also shown in **Figure 6B**). Intriguingly, high extracellular K⁺ abolishes the \$\Delta\$ exon3 IL-18 signal while leaving the WT signal largely intact (as noted by reviewer 1 also for LDH release data). Our interpretation is that imiquimod still produces a modest K⁺ efflux (also observed by others¹¹); the non-oligomer pathway for \$\Delta\$ exon3 responds to that small drop, whereas the cage-based pathway in WT is comparatively K⁺ independent. We have added a paragraph in the Discussion to make this reasoning more explicit (**lines 435-442**).

We have also included the total NLRP3s expression from the two THP-1 cell lines (new **Figure 2E**). NLRP3 (Δ exon3) expression is indeed more abundant. We believe this is inherent: once WT subunits assemble into the decameric cage, a trickle of auto-activation kills the host cell and caps expression, whereas the cage-deficient protein can accumulate without penalty.

To remove expression as a variable, we sorted each line into low, medium and high mNG expression bins. An immunoblot of the low and medium fraction (new **Figure S5C**) confirms virtually identical NLRP3 levels in the two genotypes. On the other hand, WT cells cannot be stabilized in the high bin cells, again hinting at cage-driven toxicity. We also note, however, that flow sorting itself stressed THP-1 cells as they released more TNF- α when compared to the pools in Figure 4G and even appeared without LPS priming (new **Figure S5D**). Of the three fractions the medium bin cells showed the most consistent baseline and comparable TNF- α levels after priming between the two THP-1 sorted cell lines, so all downstream work was done with that subset.

Matching the absolute, *endogenous* NLRP3 level is unfortunately not feasible with a lentiviral system: even the “low” bin cells lie above the native range (**Figure S5C**), but pushing expression

that low makes live-cell imaging impossible because the fluorescent signal photobleaches within a few frames and it is impossible to track NLRP3 in living cells otherwise. Lentiviral re-constitution therefore remains the only practical compromise for tracking NLRP3 dynamics in real time. Importantly, the exon-skipping approach in parental THP-1 cells (**Figure 3E-G**), which leaves the endogenous promoter intact, recapitulates the key Δ exon3 phenotype, confirming that the pathway does not depend on over-expression.

Finally, we revisited the “weaker/delayed” imiquimod concern. Using the expression-matched medium bin cells we ran two doses of nigericin (5 μ M and 15 μ M) and two of imiquimod (50 μ M and 150 μ M) at 30 min and 90 min (new **Figure S5E**). Low-dose nigericin induced identical IL-1 β in both cell lines, while high-dose nigericin still reaches the same plateau, but WT responded more quickly, just what a pre-assembled cage would do. Imiquimod behaved differently: at 50 μ M only WT secreted IL-1 β even after 90 min, whereas Δ exon3 stayed silent; at 150 μ M both genotypes converge by the later time-point. Taken together, these dose–time curves argue that the *MTOC/cage pathway is both faster and more easily triggered at low-potency stimuli*, whereas the cytosolic pathway used by NLRP3(Δ exon3) requires either a larger K⁺ drop or a stronger ligand to reach the same output. Our previous live microscopy supports this interpretation: standard imiquimod induces significantly more TGN/MTOC-centered specks than nigericin (**Figure 6F-H**), and in regular THP-1 cells, imiquimod-induced cell death is still fully blocked by nocodazole (**Figure 6I**). Altogether, these observations indicate that imiquimod is not just “slow”; at routine concentrations it truly prefers the centrosomal pathway, though an overwhelming dose can eventually recruit the non-associated route. Accordingly, we have replaced “mostly/strongly favor/key” with “predominantly/favors/relevant” (**lines 307, 311 and 433**) and have incorporated this observation into both the Results (**lines 298-301**) and Discussion (**lines 426-432**) sections.

In short, priming-independent IL-18 data, equal protein levels in the medium bin, and an extended dose–time matrix all point to the same conclusion: deleting exon 3 allowed robust, potassium-sensitive activation through the microtubule non associated pathway, while imiquimod at standard doses still favors the TGN/MTOC associated route. We thank the reviewer for prompting these clarifications; the new results appear in **Figure 2E, Figure 6D, supplementary figure S5C-E**, and the text has been updated accordingly.

Comment #22: The authors mention in line 261 that high extracellular K⁺ did only block MSU induced IL-1 β release but not pyroptosis. However, Fig. S5B clearly shows that this treatment almost completely prevents LDH release. This needs to be clarified.

Author reply #22: We thank the reviewer for highlighting this potential inconsistency. After carefully re-checking the data in **Figure S5B**, we can confirm that high extracellular K⁺ does not appreciably reduce MSU-induced pyroptosis. Rather, the data show similar, albeit comparatively low levels of LDH release in MSU-treated cells regardless of high K⁺, indicating that limited amount of cell death proceeds essentially unimpaired, even though IL-1 β secretion was blocked. We speculate that the overall low level of MSU-induced cell death rendered an assessment of the effects of high K⁺ difficult. We apologize if the figure or text was insufficiently clear and have sought to focus on the description of these results in **lines 277-278**.

Comment #23: The authors show in Fig 1H that human NLRP3 only binds select phospholipids: PA and PtdIns (3,4,5)-P3 and NLRP3 delta exon 3 failed to bind to any membrane lipids. The authors discuss (line 328) that while they show above result, mouse Nlrp3 has been shown to be more promiscuous and that this hints at important species differences. However, with the shown result, this cannot be concluded, and this assay would need to be performed in parallel with human and mouse NLRP3 for any conclusion.

Author reply #23: We thank the reviewer for flagging up this important point. We have followed the reviewer's suggestion, and the authors from Bonn have purified the mouse Nlrp3 protein to conduct parallel lipid-binding assays in their lab (the initial analysis was performed in Tübingen on protein transferred from Bonn). Their data again confirmed that NLRP3 lacking exon 3 presented a reduced ability to bind lipids. However, to our surprise, in their hands, both human and mouse NLRP3 proteins bound exactly the same lipids, including the already published PI4P³³. One possibility is that during the initial transport from Bonn to Tübingen, the proteins lose efficiency, as confirmed by the weaker binding that we observed in general in our hands. Another possibility is that the TEV digestion (to cut MBP from NLRP3 constructs) did not work properly in our hands, which may have led to masking of lipid-binding sites by the presence of MBP. Nevertheless, these in situ results, performed in the same lab where the proteins were purified, give us strong confidence that both mouse and human NLRP3 proteins exhibit similar lipid-binding behavior, thus unifying the pathways between both species and further confirms that NLRP3 exon 3-deficient protein exhibit impaired lipid binding. Accordingly, a revised **Figure 1H** and a new **Figure S2D** has been added and the manuscript text modified accordingly in the Results (lines **159-163**) and Discussion (lines **393-394**) sections. As our collaborators in Bonn still observed a weak lipid binding of the exon 3 deficient NLRP3 protein we have modified „unable to bind“ to „reduced ability to bind“ (e.g, lines **89, 163**). We thank the reviewer for the concern raised as this allowed us to revisit and repeat our approach and improve the reliability and consistency of our results

Comment #24: The analysis of human neutrophils and evidence for only a microtubule-independent response in contrast to THP-1 cells misses a positive control of macrophages treated in parallel. This would need to be primary peripheral blood derived macrophages from the same donor differentiated in parallel from the same blood donation.

Author reply #24: We sincerely thank the reviewer for suggesting this approach to further investigate the existence of both TGN-associated and non-associated NLRP3 activation pathways in primary neutrophils and monocyte-derived macrophages from the same blood donor. This has helped us improve the clarity and consistency of our results. We recently demonstrated in another matched analysis that these cell types can respond very differently in terms of NLRP3 regulation, although their dependence on factors such as ASC and caspase-1 remains consistent⁴. A total of four blood donors were included in this analysis. For each donor, neutrophils and monocytes were isolated simultaneously, with monocytes subsequently differentiated into macrophages over the course of one week, as detailed in the Methods sections. IL-1 β release was then measured in parallel in neutrophils and monocyte-derived macrophages from the same donor. Due to intra-donor variability, we chose to present the data separately for each donor to avoid overinterpretation of the results (new **Figure 7B**). The description of these results can be found in lines **325-330**. We also would like to add that additional neutrophil samples were now also investigated by microscopy, to prove the disruption of the microtubular network by inhibitors and also the distance between MTOC (pericentrin) and ASC specks (ASC staining), see lines **331-333** and new **Figure 7D-E**.

Comment #25: Fig 6I: I believe the authors mislabeled treatment conditions of colchicine, nocodazole and nigericin. As shown, it does not make any sense and I believe line 2 rather than line 4 should be nigericin.

Author reply #25: We appreciate the reviewer drawing attention to this labeling mistake. We have corrected Figure 6I (now **Figure 7C**) so that each treatment condition now accurately corresponds to the correct sample (i.e., line 2 is indeed nigericin). We apologize for the confusion this caused and confirm that this amendment clarifies the data presentation without altering our conclusions.

Comment #26: Methods describe the isolation of primary human peripheral blood derived macrophages, but these cells appear to not have been utilized in this manuscript and the method section should be removed.

Author reply #26: We thank the reviewer for noting this detail and apologize if the description in the Results was not clear enough. As was shown in previous Figure S3C, now included as main Figure 7F, we did in fact employ primary human macrophages to confirm TGN-associated speck formation, thereby supporting the physiological relevance of our observations. For these important results not to be overlooked too easily we decided to integrate them into the new Figure 7 which also shows additional data from other primary human cells. Please refer to lines 333-334 of the revised manuscript.

Reviewer 3

In this paper the authors essentially contend that the ability of NLRP3 to form a cage dictates the path of inflammasome activation. The authors base their findings on use of an exon3 deleted version of NLRP3 expressed in THP1 cells that does not colocalise to the TGN or the MTOC but forms inflammasomes in response to nigericin, but not the K⁺ efflux independent imiquimod. The research area is topical and relevant, but further evidence and significant revising of the manuscript is required before this paper can be accepted. Suggestions are as follows:

Comment #27: Some further explanation/analysis is required as to why WT NLRP3 binds to PA and PI3P3 but not PI4P as previously reported (e.g., PMID: 30487600).

Author reply #27: We thank the reviewer for highlighting this important point and for giving us the opportunity to clarify it, as addressed in comment #23 above. We note that at present it is difficult to fully reconcile lipid binding data and subcellular localization data. A full exploration is beyond the current scope, especially as our emphasis was on exploring the lipid-independent novel pathway.

Comment #28: How do the authors consolidate their observation of ‘smaller ASC puncta’ (Fig S3) with previous work using cryo-light microscopy and cryo-electron tomography to show the ASC speck as a network of filaments (PMID: 37945612), or indeed the superresolution microscopy of endogenous ASC in BMDMs (PMID: 38047065)?

Author reply #28: This appears to be a general conundrum in the field, namely that different techniques, e.g. cryo-EM of purified proteins, EM-tomography and different microscopy settings report a different size and shape of what are commonly called “ASC specks”. Several technical considerations explain why we detect multiple, “smaller” ASC-positive foci surrounding the main speck, whereas the cited high-resolution studies report a single, filamentous lattice.

- **Imaging resolution and timing:** our “puncta” were recorded by live-cell confocal microscopy (≈200 nm lateral resolution). At this resolution, 10–15 nm ASC filaments resolved by cryo-ET cannot be visualized as continuous strands; only the thicker filament bundles appear, and the inter-bundle gaps are rendered as discrete spots. In our time-lapse movies the smaller puncta are transient: within minutes they coalesce into the larger central speck, consistent with those early structures being assembly intermediates of the filament lattice subsequently captured by cryo-ET and super-resolution.
- **MTOC-distal vs. MTOC-proximal specks:** both high-resolution studies detected only the cytoplasmic (i.e., MTOC-distal) speck and specifically noted their inability to locate the speck at the centrosome. Our work shows that the MTOC-proximal assembly naturally consists of one main ASC core plus multiple peripheral ASC inclusions—exactly the pattern

we call “smaller puncta”. In contrast, the MTOC-distal speck forms as a single compact platform in the cytosol and does not display these satellite foci. Therefore, the structures resolved by cryo-ET and STORM probably correspond to the MTOC-distal architecture; our confocal images, on the other hand, capture both architectures. Whether these two assemblies would exhibit additional structural differences when subjected to cryo-ET or STORM remains speculative.

We have inserted clarifications in the revised manuscript (**Discussion lines 363-376**) to explain how our observations differ with the ultrastructural literature. As much as we have sought to reconcile these sets of data generated by complementary but also distinct methods, it would be outside the scope of this study to harmonize all aspects of the observed differences.

Comment #29: To conclude that an active inflammasome is formed at the MTOC colocalization with the ASC speck MUST be shown. It is known from published work that NLRP3 can go to the MTOC but the ASC speck is not observed there (this reference was overlooked (PMID: 39117604)). This must be discussed and data presented showing colocalization NLRP3/ASC/and gamma tubulin. This should be relatively easy to test, and is somewhat surprising that these data are not included.

Author reply #29: We apologize that the reference was overlooked – probably because the paper appeared in August 2024 after our first submission to the journal (July 2024). As we are in good exchange with the authors, we are more than happy to reference this work, which was also useful to address the reviewer’s comment (see below). We do take the reviewer’s point that this is a vital analysis to show. Consequently, we performed experiments staining for ASC as before using the previously used mouse monoclonal antibody). Unfortunately, the reliable tubulin antibody was also raised in mice and other tubulin antibodies did not perform well in IF. Hence, based on the work by Bai *et al*⁶, we stained the MTOC using pericentrin (PCTN) antibodies. The resulting data are shown in new **Figure S3C** and demonstrate that staining for the MTOC confirmed the localization of NLRP3-ASC specks within this region (see **lines 196-198**). The analysis was then also extended to primary neutrophils. For primary macrophages ASC was stained together with RCAS1 as a TGN marker (**Figure 7F**). As much as we would have liked to co-stain NLRP3, we would like to remind that so far no studies were able to capture endogenous NLRP3 specks from primary cells in IF but with the new data the localization of the ASC speck seems unequivocally confirmed.

Comment #30: The authors mention, when describing figure 4, that NLRP3 is recruited to endosomal-like structures that traffic to the MTOC. This observation should be validated with colabelling for endosomal markers.

Author reply #30: We agree that a definitive assignment would require co-staining with canonical endosomal markers. Because detailed characterization of this highly transient state between TGN and MTOC was not the objective of the present study—and in view of the same concern raised by Reviewer #1 (comment #8)—we have replaced the term “endosomal-like structures” with the neutral term “vesicular structures” in the text (**line 235**) and figure legend 8. This wording better reflects what we directly observed by live-cell imaging without implying a specific organelle identity.

Comment #31: The work presented on neutrophils is not enough to conclude that the proposed pathways are the same and further characterisation in these cells should be completed.

Author reply #31: We thank the reviewer for the suggestion and have further characterized these cells accordingly. Specifically, we used matched cell populations, confirmed microtubule network disruption, and performed additional immunofluorescence analyses. These new data are presented in **Figure 7B, D** and Supplementary **Figure S6A**, and are described in the newly added manuscript section: “*Human iPSC-microglia and macrophages activate both NLRP3 pathways, while*

neutrophils rely on the MTOC-independent one." We also like to refer the reviewer to Author Reply #6 for a related comment raised by Reviewer 1 but hope that this additional data clearly confirms the dominance of a non-MTOC-associated pathways in primary human neutrophils, the most dominant human leukocyte population.

Comment #32: The second sentence in the discussion states that the TGN to MTOC is the 'accepted' localisation pattern of NLRP3 activation. This is not the case and the statement is misleading.

Author reply #32: We agree that describing the TGN-to-MTOC route as "accepted" possibly overstates the consensus in the field. With "accepted" we meant to express, albeit suboptimally, that several influential papers have supported this model and at the recent EMBO Workshop "The inflammasomes" 2025 Munich this pathway was considered valid. But other studies—and our own data—demonstrate alternative activation sites. We have therefore re-worded the sentence to read: "Our goal was to elucidate the interplay between different oligomeric states of NLRP3 and the widely cited model of a spatiotemporal maturation of the NLRP3 inflammasome, initiating at the TGN and terminating at the MTOC" (lines 343- 345). The revised phrasing acknowledges the model's prominence without implying universal acceptance and sets the stage for our demonstration of an additional, TGN/MTOC-associated pathway.

Comment #33: The last sentence of the first paragraph of the discussion is not accurate. ASC oligomerisation has been observed on endosomes (PMID: 36443515).

Author reply #33: We thank the reviewer for pointing this out. PMID 36443515³⁴ indeed reports ASC polymerisation on endosomal membranes; however, that study used HeLa cells engineered to over-express only the ASC PYD, not full-length ASC, and the observations were made in a non-myeloid background. To avoid over-generalisation, we have revised the sentence to read (lines 347-348): "To date, endosome-associated ASC oligomerization has not been demonstrated for endogenous, full-length ASC in myeloid cells yet." This wording acknowledges the existing report while clarifying its experimental context and preserves the accuracy of our statement for the immune-cell systems studied here.

References cited in the point-by-point reply

1. Mateo-Tórtola, M. *et al.* Non-decameric NLRP3 forms an MTOC-independent inflammasome. *bioRxiv* 2023.07.07.548075 (2023) doi:10.1101/2023.07.07.548075.
2. Boršić, E. *et al.* Clustering of NLRP3 induced by membrane or protein scaffolds promotes inflammasome assembly. *Nature Communications* **16**, (2025).
3. Saller, B. S. *et al.* Acute suppression of mitochondrial ATP production prevents apoptosis and provides an essential signal for NLRP3 inflammasome activation. *Immunity* **58**, 90-107.e11 (2025).
4. Leal, V. N. C. *et al.* Bruton's tyrosine kinase (BTK) and matrix metalloproteinase-9 (MMP-9) regulate NLRP3 inflammasome-dependent cytokine and neutrophil extracellular trap responses in primary neutrophils. *Journal of Allergy and Clinical Immunology* **155**, 569–582 (2024).
5. Nie, L. *et al.* Consecutive palmitoylation and phosphorylation orchestrates NLRP3 membrane trafficking and inflammasome activation. *Mol Cell* **84**, 3336-3353.e7 (2024).
6. Bai, S., Martin-Sanchez, F., Brough, D. & Lopez-Castejon, G. Pyroptosis leads to loss of centrosomal integrity in macrophages. *Cell Death Discov* **10**, (2024).
7. Williams, D. M. & Peden, A. A. S-acylation of NLRP3 provides a nigericin sensitive gating mechanism that controls access to the Golgi. *Elife* **13**, (2024).
8. Glück, I. M. *et al.* Nanoscale organization of the endogenous ASC speck. *iScience* **26**, (2023).

9. Liu, A. Y. *et al.* Title : Cryo-electron tomography of NLRP3-activated ASC complexes reveals organelle co-localization. (2023).
10. Bornancin, F. & Dekker, C. A phospho-harmonic orchestra plays the NLRP3 score. *Front Immunol* **14**, (2023).
11. Schmacke, N. A. *et al.* IKK β primes inflammasome formation by recruiting NLRP3 to the trans-Golgi network. *Immunity* **55**, 2271-2284.e7 (2022).
12. Sylvain, A. *et al.* A cereblon (CRBN) molecular glue degrader of NIMA-related kinase 7 (NEK7) reveals a context-dependent role in NLRP3 inflammasome activation. *bioRxiv* 2024.11.06.622079 (2024) doi:10.1101/2024.11.06.622079.
13. Hoss, F. *et al.* Alternative splicing regulates stochastic NLRP3 activity. *Nat Commun* **10**, 3238 (2019).
14. Tuladhar, R. *et al.* CRISPR-Cas9-based mutagenesis frequently provokes on-target mRNA misregulation. *Nat Commun* **10**, (2019).
15. Andersson, R. *et al.* An atlas of active enhancers across human cell types and tissues. *Nature* 2014 507:7493 **507**, 455–461 (2014).
16. Ernst, J. *et al.* Mapping and analysis of chromatin state dynamics in nine human cell types. *Nature* 2011 473:7345 **473**, 43–49 (2011).
17. Javierre, B. M. *et al.* Lineage-Specific Genome Architecture Links Enhancers and Non-coding Disease Variants to Target Gene Promoters. *Cell* **167**, 1369-1384.e19 (2016).
18. Zhang, S. *et al.* Allele-specific open chromatin in human iPSC neurons elucidates functional disease variants. *Science (1979)* **369**, 561–565 (2020).
19. Groß, C. J. *et al.* K + Efflux-Independent NLRP3 Inflammasome Activation by Small Molecules Targeting Mitochondria. *Immunity* **45**, 761–773 (2016).
20. Magupalli, V. G. *et al.* HDAC6 mediates an aggresome-like mechanism for NLRP3 and pyrin inflammasome activation. *Science (1979)* **369**, (2020).
21. Misawa, T. *et al.* Microtubule-driven spatial arrangement of mitochondria promotes activation of the NLRP3 inflammasome. *Nat Immunol* (2013) doi:10.1038/ni.2550.
22. Hafner-Bratkovič, I. *et al.* NLRP3 lacking the leucine-rich repeat domain can be fully activated via the canonical inflammasome pathway. *Nat Commun* **9**, 5182 (2018).
23. Peng, L. *et al.* Chicken cathelicidin-2 promotes NLRP3 inflammasome activation in macrophages. *Vet Res* **53**, 69 (2022).
24. Wang, B. *et al.* Hypervirulent FAdV-4 infection induces activation of the NLRP3 inflammasome in chicken macrophages. *Poult Sci* **101**, 101695 (2022).
25. Li, J.-Y. *et al.* The zebrafish NLRP3 inflammasome has functional roles in ASC-dependent interleukin-1 maturation and gasdermin E-mediated pyroptosis. (2019) doi:10.1074/jbc.RA119.011751.
26. Gangopadhyay, A. *et al.* NLRP3 licenses NLRP11 for inflammasome activation in human macrophages. *Nat Immunol* **23**, 892–903 (2022).
27. Próchnicki, T. *et al.* Mitochondrial damage activates the NLRP10 inflammasome. *Nat Immunol* **24**, 595–603 (2023).
28. Balon, K. & Wiatrak, B. PC12 and THP-1 Cell Lines as Neuronal and Microglia Model in Neurobiological Research. *Applied Sciences* 2021, Vol. 11, Page 3729 **11**, 3729 (2021).
29. Kaur, G. & Dufour, J. M. Cell lines: Valuable tools or useless artifacts. *Spermatogenesis* **2**, 1–5 (2012).
30. Ohto, U. *et al.* Structural basis for the oligomerization-mediated regulation of NLRP3 inflammasome activation. *Proceedings of the National Academy of Sciences* **119**, (2022).
31. Matico, R. *et al.* Navigating from cellular phenotypic screen to clinical candidate: selective targeting of the NLRP3 inflammasome. *EMBO Mol Med* **17**, (2024).
32. Andreeva, L. *et al.* NLRP3 cages revealed by full-length mouse NLRP3 structure control pathway activation. *Cell* **184**, 6299-6312.e22 (2021).
33. Chen, J. & Chen, Z. J. PtdIns4P on dispersed trans-Golgi network mediates NLRP3 inflammasome activation. *Nature* **564**, 71–76 (2018).
34. Zhang, Z. *et al.* Distinct changes in endosomal composition promote NLRP3 inflammasome activation. *Nat Immunol* **24**, 30–41 (2023).

Reviewer Fig. 1: NEK7 degrader

Figure 1. Effect of NEK7 degradation in THP-1 NLRP3-mNG reconstituted cells. **A)** Representative immunoblot of NEK7 expression levels from WT Null 2, NLRP3 WT-mNG and NLRP3 Δ Exon3-mNG THP-1 pre-incubated with either DMSO or NEK7-specific degrader NK7-902 prior to LPS priming following different stimulations, as shown in the figure. **B)** Cell culture supernatants were collected and hIL-1 β levels were measured by ELISA. Each dot represents one biological replicate.

Reviewer Fig. 2: CRISPR Cas9-mediated exon 3 deletion - caveats

Figure 2: CRISPR Cas9-mediated NLRP3 exon 3 deletion. **A)** Genomic DNA was purified from THP-1 CRISPR NLRP3 exon 3 KO pool cells (pool) and THP-1 CRISPR NLRP3 exon 3 KO single clones 1 to 6. Knockout of NLRP3 exon 3 and the merge of NLRP3 exon 2 with exon 4 was analyzed by PCR using one primer pair annealing to exon 3 and exon 4 and a second primer pair annealing to exon 2 and exon 4. The expected amplicon lengths for the exon 3 - exon 4 product and the exon 2 - exon 4 (merged) product are 660 bp and 870 bp, respectively. **B)** Merge of exon 2 with exon 4 was verified by Sanger Sequencing of the 870 bp amplicon. Sanger Sequencing was done by Microsynth AG. **C)** Whole cell lysates of THP-1 CRISPR NLRP3 exon 3 k.o. single clones 1, 2, 4, 5 and 6 and of their parental THP-1 NLRP3 WT cells were subjected to SDS-PAGE and immunoblotted for NLRP3. **(D)** THP-1 CRISPR NLRP3 exon 3 KO single clones 2, 4, and 5, as well as NLRP3 WT cell line were differentiated with 100 ng/mL PMA for 24 h, rested for 48 h, then primed with 10 ng/mL LPS for 4 h and then stimulated with either 10 μM Nigericin (NIG) or 100 μM Imiquimod (IMQ) for 1 h. MCC950-treated cells were pre-incubated 30 min before stimulation at 10 μM. Then, IL-1β was measured in the supernatants. **E)** RT-qPCR showing one example of endogenous skipping of the *NLRP3* exon 3 in PMA-differentiated THP-1 cells. Left, fold change of NLRP3 exon 3 mRNA expression. Middle, fold change of NLRP3 exon 7 mRNA expression. Right, ratio of skipped exon 3 compared to total NLRP3 mRNA (using exon 7 as negative control). **F-G)** TNF-α and IL-1β release from one experiment of THP-1 WT treated as in E and further stimulated with nigericin in the case of IL-1β measurement.

Reviewer Fig. 3: Human vs Mouse NLRP3 TGN localization

Figure 3. A) Representative confocal microscopy images of HeLa cells transfected with human or mouse NLRP3 WT, both C-t tagged with mNG (green), and B) also co-transfected with the TGN marker, TGN46-mCherry (red). Nuclei were stained with Hoechst 33342 (blue). Scale bar 10 μ m, N=3 independent experiments.

Reviewer #1 (Remarks to the Author):

I thank the authors for addressing most of my previous comments. The additional data using primary cells are particularly compelling.

We sincerely thank the reviewer for the positive feedback on these results and for considering other comments as satisfactorily addressed.

However, one issue remains inadequately addressed: the analysis of the delta-exon3 variant upon exogenous expression.

(a) Regarding the genetic deletion of exon 3 in THP1 cells: I commend the authors for performing the critical experiment of genetically deleting exon 3 (along with the two adjacent introns) in THP1 cells and functionally analyzing multiple clones. The results indicate that two out of three delta-exon3 clones do not show substantial responses, while one clone exhibits a significantly reduced (10-fold lower) IL-1 β response compared to wild-type cells (R-Fig. 2D). I disagree with the authors' conclusion that these data demonstrate functionality of the delta-exon3 variant when endogenously expressed. I recommend analyzing all clones (including the remaining clones 1 and 6) using an IL-18 ELISA and activating a different inflammasome (AIM2, NLRC4, CARD8). This experiment should identify if indeed priming status or general ability for inflammasome activation is different between these clones or if the delta-exon3 variant is slightly impaired. I suggest including these additional data in the main manuscript. Confirmation of a slight functional impairment or delay of the endogenously expressed delta-exon3 variant would not detract from the authors' overall conclusions, as the variant remains a valuable tool for dissecting mechanisms upon transgenic overexpression. Additionally, the authors' suggestion that the deletion of exon 3 and adjacent introns might remove essential regulatory elements affecting expression and splicing is difficult to reconcile with the clear demonstration of proper protein expression in the Western blot (R-Fig. 2C).

We appreciate the reviewer's concerns and agree that the endogenous Δ exon3 setting deserves a careful, clone-aware interpretation, and we therefore expanded the validation and functional characterization of the CRISPR-Cas9-edited THP-1 Δ exon3 clones. When we initially screened single-cell clones (original clones 1–6), we noticed that the exon3–exon4 PCR band for original clone 3 appeared as a smear rather than a clean band, which made us doubt clonality (i.e., potential mosaic editing / mixed alleles). For that reason, we did not feel comfortable treating it as a true single clone for functional conclusions. Moreover, within the time frame allocated to the previous revision we acknowledge that we were not able to conduct a comprehensive characterization of all of the remaining clones. Based on the reviewer's suggestion and the recommendations of the editor, we therefore proceeded with the remaining five clones and renamed them D1–D5 (new Fig. S4A-E) to keep the dataset consistent and clearly traceable.

Genomic DNA was purified from WT, the CRISPR-Cas9 pool and single-cell clones. We used two diagnostic PCRs: (i) primers annealing to exon 3 and exon 4, and (ii) primers annealing to exon 2 and exon 4 to detect the merged exon2–exon4 product. The expected amplicon sizes are 660 bp (exon3–exon4, WT) and 870 bp (exon2–exon4 merged, Δ exon3), respectively (Fig. S4A). Critically, we verified the exon2–exon4 junction by Sanger sequencing of the 870 bp product (Microsynth AG), confirming exon 3 deletion at the genomic level in clones D1–D5 (Fig. S4B).

As the reviewer noted, clone-to-clone behavior for IL-1 β differed substantially, so we did not want to rely only on DNA-level confirmation. We therefore performed RNA-seq on the five validated clones (D1–D5) before and after LPS treatment. The Sashimi plots (new Fig. S4C) strikingly confirm the expected splicing pattern: in the Δ exon3 clones we do not detect junction-supporting reads that connect exon 2 to exon 3 (i.e., no exon2–exon3 junction), and instead we observe exon2→exon4 junction reads consistent with exon 3 removal. We show coverage tracks across exons 2–4, and for clarity we display the Δ exon3 clones aggregated (mean counts across D1–D5), with junction arcs labeled by supporting read counts.

To understand whether some clones might have broader transcriptional differences unrelated to NLRP3 editing, we performed PCA on size-factor–normalized whole-transcriptome read counts after arcsinh transformation ($\text{asinh}(\text{count}/64)$); libraries contained 54–66M reads per sample (Fig. S4D). In both basal and LPS-treated conditions, clones D3 and D4 cluster closest to WT, i.e., they are transcriptionally the most WT-like. This is why D3/D4 stood out as the best candidates for deeper mechanistic follow-up, while we still report the full D1–D5 panel for the functional comparisons.

We also confirmed NLRP3 protein expression by Western blot (Fig. S4F). Notably, exon 3 encodes ~40 amino acids; therefore, the predicted size difference between WT and Δ exon3 is ~4–5 kDa. Accordingly, the band shift is subtle but reproducible and consistent with exon 3 deletion. Importantly, we agree with the reviewer’s logic that clear protein expression makes it unlikely that the phenotype is explained simply by “loss of expression” and concede our initial interpretation may have been premature.

Following the reviewer’s suggestion and after characterizing the clones more comprehensively, we therefore measured IL-18 release to reduce confounding by LPS-priming-dependent IL1 β induction. In parallel, because we had already observed in the lentiviral overexpression system that the MTOC-distal pathway is slower, we have further extended nigericin stimulation to 4 hours (in addition to only 1 hour). The rationale is simple: kinetics depend on the effective concentration of inflammasome components, and in an endogenous setting the starting levels are expected to be lower than in lentiviral reconstitution, so a delayed response is easier to miss at early time points.

We also extended LPS priming to 18 hours based on data from the Latz group ¹: They suggested that prolonged priming can “license” NLRP3 variants that appear compromised under standard conditions. Notably, the usually dysfunctional Δ exon5 can regain activity after prolonged priming. Motivated by that precedent, we asked whether Δ exon3 clones might similarly show a kinetic/threshold effect rather than an absolute loss of function.

What we observed fits that idea very well. In our ELISA data, Δ exon3 clones (D1–D5) show essentially no IL-18 release at 1 hour of nigericin stimulation, but they release when stimulation is extended 4 hours (new Fig. 4H). Even more surprisingly, after prolonged LPS priming (18 hours), inflammasome activity can be restored to near-WT levels — especially in the clones that are transcriptionally closest to WT (D3 and D4) (new Fig. 4H). This matches our overexpression-based conclusion that Δ exon3-driven (MTOC-distal) inflammasome activation is delayed rather than fundamentally absent. As shown by immunoblotting (reviewer Figure 1 / our updated blots), this “licensing” after 18 hours of LPS cannot be explained simply by increased NLRP3 (or IL-1 β) protein expression, so we do not know yet the mechanism. We would therefore like to keep the interpretation strictly at the level supported by the data (delayed kinetics / altered licensing). As the data does not offer additional insights, we suggest not to include it unless stipulated by the reviewer and editor.

Reviewer Figure 1. Representatives immunoblot of CRISPR-Cas9 Δ exon3 clones D3 and D4 showing NLRP3 and IL-1 β expression at 3 and 18 hours LPS treatment (N=2).

Despite slight differences, it is easy to reconcile the results from the two endogenous approaches (morpholino skipping vs. CRISPR-Cas9 clones). That the exon-skipping (morpholino) strategy yielded robust inflammasome activation already at 1 hour, not later like the CRISPR-Cas9 clones, is most plausibly explained by (i) incomplete efficiency in the exon-skipping setup (i.e., not achieving 100% Δ exon3 at the protein level in every cell, leaving ~20% residual WT NLRP3 that can contribute early), and (ii) the CRISPR-Cas9-derived clones showing slightly reduced NLRP3 levels overall, which can shift kinetics/thresholds; (iii) slight differences in kinetics contributed by different parental THP-1 clones. Nevertheless, both approaches are consistent in the main conclusions of our work: that (i) exon 3 deletion or skipping is functional; that (ii) deletion of exon 3 delays NLRP3-dependent cytokine release.

To confirm that CRISPR-Cas9 deletion of exon 3 also had similar consequences for the subcellular localization of inflammasome formation, we analyzed MTOC association (pericentrin/PCTN staining) of ASC specks by fluorescence microscopy (new Fig. S4G), followed by unbiased quantification of microscopy images (new Fig. S4F, quantified in S4H). In WT cells, we found that at 1 hour upon nigericin stimulation many specks are MTOC-associated, while at 4 hours the majority become MTOC-distal (cells with MTOC-associated specks would be lost from the culture by that point). In contrast, Δ exon3 cells showed predominantly MTOC-distal specks at 4 hours, consistent with this time point capturing the early response window for the delayed Δ exon3 kinetics (Fig. S4F, G). This again mirrors what we see in the lentiviral reconstitution system (Fig. 3) and supports the same conceptual conclusion: Δ exon3 NLRP3 preferentially engages the MTOC-distal mode but does so with slower kinetics in the endogenous setting.

Taken together, we believe the most consistent interpretation is that endogenous Δ exon3 is not “dead,” but rather shows a clear delay/threshold shift. We genuinely thank the reviewer for pushing us to tighten this part; we feel it has considerably strengthened the manuscript.

(b) In their response, the authors justify the exon-skipping morpholino approach by referencing PMID: 31324763. I would like to highlight a crucial difference: exon 5 skipping described in PMID: 31324763 completely abrogates the response, whereas exon 3 skipping in the current manuscript does not impair the response.

As we point out above, data from the Latz lab published in the PhD thesis of Florian Hoss, Bonn¹, indicate that exon 5 skipping regains functionality upon longer stimulation times. Thus, the perceived discrepancy between exon 5 and exon 3 skipping is less striking. We added the point that (only) “early” IL-1 was affected as expected to the manuscript in line 228 but would consider it distracting to devote further space to a discussion of the exon 5 phenotype.

Thus, the authors' conclusion critically depends on the efficiency of the exon-skipping approach. Given this reliance, and particularly if the authors or editors choose not to include the exon 3-deleted THP1 clones, I strongly recommend validating the morpholino-based exon 3 skipping efficiency shown in Fig. 3E-G by Western blot. This assay is already established by the authors (demonstrated in R-Fig. 2C) and should clearly confirm that most of the NLRP3 protein corresponds to the delta-exon3 molecular weight upon morpholino treatment. The genetically edited clones could also serve as controls to confirm the correct protein size.

We appreciate the reviewer's point here and agree that protein-level validation is the cleanest way to demonstrate exon-skipping efficiency. We have now added this experiment. Specifically, we performed Western blotting after exon-3 skipping in THP-1 cells (endogenous NLRP3) with our designed morpholino and now include these data in Fig. S3D. We can clearly detect a lower-migrating NLRP3 band upon exon-3 morpholino treatment, consistent with Δ exon3. This fits well with the RNA-based analysis in Fig. 3E and confirms that the AON used is able to induce skipping events leading to an altered NLRP3 protein, similar to what is observed in the CRISPR-Cas9-edited clones. Combined with the additional characterization of the CRISPR-Cas9-edited clones by ELISA and microscopy, we hope that the remaining concern of the reviewer has been resolved using complementary approaches.

One minor additional point: the wording used to describe the exon skipping approach is confusing. For example, Fig. 3E refers to "Exon 3 skipped mRNA,". To me, the wording "unskipped mRNA" makes more sense because the assay detects exon 3 in cDNA. Consistently, the signal decreases upon exon 3-targeting morpholino treatment.

We also agree that our wording was confusing and we have fixed it. Because the qPCR assay detects exon-3-containing cDNA, it is indeed clearer to describe the readout as “exon-3 transcript.” We therefore amended the labels/legend accordingly and explicitly state that the signal decreases upon exon-3 morpholino treatment because exon 3 is being skipped.

Overall, we hope that the new data provided (new Figs. S3D, S4C, D, F, G and 4H) will have satisfactorily addressed the remaining concern of reviewer 1. Moreover, clones D3 and D4 emerge as useful new reagents to study exclusively the MTOC-distal pathway further.

References:

- 1. Hoss, F. Alternative splicing as a regulatory mechanism of the NLRP3 inflammasome. (University of Bonn, Bonn, 2018).*